# Supporting cells orchestrate noise-induced hearing loss via a Gasdermin D-dependent signaling loop with hair cells

Lili Xiao[1,2,3,8], Jianju Liu[4,8], Yi Chen[1,2,3], Liang Xia[1,2,3], Yumeng Jiang[1,2,3], Xiaoyan Chen[5], Tianjiao Zhou[1,2,3], Yan Sun[6], Wen Lu[1,2,3], Hui Wang [1,2,3] ✉, Jian Wang[1,7] ✉, Yanmei Feng[1,2,3] ✉, Zhen Zhang [1,2,3] ✉ & Shankai Yin[1,2,3]

Noise-induced hearing loss (NIHL), a common sensory disorder, is traditionally thought to stem primarily from direct damage to sound-sensing hair cells (HCs). Here, we demonstrate that supporting cells (SCs), neighboring cells not previously implicated in NIHL pathogenesis, orchestrate hearing loss and HC degeneration through Gasdermin D (GSDMD) activation. Mechanistically, noise-induced oxidative stress in HCs triggers activation of epidermal growth factor receptor in SCs, leading to extracellular-regulated kinase phosphorylation and caspase-11-dependent cleavage of GSDMD, thereby establishing an HC-to-SC signaling cascade. Furthermore, GSDMD activation in SCs reciprocally exacerbates oxidative injury in HCs, creating a pathogenic positive feedback loop between the two cell types. Our findings uncover a central role for SCs in noise-induced hearing loss and identify GSDMD-mediated intercellular communication as a potential therapeutic target.

Noise-induced hearing loss (NIHL), the predominant form of acquired sensorineural hearing loss, has been conventionally attributed to hair cell (HC) degeneration caused by mechanical and metabolic stress[1–3]. While oxidative damage and inflammation are established central mediators of cochlear injury[4,5], it remains unclear how these two processes interact to determine HC fate. Gasdermin D (GSDMD) is a prime candidate to integrate these pathological signals. As the executor of inflammatory cell death, GSDMD forms transmembrane pores that can trigger mitochondrial-dependent oxidative stress or release pro-inflammatory mediators, such as interleukin-1β (IL-1β) and high-mobility group box 1 (HMGB1)[6–12]. Our previous finding that pharmacological inhibition of extracellular HMGB1 attenuated NIHL supports the relevance of this pathway, yet how GSDMD is activated by acoustic overstimulation and contributes to HC loss is unknown[13].

A further layer of complexity arises from context-specific GSDMD biology[8,11,12,14–16]. Beyond its canonical activation pathways, GSDMD exhibits cell type-specific cleavage and can paradoxically promote both inflammatory damage and tissue repair[9–11,14,16–19]. Here we found that GSDMD is predominantly expressed in cochlear supporting cells (SCs) —cells not classically associated with NIHL pathogenesis despite their intimate contact with HCs[20,21]. By delineating the relationship between the key inflammatory mediator GSDMD in SCs and oxidative damage in HCs, we defined an underlying intercellular signaling circuit in NIHL. We demonstrate that: (1) SC-specific GSDMD deletion protects against NIHL, (2) noise-induced oxidative stress in HCs triggers a signaling cascade via epidermal growth factor receptor/extracellular-regulated kinases (EGFR/ERK), which activates caspase-11-dependent GSDMD cleavage in SCs; and (3) activated GSDMD in SCs reciprocally exacerbates oxidative damage in HCs. Our results establish SCs as active

[1]Department of Otorhinolaryngology Head and Neck Surgery, Shanghai Sixth People's Hospital Affiliated to Shanghai Jiao Tong University School of Medicine, Shanghai, China. [2]Shanghai Key Laboratory of Sleep Disordered Breathing, Shanghai, China. [3]Otolaryngology Institute of Shanghai Jiao Tong University, Shanghai, China. [4]School of Rehabilitation Science of Shanghai University of Traditional Chinese Medicine, Shanghai, China. [5]Department of Otolaryngology, Shanghai Jiao Tong University Medical School Affiliated Ruijin Hospital, Shanghai, China. [6]Zhangjiagang TCM Hospital Affiliated to Nanjing University of Chinese Medicine, Zhangjiagang, China. [7]School of Communication Science and Disorders, Dalhousie University, Halifax, NS, Canada. [8]These authors contributed equally: Lili Xiao, Jianju Liu. ✉e-mail: wangh2005@sjtu.edu.cn; jian.wang@dal.ca; ymfeng@sjtu.edu.cn; zhangzhen1994@sjtu.edu.cn

signaling hubs in NIHL and identify GSDMD as the central mediator of a pathogenic positive feedback loop between HCs and SCs.

## Results

### *Gsdmd* global knockout (KO) attenuates NIHL in mice

Western blotting (WB) was performed on cochlear samples from *Gsdmd* KO and wild-type (WT) mice. Of the five commercially available

GSDMD antibodies tested, two were validated with effective and specific detection for cochlear tissues (Fig. 1a; Supplementary Fig. 1a). Immunohistochemistry and immunofluorescence staining both revealed that GSDMD was predominantly localized to supporting cells (SCs) within the organ of Corti, particularly pillar cells and Claudius' cells, with minimal expression in hair cells (HCs) (Fig. 1a–c; Supplementary Fig. 1b, c).

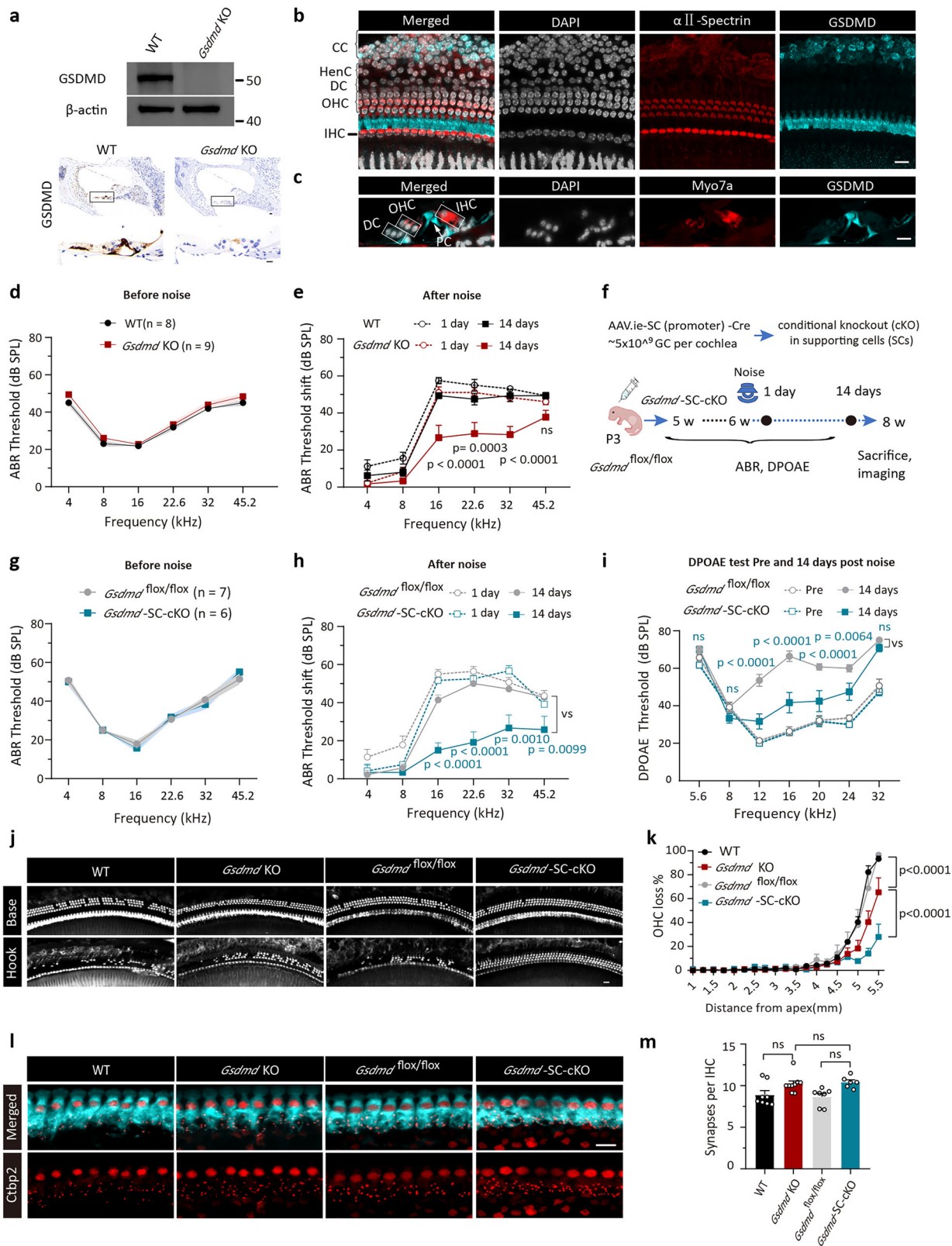

**Fig. 1 | *Gsdmd* knockout globally or in supporting cells specifically both attenuates noise-induced cochlear damage. a** Validation of *Gsdmd* knockout (KO) in cochlear tissue by western blotting and immunohistochemistry. **b** Representative image of GSDMD expression in the adult mouse cochlea. Hair cells (HCs) are labeled with αII-Spectrin (red). **c** Cochlear cross-section showing GSDMD (cyan) mainly in SCs, not HCs (Myo7a). **d** Baseline ABR thresholds of WT (black) and *Gsdmd* KO (red) mice. **e** ABR thresholds shifts in *Gsdmd* KO mice (red solid line) largely recovered by 14 d-PNE, whereas no recovery occurred in WT mice (black solid line). **f** Experimental design: AAV.ie-SC-Cre ($5 \times 10^9$ GCs/ear) was injected into the cochleae of *Gsdmd* ^flox/flox^ mice at P3. **g** Baseline ABR of *Gsdmd* ^flox/flox^ (gray) and *Gsdmd*-SC-cKO mice (cyan). **h** ABR threshold shifts largely recovered in *Gsdmd*-SC-cKO mice at 14 d-PNE (cyan), with no recovery observed in *Gsdmd* ^flox/flox^ mice (gray). **i** *Gsdmd*-SC-cKO mice showed decreased DPOAE thresholds compared to *Gsdmd* ^flox/flox^ mice at 14 d-PNE (solid line), with no significant differences at Pre-noise exposure (dashed line). **j** Representative images of Myo7a-stained cochlear basal and hook regions from WT, *Gsdmd* KO, *Gsdmd* ^flox/flox^, and *Gsdmd*-SC-cKO mice at 14 d-PNE. **k** OHC loss rate of mice in (**j**). **l** Representative images of Myo7a (cyan) and Ctbp2 (red) staining in the 32 kHz regions from mice at 14 d-PNE. **m** Statistics of samples shown in (**l**) from 32 kHz region, where OHC loss is minimal or indeed absent, showing no statistical differences in synapse density (averaged from eight IHCs per cochlea) across groups. A circle represents one mouse. Data are shown as means ± SEM. 6–9 mice for each group. Sample sizes are labeled in panels. Statistical analyses were performed using two-way (**d**, **e**, **g**–**i**, **k**) and one-way (**m**) ANOVA with Bonferroni post hoc test. Scale bar = 10 μm. ns, no statistical difference (*p*-value > 0.05). WT wild type, OHC outer hair cell, IHC inner hair cell, PC pillar cell, DC Deiter cell, CC Claudius' cell, HenC Hensen's cell, SC supporting cell, PNE post-noise exposure, GCs genome-containing particles, P3 postnatal day 3.

Auditory brainstem response (ABR) testing showed no significant differences in baseline hearing thresholds between *Gsdmd* KO and WT mice (6 weeks old), indicating that GSDMD deficiency did not impact normal auditory function (Fig. 1d). Subsequently, these mice were exposed to octave band noise (8–16 kHz) at 100 dB SPL for 2 h, which was used in all the experiments involving noise exposure in our present report. ABR testing 14 days post-exposure (14 d-PNE) revealed significantly smaller threshold shifts in *Gsdmd* KO mice at 16, 22.6, and 32 kHz compared to WT controls (Fig. 1e).

### *Gsdmd* conditional knockout (cKO) in SCs rather than HCs reduces acoustic trauma

To determine whether the protective effect of *Gsdmd* KO was SC specific, we achieved SCs conditional knockout (SC-cKO) in *Gsdmd* ^flox/flox^ mice using Cre recombinase expressed via adeno-associated virus-ie (AAV-ie), a serotype with high transfection efficiency in SCs[22]. The feasibility of SC promoter loading and Cre-driven knockout was confirmed in double-fluorescent Cre reporter mice (CAG-LoxP-ZsGreen-Stop-LoxP-tdTomato). tdTomato expression in SCs, including pillar cells, with no detectable expression in HCs, confirmed SCs targeting and the non-HC effectiveness of AAV.ie-SC-Cre (Supplementary Fig. 2a–d). Injection of AAV.ie-SC-Cre into the cochleae of *Gsdmd* ^flox/flox^ mice generated *Gsdmd*-SC-cKO mice (Fig. 1f). WB analysis revealed substantially reduced GSDMD level in cochlear tissues of *Gsdmd*-SC-cKO mice, consistent with successful GSDMD ablation in SCs via immunohistochemistry test (Supplementary Fig. 2e–h).

When challenged with the noise exposure, SC-cKO mice exhibited a reduction in noise-induced threshold shifts, almost 100% abolishment of HC loss, and marked recovery of distortion product otoacoustic emissions (DPOAE) thresholds at 14 d-PNE (Fig. 1g–k). Specifically, the reduction in OHC loss was more pronounced in SC-cKO mice than in global KO mice (Fig. 1j, k). This finding suggests that GSDMD in non-SC cells may play a protective role in acoustic trauma, rather than the harmful role in SCs. Additionally, neither *Gsdmd*-KO nor SC-cKO mice showed protective effects on cochlear synapse counts (Fig. 1l, m).

To explore whether GSDMD in HCs plays a role in NIHL, *Gsdmd*-HC-cKO mice were generated using Cre recombinase expressed via the AAV.PHP.eB serotype, which has high transfection efficiency in HCs[23]. The HC promoter (reported in a previous study[24]) was loaded to ensure Cre expression specific to HCs (Supplementary Fig. 3a–e). Injection of AAV.PHP.eB-HC-Cre into the cochleae of *Gsdmd* ^flox/flox^ mice generated *Gsdmd*-HC-cKO mice. No protective effects in ABR threshold shifts, OHC loss or synapse loss were observed in *Gsdmd*-HC-cKO mice following the same noise exposure (Supplementary Fig. 3f–k).

### GSDMD is activated following noise exposure via caspase-11-dependent pathway

GSDMD activation typically occurs through proteolytic cleavage of its N-terminal fragment (GSDMD-N), which facilitates pore formation across the cell membrane, leading to inflammatory cell damage[9]. In the WB test, intensive parameters are required for visualizing the GSDMD-N band, which can lead to overexposure of the full-length GSDMD (GSDMD-FL) band. A substantial increase in cochlear GSDMD-N levels was observed at 8 h-PNE and persisted for 7 days (Fig. 2a). Intervention with disulfiram, an FDA-approved drug that inhibits GSDMD-N pore formation[25], partially attenuated noise-induced hearing dysfunction (Supplementary Fig. 4). These results suggest a role for GSDMD-mediated pore formation in cochlear damage, although disulfiram may also exert additional effects[26].

GSDMD activation is fundamentally dependent on proteolytic cleavage—mediated by caspases including caspase-1, -4, -5, -8, or -11—with distinct cleavage patterns and sites determined by cell type and pathological context[8,9,11,12,19]. Using caspase-1 and caspase-11 double-knockout (*Casp1/11* DKO) mice, we observed that noise exposure resulted in significantly reduced GSDMD-N levels compared to WT controls (Fig. 2b, e). In WT mice, noise exposure markedly elevated cleaved IL-1β—a canonical caspase-1 substrate—whereas *Casp1/11* DKO mice showed complete ablation of IL-1β processing (Fig. 2b, e). Notably, despite caspase-1's essential role in IL-1β maturation, its involvement in GSDMD cleavage was excluded by comparable GSDMD-N elevation in *Casp1* KO and WT cochleae after noise exposure (Fig. 2c, f). These results establish that caspase-11, but not caspase-1, is the non-redundant executor of cochlear GSDMD cleavage following acoustic trauma. To exclude compensatory interactions between caspases, *Casp11* KO mice demonstrated significantly reduced GSDMD-N and cleaved IL-1β levels versus WT controls post-noise exposure (Fig. 2d, g). This confirms caspase-11 as the primary mediator of GSDMD proteolysis and upstream regulator of caspase-1-dependent inflammation in noise-induced cochlear injury.

### Oxidative damage drives cochlear GSDMD activation

Previous reports have shown that oxidative stress is a crucial process in cochlear acoustic trauma[5,27,28]. Figure 3 demonstrates the time-dependent relationship between GSDMD-N and 4-hydroxynonenal (4-HNE, a marker of oxidative stress) in the cochlea following acoustic trauma. WB images (Fig. 3a), quantitative analysis (Fig. 3b), and Pearson correlation analysis (Fig. 3c) reveal a strong positive correlation between GSDMD-N and 4-HNE levels (Pearson $r = 0.917$, $p < 0.0001$). Furthermore, treatment with the antioxidant N-acetylcysteine (NAC) significantly reduced noise-induced 4-HNE and was accompanied by decreased levels of cleaved caspase-11, caspase-1, and GSDMD-N (Fig. 3d, e). Taken together, these findings suggest that caspase-11 activation occurs upstream of GSDMD cleavage in the oxidative stress pathway during noise-induced cochlear injury.

### GSDMD propagates oxidative damage to HCs via a self-reinforcing intercellular loop

While oxidative stress initiates upstream of GSDMD signaling (Fig. 3), GSDMD can also propagate oxidative damage to hair cells through a self-reinforcing intercellular loop. This is evidenced by significantly

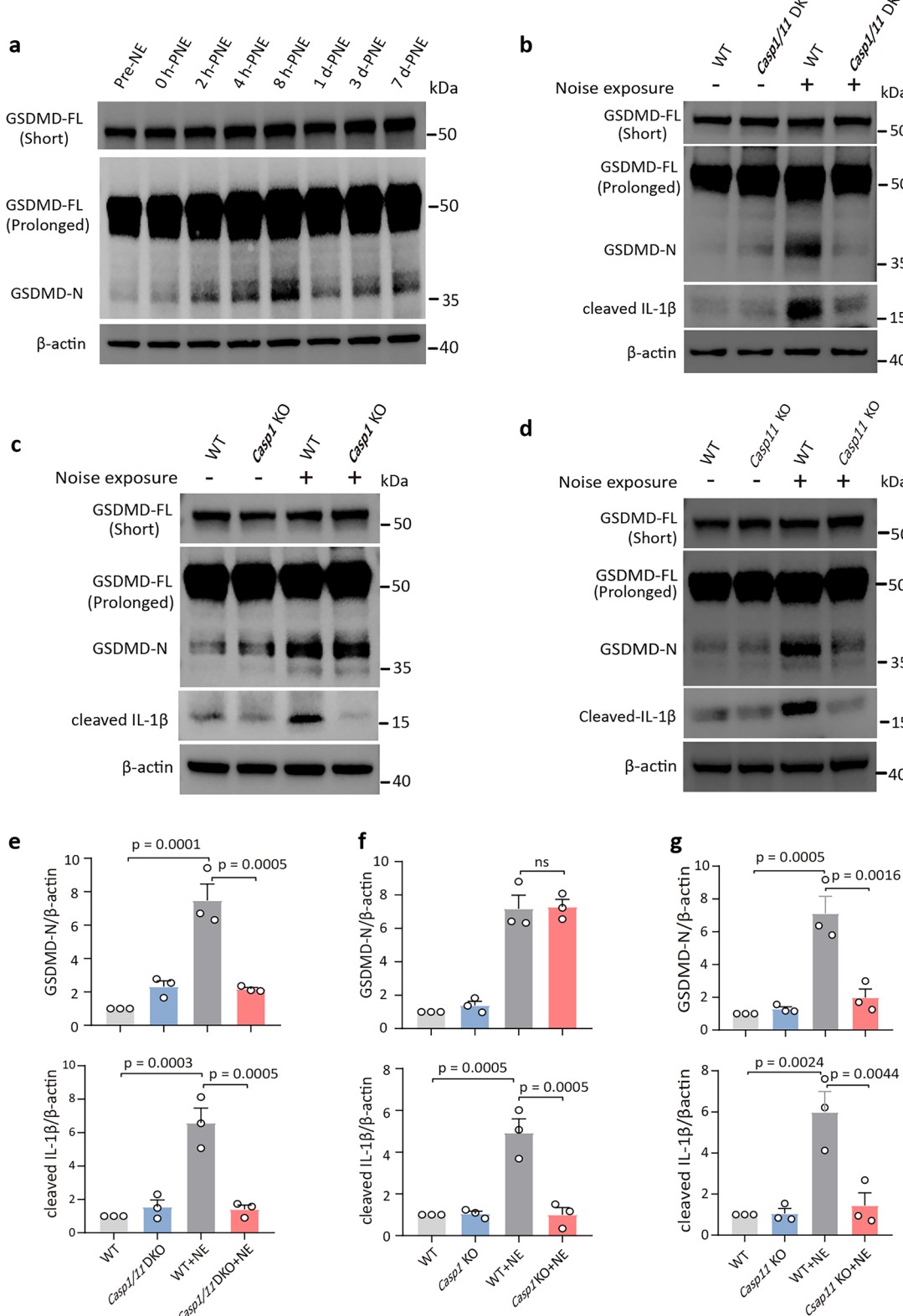

**Fig. 2 | GSDMD cleavage occurs in caspase-11 but not caspase-1 dependent pathway after noise exposure. a** Western blotting (WB) was performed to assess GSDMD full-length (FL) and N-terminal fragment (N) levels in whole cochlear lysates from adult WT mice at various time points (Pre-NE, and 0 h, 2 h, 4 h, 8 h, 1 d, 3 d, and 7 d post-noise exposure [PNE]). Detection parameters included short exposure (5 s) for visualizing GSDMD-FL and prolonged exposure (30 s) for visualizing GSDMD-N bands. **b–d** Representative WB images of cleaved GSDMD and IL-1β in cochlear tissues from *Casp1/11* DKO, *Casp1* KO, *Casp11* KO and WT controls mice

with/without noise exposure. Cochleae were collected 8 h-PNE from noise-challenged mice. **e–g** Quantitative analysis to (**b–d**): target bands were normalized to β-actin and presented as fold-change relative to the WT group (set to 1). All data are presented as mean ± SEM, *n* = 3 replicates (each point represents a sample containing four cochleae). Statistical analysis was performed using one-way ANOVA with Bonferroni post hoc test. ns not significant, WT wild type, NE noise exposure, PNE post-noise exposure.

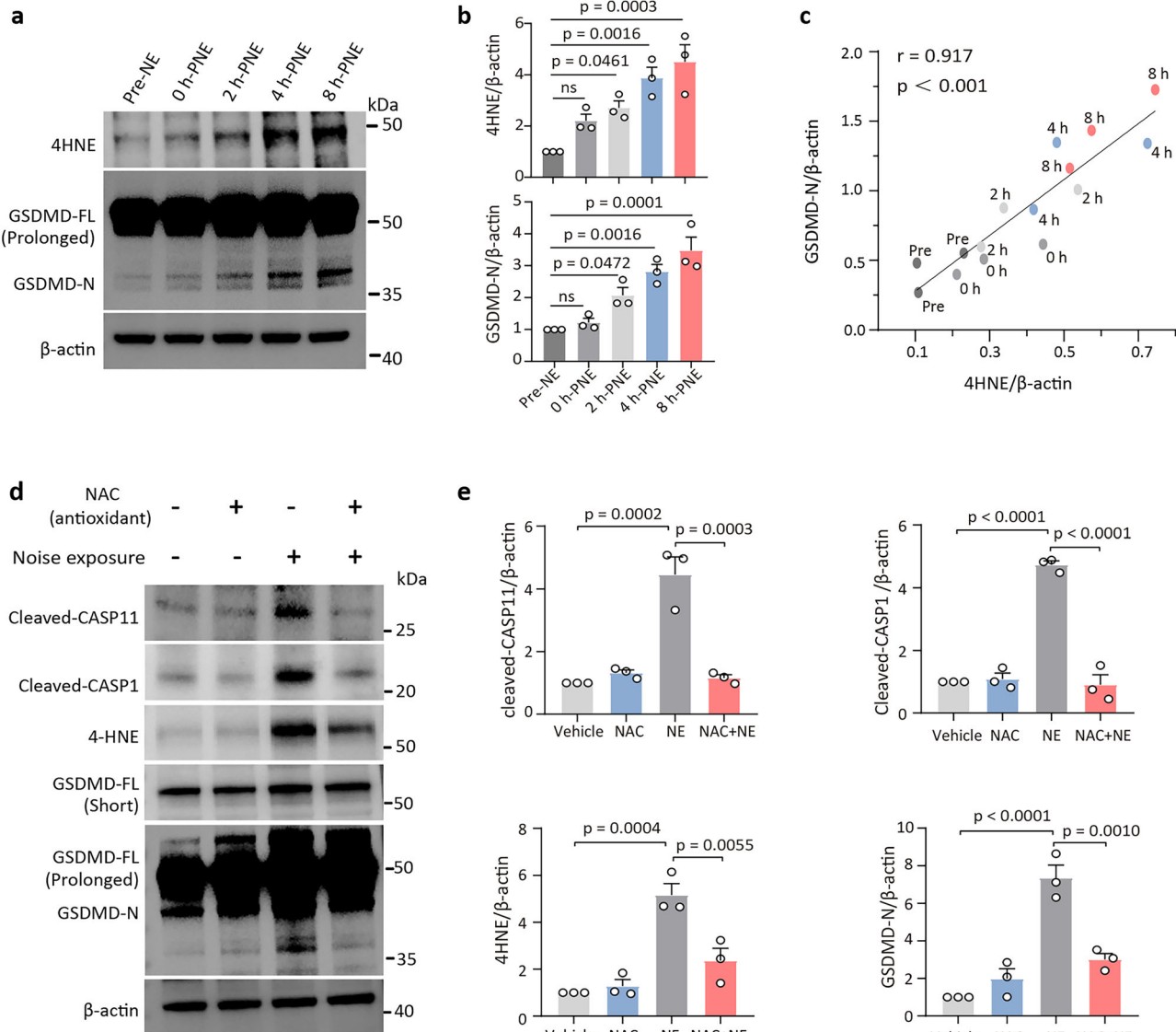

**Fig. 3 | Oxidative stress activates cochlear GSDMD signaling. a** Levels of cochlear 4-HNE and GSDMD-N increased progressively within 8 h-PNE. Detection parameters included short exposure (5 s) and prolonged exposure (30 s) to visualize GSDMD-FL and GSDMD-N bands, respectively. **b** Normalized changes in 4-HNE and GSDMD-N levels. **c** Pearson correlation analysis between 4-HNE and GSDMD-N expression at various time points following noise exposure (Pearson $r = 0.917$, $p < 0.0001$). A circle represents one biological replicate. **d**, **e** WB analysis of the indicated proteins

in cochlear tissues at 8 h-PNE, with/without treatment using the antioxidant NAC (400 mg/kg/day, intraperitoneally [i.p.] for 3 days, plus an additional dose at 2 h-PNE). Data are presented as mean ± SEM ($n = 3$ biological replicates). Statistical analysis was performed using one-way ANOVA with Bonferroni post hoc test. ns not significant, WT wild type, NE noise exposure, PNE post-noise exposure, NAC N-acetylcysteine.

---

reduced 4-HNE levels and near-abolition of cleaved caspase-11/caspase-1/IL-1β in global *Gsdmd* KO cochleae versus WT post-noise exposure (Fig. 4a, b).

Spatial analysis confirmed attenuated 4-HNE accumulation specifically in OHCs of *Gsdmd* KO mice (Fig. 4c, d). Critically, *Gsdmd*-SC-cKO reduced both cochlear 4-HNE (WB; Fig. 4e, f) and OHC 4-HNE fluorescence (Fig. 4g, h; Supplementary Fig. 5) post-noise exposure. These results establish that GSDMD activation in SCs amplifies oxidative damage in HCs, defining an SCs-to-HCs intercellular feedback loop that perpetuates noise-induced cochlear pathology.

### GPX4, but not GPX1, dysfunction primes ROS-GSDMD feedback loop formation

The glutathione peroxidase (GPX) system refers to a group of antioxidant enzymes that are critical for controlling reactive oxygen species (ROS). Among the *Gpx* family genes, deficiency in *Gpx1* or *Gpx4* in mice

is associated with congenital hearing dysfunction[29,30]. In this study, WB analysis revealed no significant noise-induced changes in GPX1 levels but detected a transient reduction in GPX4 at 2 h-PNE (Fig. 5a, b).

We achieved cochlear overexpression of GPX1 and GPX4, respectively, by using the AAV.PHP.eB vector loaded with the CMV promoter (Fig. 5c; Supplementary Fig. 6). ABR testing showed no significant differences in baseline hearing thresholds between mice with GPX1 or GPX4 overexpression and control mice, indicating that the manipulations did not affect normal auditory function (Fig. 5d). However, mice overexpressing GPX4 showed significantly smaller threshold shifts (Fig. 5f), better DPOAE preservation (Fig. 5g, h), reduced OHC loss (Fig. 5i, j), and less synapse loss (Fig. 5k, l) at 14 d-PNE, as compared to control mice. No such protection was observed in mice overexpressing GPX1 (Fig. 5f–l).

Furthermore, overexpressing GPX4 but not GPX1 markedly reduced GSDMD-N and 4-HNE accumulation induced by noise

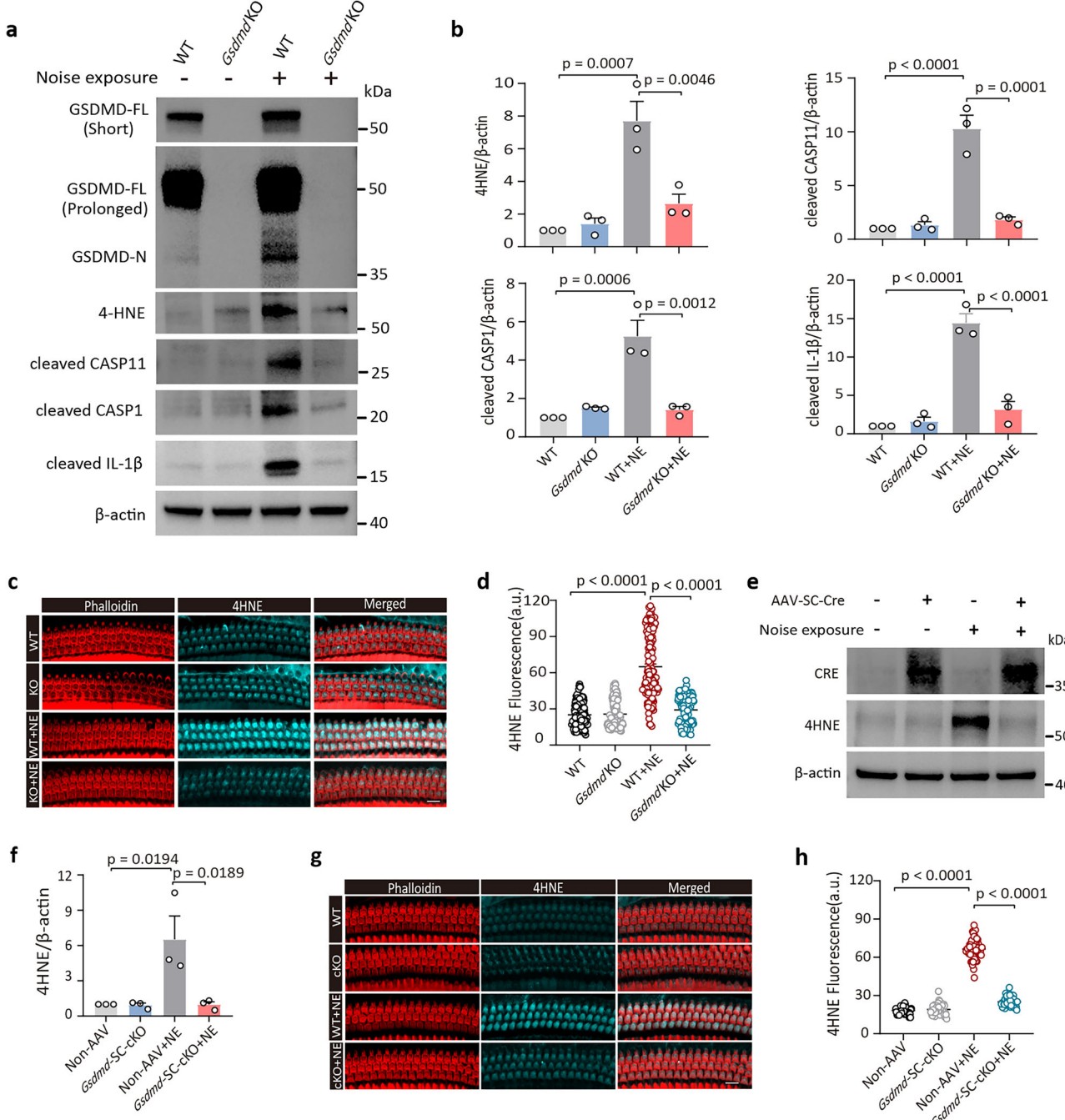

**Fig. 4 | GSDMD exacerbates noise-induced oxidative damage through a positive feedback loop. a, b** WB analysis of 4-HNE, cleaved caspase-11, caspase-1, and IL-1β levels in cochlea from WT and *Gsdmd* KO mice with/without noise exposure. Cochleae from noise-exposed mice were collected at 8 h-PNE. **c, d** Immunolabeling analysis of 4-HNE (cyan) in outer hair cells (OHCs) from WT and *Gsdmd* KO mice with/without noise exposure (scale bar = 10 μm). Cochleae from noise-exposed mice were collected 8 h-PNE. The 22.6–32 kHz frequency region was focused to analysis, and the circle in (**d**) represents an individual OHC. The confocal plane showing cytoplasm near the cuticular plate of OHCs was confirmed using phalloidin (red) staining. **e, f** WB analysis of 4-HNE in cochlear tissues from non-AAV-injected and *Gsdmd*-SC-cKO mice with/without noise exposure. Cochleae of mice challenged with noise exposure were collected at 8 h-PNE. **g** Immunolabeling of 4-HNE (cyan) in OHCs co-stained with phalloidin (red) from non-AAV-injected and *Gsdmd*-SC-cKO mice. **h** Quantitative analysis of 4HNE immunolabeling in OHCs. Each circle represents an individual OHC. Data are presented as mean ± SEM (*n* = 3 biological replicates). Statistical analysis was performed using one-way ANOVA with Bonferroni post hoc test. WT wild type, NE noise exposure, PNE post-noise exposure.

exposure (Fig. 5m, n; Supplementary Fig. 8). These results suggest that GPX4 overexpression partially disrupts the oxidative stress-GSDMD feedback loop. In addition, GPX4 downregulation at 2 h-PNE could not be alleviated by GSDMD ablation (Supplementary Fig. 9), indicating GPX4 deficiency potentiates oxidative stress and its feedback loop with GSDMD.

## Mitigation of oxidative damage in HCs alone is sufficient to block GSDMD-associated signaling

The AAV.PHP.eB vector, when loaded with a CMV promoter, can transfect HCs with high efficiency but exhibits scattered transfection in SCs, particularly in the basal turn (Supplementary Fig. 6a–d). Here, CMV-driven global cochlear overexpression of *Gpx4* significantly

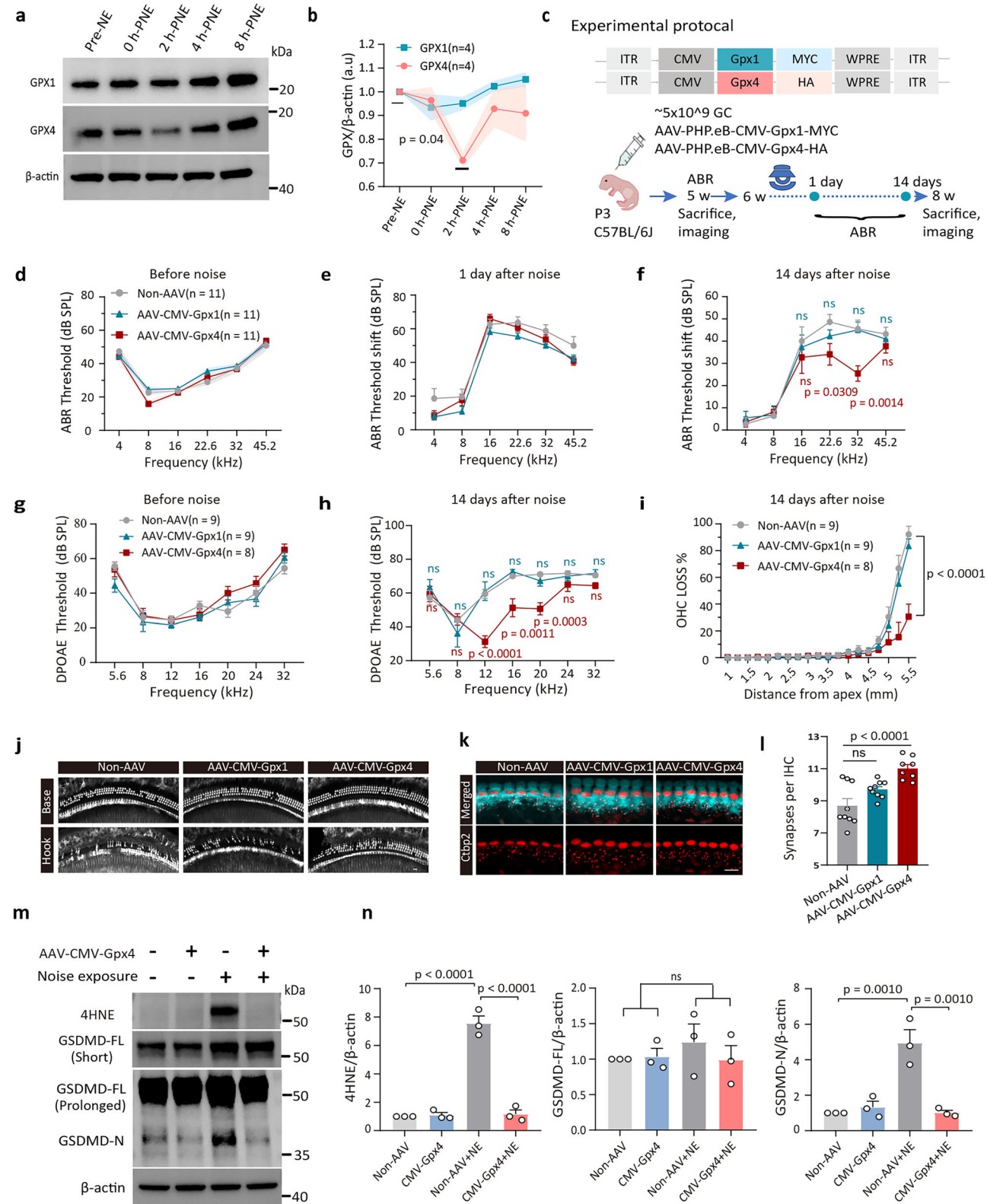

reduced noise-induced 4HNE levels in OHCs. (Fig. 6a, b). A markedly decrease in GPX4 immunoreactivity was detected in OHCs, but not SCs, at 2 h-PNE in WT mice (Fig. 6c, d, Supplementary Fig. 7), suggesting that GPX4 functional insufficiency potentiates noise-induced oxidative damage in OHCs.

To determine whether GPX4-mediated inhibition of oxidative stress in HCs regulates GSDMD activation in SCs, we generated an AAV.PHP.eB vector with an HC-specific promoter to overexpress *Gpx4*

(AAV-HC-*Gpx4* vector), thus targeting HCs exclusively (Fig. 6e). In addition to 4HNE, the AAV-HC-*Gpx4* group showed reductions in noise-induced GSDMD-N, cleaved caspase-11, cleaved caspase-1, and cleaved IL-1β levels compared to the control group (Fig. 6f, g). Notably, the reduction in GSDMD cleavage and 4HNE levels was comparable to that observed in the AAV-CMV-*Gpx4* overexpression group (Fig. 5m, n), highlighting the critical role of HCs' GPX4 in disrupting the ROS-GSDMD feedback loop. Specifically, although GSDMD is primarily

**Fig. 5 | GPX4, but not GPX1, deficiency potentiates cochlear oxidative stress and its feedback loop with GSDMD. a, b** WB analysis of GPX1 and GPX4 levels at different time points post-noise exposure (*n* = 4). **c** Schematic of the experimental design: *Gpx1* and *Gpx4* were overexpressed via transfection with AAV. PHP.eB vector driven by a CMV promoter, followed by auditory electrophysiology testing and cochlear morphological analysis. **d** Baseline ABR thresholds of non-AAV-injected mice (gray), AAV-CMV-*Gpx1* (cyan), and AAV-CMV-*Gpx4* (red) groups. **e** ABR threshold shifts in the three groups at 1 d-PNE. 11 mice for each group. **f** ABR threshold shifts at 22.6 and 32 kHz frequency at 14 d-PNE were significantly lower in AAV-CMV-*Gpx4* group (*n* = 11, red) compared with non-AAV (*n* = 11, gray) group, with no significant difference shown between the AAV-CMV-*Gpx1* (*n* = 11, cyan) group and the non-AAV group. **g** DPOAE thresholds measured in the three groups prior to noise exposure. **h** DPOAE thresholds were significantly improved in the AAV-CMV-*Gpx4* group (*n* = 8, red) compared to the non-AAV−injected mice (*n* = 9, gray) at 14 d-PNE, while the AAV-CMV-*Gpx1* group (*n* = 9, cyan) showed no significant improvement. **i** Quantification of OHC loss based on cochleograms from the three groups, with a comparison of loss between the non-AAV and AAV-CMV-*Gpx4* groups. **j** Representative confocal cochlear images of Myo7a-stained basal and hook regions from the three groups. **k, l** Representative confocal images of Myo7a (cyan) and Ctbp2 (red) staining in the 32 kHz cochlear regions. **l** Quantification of Ctbp2-immunolabeled synaptic puncta in IHCs associated with the 32 kHz region. A circle represents one mouse. **m, n** Immunoblot analysis of 4HNE, GSDMD-FL, and GSDMD-N levels in non-AAV- and AAV-CMV-*Gpx4*-injected mice with/without noise exposure. Cochleae from noise-exposed mice were collected at 8 h-PNE. Detection was performed under short (5 s) and prolonged (30 s) exposure conditions to visualize GSDMD-FL and GSDMD-N bands, respectively. Data are presented as mean ± SEM (*n* = 3 biological replicates). Statistical analyses were conducted using two-way (**d**−**i**) and one-way (**b, l, n**) ANOVA with Bonferroni post hoc test. Scale bar = 10 μm. NE noise exposure, GC genome copies, P3 postnatal day 3.

expressed and functions in SCs (Fig. 1), GPX4 overexpression in HCs, but not in SCs, was sufficient to attenuate cochlear oxidative stress and GSDMD activation (Fig. 6e−g, Supplementary Fig. 10). This finding suggests the presence of intercellular communication between HCs and SCs, enabling the ROS-GSDMD feedback loop to mediate cochlear damage.

### EGFR/ERK maintains the noise-induced ROS-GSDMD loop running between HCs and SCs

Intercellular communication between HCs and SCs likely occurs via membrane receptors. Toll-like receptor 4 (TLR4) is a potential membrane receptor in SCs that may mediate GSDMD cleavage, due to its role in multi-pathogenesis conditions of the non-auditory system[11]. In the present study, *Tlr4*-deficient mice exhibited similar levels of both GSDMD-FL and GSDMD-N in the cochlea after noise exposure compared to WT controls (Fig. 7a). These findings suggest that TLR4 is not the principal mediator of noise-induced GSDMD activation.

EGFR has also been identified as playing a critical role in NIHL[31]. In the present study, the FDA-approved EGFR inhibitor Zorifertinib significantly reduced noise-induced 4HNE and GSDMD-N levels in mouse cochleae (Fig. 7b, c). Similar to GSDMD, EGFR expression is predominantly localized to SCs[31], and activation of EGFR is known to trigger downstream signaling via ERK phosphorylation (p-ERK)[31], a key mediator of cellular responses. Immunofluorescence analysis in the present study revealed a noise-induced elevation of p-ERK levels in cochlear SCs, but not in HCs (Fig. 7d). Pharmacological inhibition of p-ERK upstream signal with the FDA-approved drug dabrafenib[32] reduced noise-induced GSDMD-N and 4HNE levels (Fig. 7e−g). In addition, ASN007, a selective potent and orally ERK inhibitor[33], also markedly reduced noise-induced GSDMD-N and 4HNE levels (Supplementary Fig. 11). These results confirm the critical role of the EGFR/ERK pathway in the signaling from HC-derived ROS to GSDMD activation in SCs.

The connection between ROS accumulation in HCs and ERK phosphorylation in SCs was further established. Overexpression of *Gpx4* in HCs (via AAV-HC-*Gpx4* transfection) reduced p-ERK levels in SCs in response to noise exposure, as shown by both WB (Fig. 7h) and immunofluorescence analysis (Fig. 7i). Taken together, these findings suggest that oxidative stress in HCs initiates the cochlear ROS-GSDMD feedback loop via EGFR-ERK signaling across the SC membrane (Fig. 8). However, the precise mechanism and molecules linking ROS in HCs to EGFR/ERK activation in SCs require further investigation.

### Discussion

In the present study, we confirm that *Gsdmd* cKO in SCs largely reduced the threshold elevation and OHC loss caused by noise exposure (2 h, 100 dB SPL), while no protection effect was observed in *Gsdmd*-HC-cKO mice. Blocking caspase-11-dependent GSDMD activation in SCs nearly totally abolished the oxidative stress insult to HCs caused by noise. Conversely, mitigating oxidative stress specifically in HCs (via *Gpx4* overexpression) reduced ERK phosphorylation in SCs, likely through EGFR signaling across the SC membrane, leading to GSDMD activation. Based on these findings, we propose a conceptual framework in which a positive feedback loop between HCs and SCs: initial ROS accumulation in HCs triggers GSDMD activation in SCs, which in turn exacerbates oxidative damage in HCs. We further suggest that breaking this feedback loop could be an effective approach for gene therapy to prevent hearing dysfunction.

In the present study, HC-to-SC signaling in this loop is relatively clear, with EGFR acting as a signal transducer for their intercellular communication, although the exact ligand(s) of EGFR remain to be identified. Consistent with the proposed role of EGFR, previous reports revealed that EGFR expression is confined to SCs, rather than HCs, in both mouse and human cochleae[31,34]. Its cKO in the cochlea largely reduced noise-induced hearing dysfunction[31]. In the present study, EGFR is upstream of ERK phosphorylation and GSDMD cleavage following noise exposure. These downstream signaling events in SCs can be significantly suppressed by restricting ROS specifically in HCs, supporting the notion that EGFR in SCs serves as a receptor for oxidative stress recognition. Notably, EGFR inhibitor alleviated cochlear oxidative stress damage, further confirming EGFR/p-ERK as a pivotal node in the ROS-GSDMD positive feedback loop. Previous study also showed that p-ERK level is up-regulated in cochlear SCs upon mechanical and noise damage[35,36]. Our pharmacological inhibition of p-ERK activity blocked noise-induced GSDMD-dependent damage cascades, indicating p-ERK exacerbates cochlear injury. This contrasts with prior work showing *Erk2*-cKO in HCs increased susceptibility to NIHL, particularly enhancing IHC loss, which suggests ERK2 mediates protective signaling in HCs[37]. The protective role of ERK signaling has also been reported against aminoglycoside-induced hair cell injury[38,39]. This functional dichotomy demonstrates that EGFR-ERK signaling exhibits cell-type- or context-dependent heterogeneity.

SC-to-HC signaling has been reported in previous studies involving molecular or mechanical interactions[20,21,35,40,41]. In vitro studies have shown that mechanical damage to the cochlea induces transient ERK activation in SCs, which in turn triggers Ca²⁺ wave oscillation in HCs[35]. Additionally, in vivo studies revealed that hyperpolarization of specific SCs (such as outer pillar and Deiters' cells) directly leads to the suppression of cochlear amplification and IHC excitation[21]. These findings highlight the role of SCs in modulating HC molecular biology and electrophysiological characteristics, although the exact critical molecular mechanisms remain to be elucidated. In the present study, we make significant progress by demonstrating that GSDMD acts as a key SC molecule in modulating noise-induced HC damage. However, the precise mechanisms by which activated GSDMD signals back to enhance oxidative stress and promote HC death are not yet fully understood. Previous studies have reported that GSDMD cleavage and pore formation can facilitate potassium efflux[42,43], drive F-actin

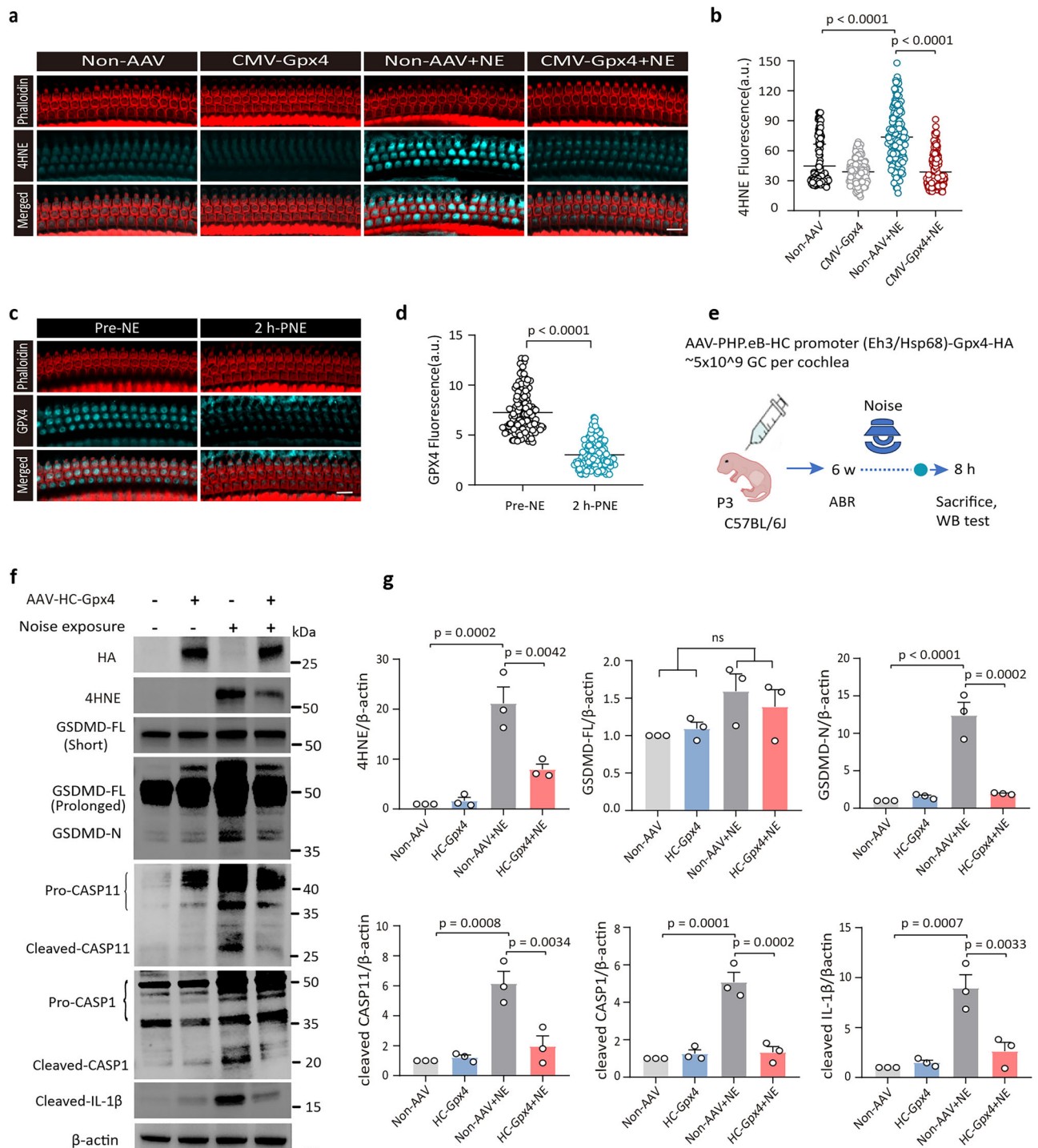

**Fig. 6 | GPX4 overexpression specific in HCs is sufficient to block noise-induced GSDMD-signaling cascades. a** Immunolabeling of 4HNE (cyan) in OHCs co-stained with phalloidin (red) in non-AAV- and AAV-CMV-*Gpx4*-injected mice with/without noise exposure. (scale bar = 10 μm). **b** Quantification of 4HNE immunolabeling in OHCs from the 22.6–32 kHz frequency region of the cochlea. 3 biological replicates. Each circle represents an individual OHC. **c** Immunolabeling of GPX4 (cyan) in OHCs co-stained with phalloidin (red). **d** Quantitative analysis of GPX4 immunolabeling in OHCs. 3 biological replicates. Each circle represents an individual OHC. **e** Schematic of the experimental design. Mice were injected with AAV.PHP.eB-*Gpx4*-HA driven by an HC promoter (*Rbm24*-Eh3-*Hsp68* promoter), followed by auditory electrophysiology and cochlear morphologic analysis. **f** WB analysis of indicated

proteins in cochlear tissues from non-AAV- and AAV-HC-*Gpx4*-injected mice with/without noise exposure. Cochleae were collected at 8 h-PNE. The first row shows HA-tagged GPX4. Detection parameters included short exposure (5 s) and prolonged exposure (30 s) to visualize GSDMD-FL and GSDMD-N bands, respectively. **g** GPX4 overexpression specifically in HCs reduced cochlear level of 4HNE, cleaved GSDMD, caspase-11, caspase-1, and IL-1β following noise exposure. All data are presented as mean ± SEM (*n* = 3 biological replicates). Statistical analysis was performed using one-way ANOVA with Bonferroni post hoc test (**b**, **g**) and unpaired Student's *t*-test (**d**). ns not significant, NE noise exposure, OHC outer hair cell, HC hair cell, GC genome copies, P3 postnatal day 3.

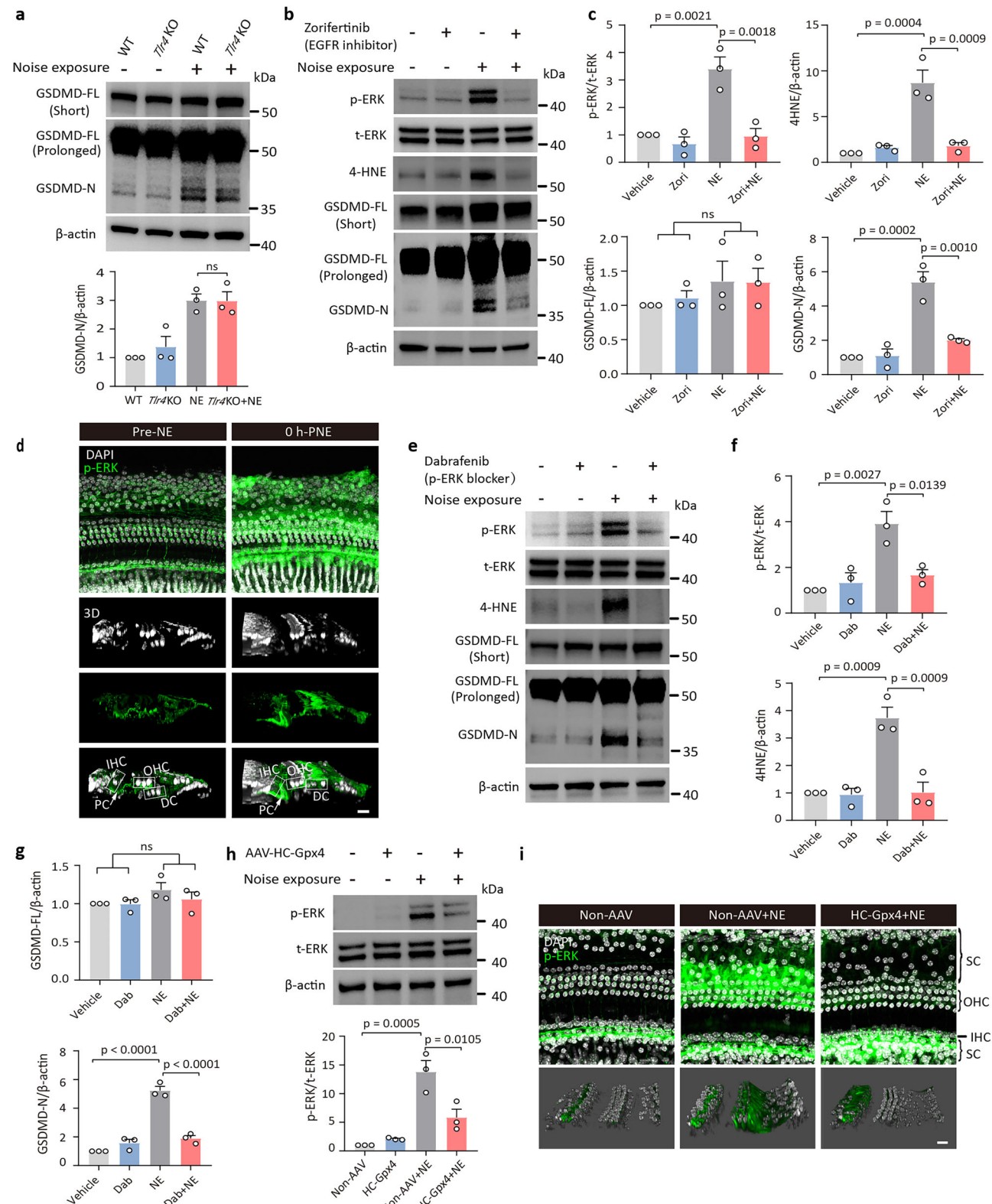

disassembly[16], or enable the release of pro-inflammatory mediators[7,10]. These biological processes may affect SC mechanical structure or signaling regulation[40,41,44], fulfilling the role of increasing HC's susceptibility to acoustic trauma.

A recent study revealed that extracellular vesicles can transplant GSDMD pores onto the plasma membrane of bystander cells, inducing their death[15]. In the cochlear context, exosomes carrying heat shock proteins have been shown to protect against aminoglycoside-induced

vestibular HCs' death via paracrine signaling[45]. These findings collectively suggest a plausible mechanism whereby GSDMD-N could be transferred indirectly—potentially via extracellular vesicles—from supporting cells to hair cell membrane and directly inducing pyroptosis. However, spatial mapping of GSDMD activation remains technically constrained. Our commercially available GSDMD-N antibodies showed nonspecific binding in cochlear tissue and failed validation using *Gsdmd*-KO cochlear sample (Supplementary Fig. 13),

**Fig. 7 | EGFR activation governs ROS-GSDMD loop between HCs and SCs.**
**a** Comparison of GSDMD changes between WT and *Tlr4* KO mice following noise exposure, evaluated at 8 h-PNE. Detection parameters included short exposure (5 s) and prolonged exposure (30 s) to visualize GSDMD-FL and GSDMD-N bands, respectively. **b, c** Effect of Zorifertinib (EGFR inhibitor) on GSDMD, 4HNE, p-ERK, and t-ERK levels in WT mice after NE, evaluated at 8 h-PNE. **d** Representative confocal and 3D images of p-ERK (green) in cochlear tissues before and immediately after noise exposure. **e**–**g** Effect of Dabrafenib (an inhibitor to reduce p-ERK levels) on GSDMD, 4HNE, p-ERK, and t-ERK levels in WT mice after NE, evaluated at 8 h-PNE. **h** Representative WB analysis of p-ERK and t-ERK in cochlea transfected with/without AAV-HC-*Gpx4*. The p-ERK/t-ERK ratios were normalized to β-actin as a loading control and presented as fold-changes relative to the vehicle group (set to 1). **i** Immunolabeling of p-ERK (green) in the 22.6–32 kHz cochlear region in non-AAV and AAV-HC-*Gpx4*-transfected mice with noise exposure, observed at 0 h-PNE. A significant reduction in p-ERK activation was observed in the OHC region of *Gpx4*-transfected mice. All data are presented as mean ± SEM (*n* = 3 biological replicates). Statistical analysis was performed using one-way ANOVA with Bonferroni post hoc test. ns, not significant. Scale bar = 10 μm. WT wild type, NE noise exposure, Zori Zorifertinib, Dab Dabrafenib, SCs supporting cells, OHC outer hair cells, IHC inner hair cells, PC pillar cells, DC Deiter cells, p-ERK phosphorylated ERK, t-ERK total ERK.

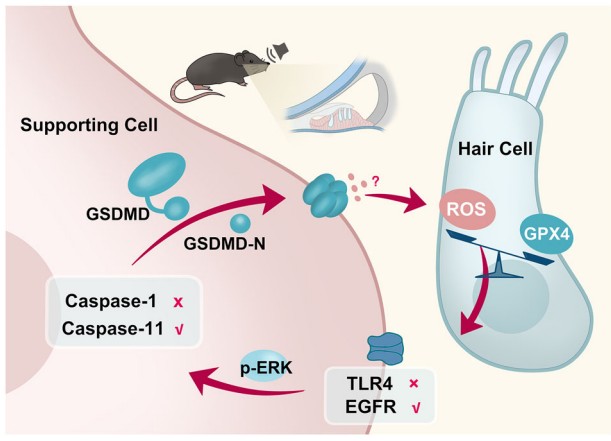

**Fig. 8 | Schematic model of GSDMD-mediated signaling between SCs and HCs after acoustic trauma.** Noise exposure induces ROS accumulation in HCs, activating caspase-11/GSDMD signaling in supporting cells via EGFR/ERK, which in turn amplifies oxidative stress in HCs, forming a damaging intercellular feedback loop.

precluding reliable immunolabeling of GSDMD-N fragments in HCs or SCs.

Using *Casp1* KO and *Casp11* KO mice, we confirmed that cochlear GSDMD cleavage induced by noise exposure is dependent on caspase-11 but not caspase-1. Additionally, *Gsdmd* deficiency in SCs almost completely abolished noise-induced caspase-1 activation and subsequent IL-1β cleavage, suggesting GSDMD's role in driving cochlear inflammation. These findings contrast with previous studies that proposed GSDMD as a downstream target of caspase-1 in multiple cochlear pathologies (ototoxic drugs[46,47], and lipopolysaccharides[48]), underscoring the need for experimental verification of classical signaling pathways. Furthermore, we confirmed that the caspase-1/IL-1β signaling axis is not a critical node in the ROS-GSDMD positive feedback loop, suggesting it plays a secondary role in noise-induced cochlear pathology. Consistently, previous studies have shown that IL-1β-deficient mice lack protection against NIHL[49,50]. The key inflammatory mediators by which GSDMD activation in SCs contributes to HC damage require further investigation.

The recognition of the damage signal positive feedback loop deepens our understanding of mechanisms and therapeutic strategies for NIHL. Specifically, our results show that global *Gsdmd* KO mice exhibit less reduction in noise-induced HC loss as compared to *Gsdmd*-SC-cKO mice, indicating that GSDMD in non-SC cells may function in acoustic trauma, opposing to its role in SCs. Scattered evidence exists for the protective role of GSDMD in the NIHL model. Macrophages are activated in the cochlea by noise, where they preserve a protective function in noise-induced cochlear injury[51]. In other tissues, GSDMD is also activated in macrophages, which, in turn, can enhance muscle stem cell mobilization for repair[14]. A plausible explanation for the weaker effect of *Gsdmd* KO is that the global deletion of *Gsdmd* eliminates the beneficial effects of this gene. These findings underscore the

need for cell-specific targeting to achieve precise therapeutics in cochlear pathology. Further clarification of the reparative role of GSDMD in cochlear-specific cells is clinically important to avoid the off-target effects of systemic drugs. Although GSDMD is highly expressed in several subtypes of SC, we cannot currently determine if GSDMD activation mediating cochlear damage originates from a specific SC subtype (e.g., OPCs). This question deserves systematic elucidation in future studies. Furthermore, it remains possible that only OHCs, not IHCs, can establish a damaging ROS-GSDMD positive feedback loop with SCs, which may lead to the failure of *Gsdmd* KO in preventing IHC's ribbon synapse from noise-induced damage.

Taken together, we have identified that GSDMD activation in SCs, rather than in HCs, mediates noise-induced cochlear damage. While whole cochlear lysates were used for WB to elucidate the mechanism due to technical constraints, we mitigated interpretative limitations through complementary approaches: (1) Cell type-specific manipulations functionally attributed molecular changes to targeted cells, such as *Gsdmd*-cKO in SCs and GPX4 overexpression in HCs; (2) Spatial immunostaining validated key SC-HC intercellular signaling events, including 4-HNE accumulation in HCs of *Gsdmd*-SC-cKO mice, and p-ERK level in SCs following HC-targeted GPX4 overexpression. Oxidative stress in HCs and GSDMD activation in SCs form a positive feedback loop that synergistically drives the progression of noise-induced cochlear injury. These findings establish a conceptual framework for understanding the mechanisms of NIHL and provide a basis for developing therapeutic strategies targeting the ROS-GSDMD feedback loop.

## Methods
### Mice
All animal procedures were approved by the Ethics Committee of Shanghai Sixth People's Hospital, affiliated to Shanghai Jiao Tong University School of Medicine (DWLL2023-0261). All efforts were made to minimize pain.

C57BL/6J mice obtained from SBF Biotechnology Laboratory Animal Field (Beijing, China); *Gsdmd* KO (T010437), *Gsdmd* ^flox/flox^ (T059954), and B6/JGpt-H11 ^em1Cin (CAG-LoxP-ZsGreen-Stop-LoxP-tdTomato)^/Gpt reporter mice (T006163) obtained from GemPharmatech (Nanjing, China); *Tlr4* KO (NM-KO-18052) mice obtained from Shanghai Model Organisms Center; *Casp1* KO (12362) mice obtained from Cyagen Biosciences (Suzhou, China). *Casp1/11* DKO mice were kindly provided by Prof. Haibing Zhang (Shanghai Institute of Nutrition and Health, Chinese Academy of Sciences, Shanghai, China). *Casp11* KO mice were generously provided by Prof. Feng Shao (National Institute of Biological Sciences, Beijing). All mouse strains were backcrossed to the C57BL/6 for at least 10 generations. For most auditory experiments, male and female mice were included in approximately equal numbers to enable sex-comparative analysis (Supplementary Fig. 12). Mice were housed in a temperature-controlled facility (24 °C) under a 12-h light/dark cycle with free access to standard laboratory chow and sterilized water. Mice were routinely genotyped using genomic DNA extracted from tail biopsies via the Direct PCR Lysis tail reagent (Viagen), in

accordance with the manufacturer's instructions (see Supplementary Table 1 for the primers used in this study).

## Viral vectors

AAV.PHP.eB vectors were prepared at PackGene Biotechnology (Guangzhou, China) respectively with the transgene cassettes: CMV-driven mCherry (titer: $1 \times 10^{13}$ GC/mL), *Gpx1*-MYC ($1 \times 10^{13}$ GC/mL), *Gpx4*-HA ($1.28 \times 10^{13}$ GC/mL), and HC-promoter (*Rbm24*-Eh3 enhancer following with *Hsp68* mini-promoter sequences)-driven *Cre* ($1 \times 10^{13}$ GC/mL). AAV.ie vectors were prepared at OBiO Technology Corp (Shanghai, China) with the SC-promoter-driven *Cre* or *Gpx4* transgene cassette ($1.43 \times 10^{13}$ GC/mL, $1.8 \times 10^{13}$ GC/mL). The SC-specific promoter used in AAV-ie constructs was approved by Prof. Guisheng Zhong's laboratory (ShanghaiTech University) and commercially provided by OBiO Technology (China, Shanghai). These vectors were stored at −80 °C until thawed immediately before injection.

## Microinjection into the cochlea of neonatal mice

P2-3 mice were anesthetized by hypothermia and placed on an ice pad for subsequent surgical procedures. As previously described[23], cochleostomy was performed via a preauricular incision to expose the cochlear bulla. AAV was delivered into the scala media at 120 nL/min using glass micropipettes controlled by Micro4 MicroSyringe Pump Controller (WPI, Sarasota, FL, USA). Surgery was performed only on the right ear of each animal and was completed within 5–10 min. The total amount of AAV injected into each cochlea was limited to $5 \times 10^9$ GC. After the injection, the skin incision was closed with tissue adhesive (3 M Vetbond, #1469SB). The pups were then placed on a 37 °C warming pad for 30 min before being returned to their mother for nursing.

## Drug administration

Disulfiram (86720, Sigma-Aldrich) and NAC (MedChemExpress) were administered intraperitoneally (i.p.). Disulfiram was dissolved in 2% dimethyl sulfoxide (DMSO, D2650, Sigma-Aldrich) and 98% corn oil (C8267, Sigma-Aldrich) and delivered at 50 mg/kg/day for seven days (starting two days before noise exposure, and continued for five days post-exposure). NAC was dissolved in saline and administered intraperitoneally at 400 mg/kg/day for three consecutive days, with an extra dose at 2 h-PNE.

Zorifertinib (HY-18750), Dabrafenib mesylate (HY-14660) and ASN007 (HY-136579A) were purchased from MedChemExpress. Zorifertinib and Dabrafenib mesylate were dissolved in 2% DMSO and 98% corn oil and were administered at 20 and 60 mg/kg/day, respectively, for three days and an additional dose at 2 h-PNE via oral gavage. ASN007 was dissolved in saline and administered via oral gavage at 50 mg/kg/day for seven days, with an extra dose at 2 h-PNE.

## Noise exposure

Noise signal was generated using the RZ6 system (TDT) software and subsequently conducted with the amplifiers (Yamaha, Japan) and speakers (Pyramid, USA). Calibration of the noise level was performed with the free-field microphone (PCB). Mice (6-week-old) were awake and unrestrained in cages. They were exposed to octave band noise (8–16 kHz) at 100 dB SPL for 2 h.

## Electrophysiological evaluation

ABR and DPOAE were measured in mice using the RZ6 BioAMP Processor (TDT, USA) according to established protocols. Mice were anesthetized via intraperitoneal injection of sodium pentobarbital (100 mg/kg, 1%) and placed on a homeothermic heating pad. In ABR testing, subdermal needle electrodes were inserted at the vertex of the skull (active), the pinna of the tested ear (reference), and the hind limb (ground). The stimulus generation, ABR wave acquisition, equipment control, and data management were similar with a previous study[52].

Briefly, acoustic stimuli were presented via custom-designed plastic tubing connected to an MF1 Multi-Field Magnetic Speaker (TDT, USA) positioned in the ear canal. Pure-tone stimuli of 10 ms duration (0.5-ms rise/fall) were presented at 21.1/s across intended frequencies. At each frequency, the test started from 90 dB SPL to ensure a clear response, and stepped down in 10 dB steps at high levels and then 5 dB when approaching the threshold. Evoked responses were amplified 10,000× (PA4 bio-amplifier, TDT) and averaged 500 times with a 0.3 to 3 kHz band-pass filter. Thresholds were defined as the lowest stimulus level that produced two or more discernible ABR waveforms (waves 1 to 5). Threshold shifts were determined as the subtraction of the pre-noise exposure (pre-NE) thresholds from post-noise exposure thresholds.

DPOAE thresholds were measured using two primary tones (f2/f1 ratio = 1.2) with levels L1 = L2 + 10 dB. The distortion product at 2f1 - f2 was recorded using a custom-designed probe equipped with a low-noise ER10B+ microphone (TDT Systems, USA). The f2 stimuli were presented, ranging from 5.6 to 32 kHz in half-octave intervals from 70 to 10 dB SPL in 5 dB steps. The potential intrinsic distortion and the noise floor were examined by recording the sound in the coupler to which the primary tones were presented at the maximal level (80 dB SPL) for DPOAE test. The noise floor across the whole frequency range and around the targeted frequency for DPOAE was more than 80 dB below the level of the primary tones. DPOAE threshold was defined as the interpolated value of f2 intensity required to generate a 0 dB SPL DPOAE.

## Sample preparation and analysis of immunofluorescence

Cochlear basement membranes and cryosections from adult mice were processed according to our previous study[13]. After the final auditory measurements, cochleae were collected and immediately fixed in 4% paraformaldehyde at 24 °C for 2 h. After fixation, the cochleae were decalcified in 120 mM EDTA for 1–2 days at 4 °C. Cochleae were micro-dissected into apical, middle, basal, and hooked segments. Dissected pieces were permeabilized and preincubated in a solution containing 5% normal goat serum and 0.5% Triton X-100 in phosphate-buffered saline (PBS) at 24 °C for 1 h. Cochlear pieces were incubated overnight at 37 °C with primary antibodies diluted in 5% normal goat serum. The details of the antibodies are provided in Supplementary Table 2. Secondary antibodies conjugated to Alexa Fluor 488, 555, or 633 (Invitrogen, 1:500) were applied for 2 h at room temperature. Phalloidin-Alexa Fluor 488/647 (1:5000) and DAPI (Abcam, ab104139) were used for stereocilia and nuclear labeling, respectively.

Confocal laser scanning microscopy (LSM 710 META; Zeiss, Shanghai, China) was employed to capture images. 3D reconstructions of p-ERK expression were performed using a Leica Stellaris 5 confocal microscope. Fluorescence intensity of 4HNE and GPX4 in OHCs was quantified using ImageJ software (National Institutes of Health, Bethesda, MD, USA). Semi-quantitative analysis of immunofluorescence signals for target proteins in OHCs was specifically performed in the 22.6–32 kHz frequency region of the cochlea, where high cellular resolution was ensured. Data normalization and statistical analyses were performed to account for variations in staining and processing.

## Immunohistochemistry for cochlear

Following fixation and decalcification, cochlear tissues were dehydrated through a graded ethanol series (75% 4 h, 85% 2 h, 90% 2 h, 95% 1 h, 100% ethanol I & II 30 min each), cleared in alcohol-benzene (5–10 min) and xylene (I & II, 5–10 min each), then infiltrated with molten paraffin at 65 °C (three changes, 1 h each) and embedded. Tissue sections (4 μm) were dewaxed using an eco-friendly solution series (3 x 10min), rehydrated through a descending ethanol (3 × 5min) to distilled water, and subjected to antigen retrieval (method unspecified). After natural cooling and PBS washing, endogenous peroxidase activity was quenched using 3% $H_2O_2$ (25 min, room temperature, in

the dark), followed by PBS washes. Sections were blocked with 3% BSA (30 min, room temperature), then incubated with a primary antibody against GSDMD (ab209845) overnight at 4 °C. After PBS washing, sections were incubated with an HRP-conjugated, species-matched secondary antibody (50 min, room temperature), followed by further PBS washes. DAB chromogen was applied, and the color reaction was monitored microscopically for a brown-yellow signal, then stopped with tap water. Counterstaining was performed using hematoxylin (3 min), followed by water rinsing, differentiation, bluing, and a final rinse. Sections were dehydrated through a graded ethanol series (75%, 85%, 100% I & II, 5 min each), cleared in n-butanol and xylene (5 min each), briefly air-dried, and mounted. Finally, the slides were imaged using a Panoramic scanner and corresponding analysis software.

### Quantification of synaptic ribbons and OHCs
DAPI-stained images of cochlear pieces, spanning from the apex to the hook, were captured at ×0.7 digital zoom with a ×10 magnification objective on a Zeiss microscope. Stitched composite images were analyzed to identify the frequency map of cochlear basement membrane via ImageJ software loading relative plugin from Mass Eye and Ear Aton-Peabody Laboratories' histology cores. Ctbp2-labeled puncta were counted as an individual synaptic ribbon. The total number of puncta from 7–10 consecutive IHCs within the 32 kHz frequency region was calculated to show the synaptic density (synapse per IHC). Myo7a-labeled OHCs were counted across the whole frequency region. OHC loss was plotted as percentages per 0.25 mm epithelial segment to generate cochleograms.

### WB analysis
Cochleae were collected, rinsed in cold PBS, and homogenized in lysis buffer (Invitrogen, SD-001/SN-002) containing protease/phosphatase inhibitors (1:100, Epizyme Biomedical Technology Co., L). Each biological sample consisted of four cochleae from two mice and was repeated three times independently for each experimental group. The homogenized tissue was then subjected to centrifugation at 12,000 × $g$ for 30 min at 4 °C to collect the supernatants. Protein concentrations were quantified using the Omni-Easy™ Ready-to-use BCA Protein Assay Kit (ZJ102) according to the product instructions.

To prepare the samples for subsequent electrophoresis, an equal volume of 5× loading buffer (one-fourth the volume of the supernatant) was added. The mixture was vortexed to ensure thorough mixing and then heated at 98 °C for 10 min to denature the proteins. For WB analysis, 30 μg of protein lysate was loaded onto SDS-PAGE gels (MeilunGel Precast PAGE Gel, Bis-tris, 1.0 mm, 10% or 12%). After electrophoresis, the proteins were transferred to nitrocellulose membranes (NC), which were subsequently blocked with 5% nonfat milk for 1 h at room temperature to minimize nonspecific binding. The membranes were then incubated overnight at 4 °C with primary antibodies, the details of which are provided in Supplementary Table 2. After primary antibody incubation, the membranes were washed and incubated for 1 h at room temperature with the appropriate secondary antibodies: anti-rabbit (ABclonal, AS014), anti-mouse (ABclonal, AS003), or anti-rat (ABclonal, AS028), all diluted 1:5000.

For the analysis of phosphorylated proteins, phospho-specific antibodies were applied, followed by a stripping procedure to remove phosphorylated antibodies. The membranes were then incubated with antibodies targeting the total proteins. Chemiluminescent detection was performed using the Omni-ECL™ Femto Light Chemiluminescence Kit. Protein bands were visualized using a Tanon 5200 Imaging System (Tanon, Shanghai, China). Band intensities were quantified using ImageJ software and normalized to β-actin as a loading control.

### Statistical analysis
Results are presented as the mean or means ± SEM. Statistical analyses and data visualization were performed using GraphPad Prism 8 (GraphPad Software, San Diego, CA, USA). Experiments with more than two groups were tested using ANOVA, followed by Bonferroni post hoc tests. Experiments with two groups were analyzed using an unpaired Student's $t$-test. Statistical significance was set at $p < 0.05$.

### Reporting summary
Further information on research design is available in the Nature Portfolio Reporting Summary linked to this article.

## Data availability
All data underlying the findings presented in this manuscript are included in the article, provided in the Supplementary Information, or available from the corresponding author upon request. Correspondence and material requests should be addressed to Z.Z. (email: zhangzhen1994@sjtu.edu.cn). Source data are provided with this paper.

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

## Acknowledgements

This research was supported by the Key Program of the National Natural Science Foundation of China (No. 82330034 to S.Y.), the National Natural Science Foundation of China (No. 82401354 to Z.Z., 82171139 and 82371152 to Y.F., 82205201 to Y.S. and 82301311 to W.L.). We thank Prof. Haibing Zhang (Shanghai Institute of Nutrition and Health, Chinese Academy of Sciences), Prof. Tao Xu (Shanghai Sixth People's Hospital Affiliated to Shanghai Jiao Tong University School of Medicine), and Prof. Feng Shao (National Institute of Biological Sciences, Beijing) for their support in conducting the mouse studies. We thank Pro. Guisheng Zhong and his team at ShanghaiTech University for providing AAV.ie and SC-specific promoter system.

## Author contributions

J.W., Y.F., Z.Z. conceptualized and designed the study. J.W., Z.Z. and S.Y. revised the manuscript. Data analysis and manuscript drafting were carried out by Z.Z. and L.X. L.X. and J.L. completed most of the electrophysiological evaluations, immunolabeling, drug administration, and WB analysis. Y.C. and T.Z. performed part of the ABR recordings and immunolabeling, while Y.J. participated in sample preparation and created the schematic diagram. L.X. and X.C. were responsible for part of the DPOAE recordings and WB. W.L. and Y.S. performed all quantitative analyses. H.W. made substantial contributions to experimental design, data interpretation during the review process. All authors contributed to the discussion of the results and the refinement of the final manuscript.

## Competing interests

The authors declare no competing interests.
