## [Transparent Peer Review file · Nature Communications]

Supporting cells orchestrate noise-induced hearing loss via Gasdermin D-dependent signaling loop with hair cells

Corresponding Author: Dr Zhen Zhang

Version 1:

Reviewer comments:

Reviewer #1

(Remarks to the Author)

The paper: "ROS-GSDMD feedback loop between hair cells and supporting cells governs noise-induced hearing loss" reports some potentially important findings. Namely, they "reveal a paradigm-shifting mechanism where supporting cells (SCs) orchestrate NIHL through Gasdermin D (GSDMD) with a non-canonical pathway. Conditional knockout of GSDMD in SCs, but not in HCs, significantly reduces NIHL." My main expertise is in cochlear physiology, and I have focussed largely on the physiological measurements reported in the paper. The physiological findings are important because they underpin the major findings reported in the paper.

To assess the hearing of the mice, the authors used two non-invasive techniques. They measured the ABR and DPOAEs. ABR measurements provide an overall measure of the hearing sensitivity of the mice in their experiments. This measure does not, on its own, enable the experimenters to discover the precise origin of any alteration in the hearing of the mice from control. DPOAE measurements reflect the performance of the nonlinear amplification of the outer hair cells. As a first point, and to enable an assessment of the quality of the data, will the authors please state in the methods the intrinsic distortion levels of the measurement system and hence its threshold for measuring acoustic distortion.

Figures 1e, h. What is "dB" relative to? Is it a value common to all measurements such as SPL? Or is it relative to each measurement? Ideally it should be referred to a commonly accepted standard, like SPL. It is easier to understand. Could you please, therefor, clarify what is meant by "dB" and ideally, for increased clarity and understanding, convert to SPL.

In your methods you state:

"Briefly, acoustic stimuli were presented via custom-designed plastic tubing connected to an MF1 Multi-Field Magnetic Speaker (TDT, USA) positioned in the ear canal. Pure-tone stimuli of 10- ms (0.5- ms rise/fall) were presented at 21.1/s across intended frequencies, starting from 90 dB SPL and decreasing in steps of 5 dB. Evoked responses were amplified 10,000× (PA4 bio-amplifier, TDT) and averaged 500 times with 0.3 to 3 kHz band-pass 565 filter."

Thus, at the very beginning of your measurements you presented 500 intense tone bursts. The higher frequency regions of the mouse cochlea (above about 30 kHz) are very noise sensitive, and your regime could prejudice the sensitivity of the cochlea. In the methods, or in the results section, please explain why you chose to deliver the tones descending from a high to low levels instead of ascending from low to high levels, or perhaps with a degree of randomization in the presentation of both frequencies and sound levels, which would tend to preserve hearing sensitivity.

Figure 1i. In the methods you state:

"In DPOAEs testing, the primary tones f_1 and f_2 were delivered through two MF1 loudspeakers, respectively, positioned in a closed-field arrangement. A custom-designed probe equipped with a low-noise ER10B+ microphone (TDT, USA) facilitated precise stimulus presentation and recording. DPOAEs were measured with a fixed f_2/f_1 ratio of 1.2, where the intensity of f_2 (L_2) was set to 10 dB lower than f_1 (L_1). Frequencies ranged from 5.6 to 32 kHz in half-octave intervals. For each f_2 , L_2 was varied from 70 to 10 dB SPL in 5 dB steps. The $2f_1 - f_2$ DPOAE amplitude and the surrounding noise floor were extracted for offline analysis."

In the caption to Fig1i:

"i. DPOAE amplitudes were higher in Gsdmd-SC-cKO mice compared to Gsdmdflox/flox mice at 14 d-PNE (solid line), with no significant differences at Pre-noise exposure (dashed line)."

Again it seems you began your measurements of DPOAE at high levels and descended the SPL to low levels, which may have prejudiced the sensitivity of the cochlea. It may have been helpful to design a protocol that extracted the DPOAEs continuously during the measurement. Such a protocol would enable you to monitor the DPOAEs directly and to store them

and the raw data. This would have provided you with an immediate visual display of the sensitivity of the cochlea. Can you please explain why you chose your approach and not ones that enabled you to monitor DPOAE directly and to start from low SPLs to high, to preserve cochlear sensitivity. Also, please state the SPL at which you measured the DPOAEs. This does not appear to be given in the caption of Figure 1. Additionally, please define dBV and why DPOAE magnitude is not given as SPL. I strongly suggest that the presentation of DPOAE ISO response curves would have given a clearer understanding of the frequency dependence of DPOAE generation in the cochlea under different conditions. Amplitude can give a confusing impression because, especially at high levels, e.g F1 at 50 dB SPL, DPOAE increase with level is very nonlinear and can reach a plateau, depending also on stimulus frequency.

Figure 1k. There appears to be no OHC loss for frequencies below about the 3.5 mm location, from apex. According to a well known mouse cochlear frequency map (Müller et al., *Hearing Research* 202 (2005) 63–73), this is close to the 34 kHz frequency place. Do these findings, in part, account for why there appears to be no complete recovery from NIHL?

Figure 1m. Please state that this figure is based on sampling in the 32 kHz region where OHC loss is expected to be minimal, or indeed absent.

The comments above also pertain to figure 4d – i.

A remarkable finding, not commented on in the paper, is that there is no hair cell loss at the location of the noise exposure or at a location a half octave above this where noise is most effective at desensitizing the cochlea (Cody and Johnstone, 1981 *J. Acoust. Soc. Am.* 70, 707-711; Cody and Russell, 1985, *Nature* 315, 662-665). Instead, hair cell loss is confined to frequencies above about 34 kHz, a region where the cochlea is most sensitive noise insult. Thus presentation of an 8-16 kHz noise band causes no measurable change in sensitivity in the cochlea region apical to this, it causes loss of sensitivity to all frequency regions at and basal to the 8-16 kHz band, and causes 0% - 20% OHC loss at frequencies a half octave above the 8-16 kHz band and up to 100% OHC loss at frequencies above this (in the 40 kHz – 90 kHz region). Could it be that there are differences in the action of Gasdermin D on NIHL at the bases and hook region of the cochlea from its action in the more apical regions? Perhaps this point could be raised in the discussion.

Reviewer #2

(Remarks to the Author)

This manuscript by Xiao et al has elucidated mechanisms by which non-sensory cochlear supporting cells (SCs) interact with sensory hair cells (HCs) through Gasdermin D (GSDMD) to influence noise-induced loss of hearing and HCs. Using SC specific GSDMD knockout along with neurophysiology and molecular biology approaches, authors report that noise-induced oxidative stress in HC causes activation of GSDMD in SC through EGFR-p-ERK-Caspase-11/1 pathway in SC. Such GSDMD activation in SCs, in turn augments oxidative damage in HCs leading to their death and thus, forming a positive loop between SCs and HCs. Knocking out GSDMD from SC completely abolishes oxidative damage in HCs. Moreover, blocking GSDMD in SCs but not in HCs or overexpressing antioxidant enzyme GPX4 in HCs significantly reduces noise-induced hearing loss and preserves outer hair cells. Overall, this is a rigorously designed and well written manuscript highlighting another critical role for SCs in regulating hair cell degeneration in the context of noise-induced hearing loss. This study also reveals molecular mechanisms of coupling of oxidative stress with inflammation in auditory pathophysiology.

Following are some concerns that need author's attention.

1. The major weakness in the manuscript is the evidence for how activated GSDMD-N in SCs increases oxidative stress in HCs, leading to their degeneration. It will strengthen the manuscript if authors could show that GSDMD-N localizes to mitochondria in HCs and mediate mitochondrial damage and increase inflammatory cytokines and ROS production.
2. Authors should show the expression of GSDMD after noise exposure by immunohistochemistry to verify its expression or upregulation in HCs after injury.
3. It is interesting that GSDMD KO or GPX4 overexpression does not seem to affect the ABR thresholds and DPOAE levels at 1 day post noise exposure but rather at 14 days. How do authors interpret these data. This should be discussed in the manuscript. Also, it will be important to discuss why the protection in GSDMD KO is selective to outer hair cells since GSDMD seems to be expressed by inner and outer pillar cells besides Claudius cells.
4. GSDMD knockdown in SCs (Pillar and Claudius cell) via AAV should be verified with immunohistochemistry besides Western blot.
5. It will strengthen the manuscript if authors overexpress GSDMD-N protein in SCs and HCs to see if it reverses the protective effect of absence of GSDMD against noise-induced hearing loss.
6. Figure 1, panel c, please check the immunolabel for hair cells; is it Spectrin or Myo7a.
7. It is unclear if the western blots shown in the manuscript are performed on protein lysates prepared from whole cochlea or isolated organ of Corti. If it is the former then that confounds the data due to the presence of other cochlear compartments like spiral ligament and stria vascularis that also expresses the factors in the pathway which are studied in the manuscript.
8. Figure 3, it is unclear if GSDMD global or SCs conditional knockout was used to examine oxidative stress in HCs after acoustic trauma.
9. How do authors interpret the data on preservation of IHC synapses with GPX4 overexpression but not with GSDMD knockout.
10. Authors have indicated the use of male and female mice across the experiments; however the none of data hasn't been separated based on sex to determine if there is any sexual dimorphisms with the claimed phenomenon.
11. Please check the references, since some of references related to ERK inhibitors in noise-induced hearing loss have not been cited.

Reviewer #3

(Remarks to the Author)

This study by Xiao et al. presents an intriguing set of data on the intercellular signaling pathway between cochlear hair cells (HCs) and supporting cells (SCs) in response to noise exposure. Traditionally, HC death and synapse loss due to noise exposure have been discussed without considering SC interactions. However, SCs have recently been recognized for their diverse immune functions, which vary depending on the cochlear environment. This study contributes to this evolving understanding by demonstrating aspects of SC immune function. The authors effectively utilized two interesting AAV vectors, each selectively targeting either HCs or SCs. However, additional experiments using these vectors could be considered, as outlined below. Furthermore, spatial information is crucial when discussing signaling pathways between neighboring cells. To enhance this aspect, immunostaining should be presented alongside Western blotting.

Below are some specific notes that may help improve the manuscript.

Page 7, the authors used a cochlear SC-specific promoter for AAV-*ie* to efficiently transfer Cre to SCs. However, they do not specify which SC-specific gene's promoter was chosen for this experiment, either here or in the Methods section. Providing this detail would improve clarity and reproducibility.

Page 7, Western blotting is used to confirm the effectiveness of Gsdmd-SC-cKO in Figure S2. Since the Gsdmd expression in SCs of the WT cochlea is clearly shown by immunostaining in Figure 1, its decrease in SCs of the cKO cochlea could be exhibited also by immunostaining.

Page 8, there was no difference in ABR thresholds and OHC loss after noise exposure between Gsdmd-HC-cKO mice and Gsdmd flox mice. However, the authors suggested in Page 7 that GSDMD in non-SC cells may play a protective role in acoustic trauma, rather than the harmful role in SCs. If so, Gsdmd-HC-cKO should show elevated ABR thresholds and severer OHC loss as compared with Gsdmd flox mice.

Page 10, the elevation of GSDMD-N following noise exposure is demonstrated by Western blotting. However, other cochlear cell types besides SCs may also express GSDMD in response to noise exposure. To exclude this possibility, it would be important to confirm that GSDMD-N elevation occurs specifically in SCs using immunostaining.

Page 11, Caspase-1/11 double KO mice are used to conclude that Caspase-11 is the primary mediator of GSDMD cleavage during noise exposure. However, to rule out potential interactions between Caspase-1 and Caspase-11, conducting experiments with Caspase-11 single KO mice would be important.

Page 13-14, global Gsdmd KO mice are used to conclude that activated GSDMD enhances noise-induced oxidative stress. However, to fully support the authors' hypothesis throughout the manuscript, experiments using Gsdmd-SC-cKO mice instead of the global ones would be crucial for clarifying the intercellular signaling pathway between SCs and HCs in response to sound exposure. Also, 4HNE expression in HCs is exhibited by immunostaining, but showing the SC layer in addition to the HC layer in this immunostaining would be important.

Page 16, the authors discuss the significance of GPX4 in HCs based on Western blotting of the whole cochlea. Immunostaining for GPX4 to demonstrate its specific expression in HCs would be of great interest also here.

Page 17, GPX4 overexpression is shown to reduce noise-induced GSDMD activation and oxidative stress. To confirm that GPX1 does not contribute to this effect, additional experiments with GPX1 overexpression should be included.

Page 18, "sarifice" is a typo in Figure 4.

Page 20, the authors demonstrate the critical role of GPX4 in HCs in downregulating ROS, using GPX4 overexpression system in HCs. However, in order to exclude the possibility that GPX4 may also function in SCs, additional experiments with GPX4 overexpression in SCs would be important.

Page 27, the authors conclude that phosphorylation of ERK in SCs occurs prior to HC damage in response to noise exposure. They also cite a study (Ref. 33) showing that ERK activation in SCs leads to HC damage. However, ERK is also well known for its role in cell survival and proliferation, and some studies report that ERK and Akt activation in SCs can promote HC protection. These findings highlight the multifunctional nature of ERK in cochlear SCs. It would be valuable for the authors to discuss this broader context.

Version 2:

Reviewer comments:

Reviewer #1

(Remarks to the Author)

Many thanks for very carefully and fully dealing with my questions about your paper. I have made some comments below on

some of your responses that you may wish to consider, and most of them do not require any changes to the changes you have made to your paper in response to my original comments. Below I have reproduced by original comments and responses and have underlined those responses where I have made further comments. You may wish to consider my further questions and comments in sections 2 and 3 They are intended for clarity and understanding of your data. Other comments may influence future experimental design.

1. To assess the hearing of the mice, the authors used two non-invasive techniques. They measured the ABR and DPOAEs. ABR measurements provide an overall measure of the hearing sensitivity of the mice in their experiments. This measure does not, on its own, enable the experimenters to discover the precise origin of any alteration in the hearing of the mice from control.

We acknowledge your valid point regarding ABR's limitations in localizing the precise origin of hearing alterations. However, ABR remains the best option for non-invasive, frequency-selective assessment of hearing sensitivity, which is the major concern to the present study. It is well recognized that NIHL impacts mainly on the function of outer hair cells (OHCs) which are responsible for the hearing sensitivity. Therefore, the location of the hearing loss is already clear and is not the focus of the present study. Furthermore, ABR is the ONLY method that can be tested repeatedly (as needed in the present study) in small rodents.

This is patently not true. The standard, and much more direct, non-invasive procedure for testing OHC activity is the measurement of DPOAEs. This technique has been used in mammals including baby microchiroptera weighing less than 5gm to new borne human babies without disturbing their sleep... DPOAEs are measured routinely and regularly to test the hearing of mice at all stages of development.

While ABR cannot differentiate sensory versus neural contributions, we mitigated this limitation through complementary approaches: (1) DPOAEs specifically evaluated OHC function. (2) Cytocochleograms quantified sensory cell loss. (3) Ribbon synaptic counts assessed integrity of IHC-to-SGNs transmission. This multimodal strategy ensures ABR threshold shifts primarily reflect OHC-mediated sensitivity loss - the core focus of our NIHL investigation.

2. DPOAE measurements reflect the performance of the nonlinear amplification of the outer hair cells. As a first point, and to enable an assessment of the quality of the data, will the authors please state in the methods the intrinsic distortion levels of the measurement system and hence its threshold for measuring acoustic distortion.

Thanks for the suggestion. We revised the method section by adding the examination of the intrinsic distortion as shown below: (Line 613-618 in the revised manuscript)

The potential intrinsic distortion and the noise floor were examined by recording the sound via the coupling cavity with the stimulation used for DPOAE. The noise floor of the testing system was below 0 dB SPL and there was no significant intrinsic distortion when tested via the physical coupling cavity utilized for the DPOAE. DPOAE threshold was defined as the interpolated value of f2 intensity required to generate a 0 dB SPL DPOAE.

Many thanks for this ,but a more intuitive and direct way of presenting acoustic distortion in the sound system is to say that it is so many dB below the level of the primary tones. My guess is that for your system, it is about 80 dB. This means, for example, that there could be 10 dB SPL of distortion in tones presented at 90 dB SPL. Can you please express distortion levels in this format.

3. Figures 1e, h. What is "dB" relative to? Is it a value common to all measurements such as SPL? Or is it relative to each measurement? Ideally it should be referred to a commonly accepted standard, like SPL. It is easier to understand. Could you please, therefor, clarify what is meant by " dB " and ideally, for increased clarity and understanding, convert to SPL.

Figure 1e and h concern level shift, which is relative to baseline values, they should not be relative to a standard like SPL but relative to each other, since a shift would be the same in dB no matter whether SPL or other metric is used.

What baseline value were you using? Was it the threshold stimulus tone SPL at each particular frequency? For example, did you have to use tones close to 100 dB SPL at 32 kHz to elicit a signal from Gsdmd KO at 1 day and ~ 70 dB SPL at day 14? the absolute SPL you use is important because it shows if the stimuli fall within the linear dynamic range of your sound system.

Please note that dB SPL is used in d and g where the absolute thresholds are addressed using SPL. In such cases, a standard value (20 uPa in air) is used as reference.

4. In your methods you state:

"Briefly, acoustic stimuli were presented via custom-designed plastic tubing connected to an MF1 Multi-Field Magnetic Speaker (TDT, USA) positioned in the ear canal. Puretone

stimuli of 10- ms (0.5- ms rise/fall) were presented at 21.1/s across intended frequencies, starting from 90 dB SPL and decreasing in steps of 5 dB. Evoked responses were amplified 10,000× (PA4 bio-amplifier, TDT) and averaged 500 times with 0.3 to 3 kHz band-pass 565 filter.”

Thus, at the very beginning of your measurements you presented 500 intense tone bursts. The higher frequency regions of the mouse cochlea (above about 30 kHz) are very noise sensitive, and your regime could prejudice the sensitivity of the cochlea. In the methods, or in the results section, please explain why you chose to deliver the tones descending from a high to low levels instead of ascending from low to high levels, or perhaps with a degree of randomization in the presentation of both frequencies and sound levels, which would tend to preserve hearing sensitivity.

Thanks for your comments. The reason to start testing sequence from high sound level (90 dB SPL) is to make sure that the mouse has a good ABR response. This approach reduces uncertainty due to any wrong setting for the test. The impact of high-level stimulation on hearing sensitivity is unlikely based on several facts. First, we do not need to repeat 500 times in each trial at high sound level(s), as less as 100-200 repeats is usually good enough to obtain clear responses. Secondly, we do not really do 5-dB steps at high sound level but drop by 10 or even 20 dB. 5-dB steps are used when the test approaches the threshold. In this way, the exact noise exposure is trivial. Thirdly, the tone burst exposure at 90 dB SPL with the rate used in our testing does not significantly change the ABR threshold even up to thousands of repeats. Ideally, randomization in level should minimize the potential impact of high-level tone exposure on thresholds. But our equipment does not allow such a randomization. However, we are confident that the method we use is safe. In addition, we stuck the method in the same way across all the tests. Therefore, the results should reasonably reflect the real change in hearing sensitivity. Finally, the down-sequencing in ABR test is generally acceptable.

To address whether our descending-level protocol (90 → lower dB SPL) affects cochlear sensitivity, we performed direct comparisons in a cohort of mice (n = 6) using:

- (1) Low-to-high intensity ascending protocol
- (2) Our high-to-low intensity descending protocol

on the same ears, with all frequencies tested at 500 repetitions per stimulus level.

We found no significant differences in ABR thresholds at any frequency (8, 16, 32kHz) and in ABR wave-I amplitude at different sound level (60, 70, 80 and 90 dB SPL).

According to the data you presented, this does not appear to be the case. This finding is to be expected, according to recent papers e.g <https://doi.org/10.1523/ENEURO.0250-18.2018>. In the quoted paper, the noise level was 100 dB SPL, which is 10 dB SPL higher than the loudest test tones used in your experiments. All four ABR peaks were reduced in size by the noise exposure, but peaks 2 to 4 were sensitized. Peak 1 arises directly from the response of the auditory nerve. Peaks 2 to 4 arise from excitatory, inhibitory, interaction between neurons in the brainstem. There is evidence of this in your figure, especially in response to 16 kHz and 32 kHz tones. You say in the paper: "Thresholds were defined as the lowest stimulus level that produced two or more discernible ABR waveforms (waves 1 to 5)." Measurements should have been based on the responses of the first peak of the ABR, if the measurements are meant to most closely reflect the responses of the OHCs. Later peaks reflect brainstem responses that can be modified by neuronal interaction. These comments pertain to your choice of measurement, which you found convenient, but DPOAE measurement, which only involve carefully placing a speculum in the auditory meatus of a lightly anaesthetised mouse, provides a more direct measure of OHC responses.

These data confirmed our stimulus paradigm does not compromise ABR testing sensitivity. While our descending-level protocol does not affect standard ABR metrics (thresholds/amplitude),

we acknowledge that subtler impacts on hearing sensitivity—potentially detectable through compound action potential or other electrophysiological measures. This methodological consideration provides valuable guidance for our future experimental design where ultra-high-frequency sensitivity preservation is paramount.

This is not really the case. Compound AP, or single fiber, measurements are invasive and indirectly signal the responses of the afferent fibres. CM measurements can be confusing and provide misleading information, especially in noise studies. See: *J Neurophysiol* 127: 1574–1585, 2022. doi:10.1152/jn.00402.2021.

Here, we revised one sentence in the method section to address the sequencing: (Line 601-603)

At each frequency, the test started from 90 dB SPL to ensure a clear response; and stepped down in 10 dB steps at high levels and then 5 dB when approaching the threshold.

I would keep this revised sentence. My comments are meant for consideration in designing further experiments and improvements in future experimental techniques.

5. Figure 1i. In the methods you state:

“In DPOAEs testing, the primary tones f_1 and f_2 were delivered through two MF1 loudspeakers, respectively, positioned in a closed-field arrangement. A custom designed probe equipped with a low-noise ER10B+ microphone (TDT, USA) facilitated precise stimulus presentation and recording. DPOAEs were measured with a fixed f_2/f_1 ratio of 1.2, where the intensity of f_2 (L_2) was set to 10 dB lower than f_1 (L_1). Frequencies ranged from 5.6 to 32 kHz in half-octave intervals. For each f_2 , L_2 was varied from 70 to 10 dB SPL in 5 dB steps. The $2f_1-f_2$ DPOAE amplitude and the surrounding noise floor were extracted for offline analysis.”

In the caption to Fig1i:

“i. DPOAE amplitudes were higher in Gsdmd-SC-cKO mice compared to Gsdmdflox/flox mice at 14 d-PNE (solid line), with no significant differences at Prenoise exposure (dashed line).”

Again it seems you began your measurements of DPOAE at high levels and descended the SPL to low levels, which may have prejudiced the sensitivity of the cochlea. It may have been helpful to design a protocol that extracted the DPOAEs continuously during the measurement. Such a protocol would enable you to monitor the DPOAEs directly and to store them and the raw data. This would have provided you with an immediate visual display of the sensitivity of the cochlea. Can you please explain why you chose your approach and not ones that enabled you to monitor DPOAE directly and to start from low SPLs to high, to preserve cochlear sensitivity.

Thanks for the comments and suggestions. Again, we start our DPOAE from high sound level to ensure that we have a good result, which can be jeopardized by wrong insertion of earpiece and other actions in setting up. Do DPOAE in upward sequencing would end up with failure thereafter and waste more time, which is limited by the short effective time of the anesthesia. The anesthesia we choice is the best in terms of safety, but the appropriate anesthetics can be maintained for less than one hour with the light anesthesia protocol. This approach is taken due to the concern of losing animals.

Considering the measurements of both ABR and DPOAE, we must do everything quickly. In addition, the highest sound level for DPOAE is 70-80 dB SPL, noise exposure at level is generally safe, there is not data supporting a significant HL by noise exposure at such a low level, especially for such a short duration.

Regarding the continuous monitoring of DPOAE, we don't really understand the point of the reviewer in terms of “immediate visual display of the sensitivity of the cochlea”. We guess that the reviewer is concerned about the change of cochlear sensitivity due to the manipulation of the DPOAE test. The first reason for not taking the suggested approach is that our system does not allow us to do a continuous recording with an immediate visual display.

Our lab also present DPOAE data in the frequency domain, but we also have severe time restraints on our data collection and collect only 10s, rather than 100s of samples at each frequency. We thus get data that can be displayed very quickly. This technique might be further improved by a quasi-randomised technique where sampling could be stopped at particular frequencies and levels where the sampling has met a pre-determined signal to noise criteria. We have yet to implement such a technique.

DPOAE results are shown in frequency domain, while recording is the temporal waveform. Converting the signal in real time to frequency domain is very challenging. Therefore, we record the waveform in time domain for each frequency and level long enough for averaging and converting to frequency domain signal, which is saved. Moreover, continuous monitoring requires stimulation to be fixed in level and frequency, while a comprehensive evaluation on DPOAE requires it to be tested across a wide range of frequency and level. In conclusion, we set up our DPOAE protocol with careful planning and tried different ways. We are pleased to validate your proposed approach in follow-up studies. Should it demonstrate superior protection of auditory sensitivity, we will refine our current experimental protocol accordingly. We sincerely appreciate your rigorous perspective, which strengthens our methodology.

I have no comments on your responses from and including 6 onwards

Reviewer #2

(Remarks to the Author)

The authors have addressed the comments appropriately in the revised manuscript.

Reviewer #3

(Remarks to the Author)

In the revised version of this manuscript, the authors added data using Caspase-11 single KO mice, Gsdmd-supporting cell (SC)-cKO mice, AAV.ie-CMV-Gpx1 mice, and AAV.ie-SC-Gpx4 mice. These datasets are impressive and provide valuable insights for readers into the complex intercellular signaling pathways between cochlear hair cells (HCs) and supporting cells. However, I would like to suggest additional revisions on some specific points. Each numbered comment corresponds to the numbering in the authors' response. At the end, I have included a crucial comment addressing an important enigma in the signaling pathway.

3. As the authors suggest, GSDMD in non-SC cells (including HCs) may play a beneficial role based on Fig. 1h, i, k. However, in Extended Data Fig. S3h, k, GSDMD-HC-cKO mice did not exhibit higher ABR thresholds or greater OHC loss than control mice. If GSDMD in HCs does not contribute to protection against acoustic trauma, in which cochlear cell types do the authors hypothesize that GSDMD exerts its protective effect? For example, could this involve other cell components within the organ of Corti or specific populations in the lateral wall? Clarifying this point would help readers better understand the proposed cellular targets and mechanisms.

6. I would like to suggest that the authors include in the revised manuscript the images of the SC layer (prepared in their response) showing that Sox2-positive SCs and 4HNE-positive cells (HCs) do not overlap. Adding these data will make the point clearer to readers and strengthen the manuscript.

7. In Fig. 6c, d, the authors demonstrate that GPX4 is downregulated in HCs at 2 hours post noise exposure, but they do not provide evidence that GPX4 expression in SCs is unchanged. The cryosection images provided in their response are not explicit on this point. I therefore suggest that the authors include representative images of the SC layer in the manuscript, in order to clearly demonstrate HC specificity in the role of GPX4.

11. In the revised manuscript, the authors acknowledge the multifunctional nature of ERK in SCs but there are more key studies demonstrating ERK's protective effect on HCs. For completeness and balance, I recommend citing representative work in this area, including the study from Kyoto University (Insulin-like growth factor 1 inhibits hair cell apoptosis and promotes the cell cycle of supporting cells by activating different downstream cascades after pharmacological hair cell injury in neonatal mice. Netrin 1 mediates protective effects exerted by insulin-like growth factor 1 on cochlear hair cells.), which specifically demonstrated ERK-mediated HC protection. Including these references will provide readers with a more accurate and nuanced understanding of ERK's dual roles in cochlear SCs.

There are several types of SCs in the organ of Corti, including pillar cells, Deiter's cells, Hensen's cells, Claudius' cells, and inner phalangeal cells. Each SC type has distinct phenotypes and roles in supporting HCs. Overall, it would be helpful if the authors could clarify which specific SC types constitute or are most crucial in the intercellular signaling pathway described in this manuscript.

Version 3:

Reviewer comments:

Reviewer #1

(Remarks to the Author)

My comments are focused on the physiological methods employed in the investigation of "ROS-GSDMD feedback loop between hair cells and supporting cells governs noise-induced hearing loss". I am satisfied with the authors' responses to my questions and comments.

Reviewer #3

(Remarks to the Author)

In the revised version of this manuscript, the authors added some discussion in response to my earlier comments. While no new experiments were performed, they outlined future directions. I have one additional suggestion:

7. I would like to suggest that the authors include in the revised manuscript the images and fluorescence analysis of GPX4 expression showing that there was no significant difference in its expression in supporting cells between pre-NE and 2 h-PNE. Adding these data will help readers better understand the intercellular signaling pathway between hair cells and supporting cells.

We thank the reviewers for their thorough and constructive feedback. Addressing these comments has significantly strengthened our study. In revising the manuscript, we have conducted additional experiments and comprehensively addressed all points through careful analysis of available data.

The paper: “ROS-GSDMD feedback loop between hair cells and supporting cells governs noise-induced hearing loss” reports some potentially important findings.

Namely, they “reveal a paradigm-shifting mechanism where supporting cells (SCs) orchestrate NIHL through Gasdermin D (GSDMD) with a non-canonical pathway.

Conditional knockout of GSDMD in SCs, but not in HCs, significantly reduces NIHL.”

My main expertise is in cochlear physiology, and I have focussed largely on the physiological measurements reported in the paper. The physiological findings are important because they underpin the major findings reported in the paper.

1. To assess the hearing of the mice, the authors used two non-invasive techniques. They measured the ABR and DPOAEs. ABR measurements provide an overall measure of the hearing sensitivity of the mice in their experiments. This measure does not, on its own, enable the experimenters to discover the precise origin of any alteration in the hearing of the mice from control.

We acknowledge your valid point regarding ABR's limitations in localizing the precise origin of hearing alterations. However, ABR remains the best option for non-invasive, frequency-selective assessment of hearing sensitivity, which is the major concern to the present study. It is well recognized that NIHL impacts mainly on the function of outer hair cells (OHCs) which are responsible for the hearing sensitivity. Therefore, the location of the hearing loss is already clear and is not the focus of the present study. Furthermore, ABR is the ONLY method that can be tested repeatedly (as needed in the present study) in small rodents.

While ABR cannot differentiate sensory versus neural contributions, we mitigated this limitation through complementary approaches: (1) DPOAEs specifically evaluated OHC function. (2) Cytocochleograms quantified sensory cell loss. (3) Ribbon synaptic counts assessed integrity of IHC-to-SGNs transmission. This multimodal strategy ensures ABR threshold shifts primarily reflect OHC-mediated sensitivity loss - the core focus of our NIHL investigation.

2. DPOAE measurements reflect the performance of the nonlinear amplification of the outer hair cells. As a first point, and to enable an assessment of the quality of the data, will the authors please state in the methods the intrinsic distortion levels of the measurement system and hence its threshold for measuring acoustic distortion.

Thanks for the suggestion. We revised the method section by adding the examination of the intrinsic distortion as shown below: (**Line 613-618** in the revised manuscript)

The potential intrinsic distortion and the noise floor were examined by recording the sound via the coupling cavity with the stimulation used for DPOAE. The noise floor of the testing system was below 0 dB SPL and there was no significant intrinsic distortion when tested via the physical coupling cavity utilized for the DPOAE. DPOAE threshold was defined as the interpolated value of f_2 intensity required to generate a 0 dB SPL DPOAE.

3. Figures 1e, h. What is “dB” relative to? Is it a value common to all measurements such as SPL? Or is it relative to each measurement? Ideally it should be referred to a commonly accepted standard, like SPL. It is easier to understand. Could you please, therefor, clarify what is meant by “dB” and ideally, for increased clarity and understanding, convert to SPL.

Figure 1e and h concern level shift, which is relative to baseline values, they should not be relative to a standard like SPL but relative to each other, since a shift would be the same in dB no matter whether SPL or other metric is used. Please note that dB SPL is used in d and g where the absolute thresholds are addressed using SPL. In such cases, a standard value (20 uPa in air) is used as reference.

4. In your methods you state:

“Briefly, acoustic stimuli were presented via custom-designed plastic tubing connected

to an MF1 Multi-Field Magnetic Speaker (TDT, USA) positioned in the ear canal. Pure-tone stimuli of 10- ms (0.5- ms rise/fall) were presented at 21.1/s across intended frequencies, starting from 90 dB SPL and decreasing in steps of 5 dB. Evoked responses were amplified 10,000× (PA4 bio-amplifier, TDT) and averaged 500 times with 0.3 to 3 kHz band-pass 565 filter."

Thus, at the very beginning of your measurements you presented 500 intense tone bursts. The higher frequency regions of the mouse cochlea (above about 30 kHz) are very noise sensitive, and your regime could prejudice the sensitivity of the cochlea. In the methods, or in the results section, please explain why you chose to deliver the tones descending from a high to low levels instead of ascending from low to high levels, or perhaps with a degree of randomization in the presentation of both frequencies and sound levels, which would tend to preserve hearing sensitivity.

Thanks for your comments. The reason to start testing sequence from high sound level (90 dB SPL) is to make sure that the mouse has a good ABR response. This approach reduces uncertainty due to any wrong setting for the test. The impact of high-level stimulation on hearing sensitivity is unlikely based on several facts. First, we do not need to repeat 500 times in each trial at high sound level(s), as less as 100-200 repeats is usually good enough to obtain clear responses. Secondly, we do not really do 5-dB steps at high sound level but drop by 10 or even 20 dB. 5-dB steps are used when the test approaches the threshold. In this way, the exact noise exposure is trivial. Thirdly, the tone burst exposure at 90 dB SPL with the rate used in our testing does not significantly change the ABR threshold even up to thousands of repeats. Ideally, randomization in level should minimize the potential impact of high-level tone exposure on thresholds. But our equipment does not allow such a randomization. However, we are confident that the method we use is safe. In addition, we stuck the method in the same way across all the tests. Therefore, the results should reasonably reflect the real change in hearing sensitivity. Finally, the down-sequencing in ABR test is generally acceptable.

To address whether our descending-level protocol (90 → lower dB SPL) affects cochlear sensitivity, we performed direct comparisons in a cohort of mice (n = 6) using:

- (1) Low-to-high intensity ascending protocol
- (2) Our high-to-low intensity descending protocol

on the same ears, with all frequencies tested at 500 repetitions per stimulus level.

We found no significant differences in ABR thresholds at any frequency (8, 16, 32kHz) and in ABR wave-I amplitude at different sound level (60, 70, 80 and 90 dB SPL). These data confirmed our stimulus paradigm does not compromise ABR testing sensitivity. While our descending-level protocol does not affect standard ABR metrics (thresholds/amplitude), we acknowledge that subtler impacts on hearing sensitivity—potentially detectable through compound action potential or other electrophysiological measures. This methodological consideration provides valuable guidance for our future experimental design where ultra-high-frequency sensitivity preservation is paramount.

Here, we revised one sentence in the method section to address the sequencing: (*Line 601-603*)

At each frequency, the test started from 90 dB SPL to ensure a clear response; and stepped down in 10 dB steps at high levels and then 5 dB when approaching the threshold.

5. Figure 1i. In the methods you state:

"In DPOAEs testing, the primary tones f1 and f2 were delivered through two MF1 loudspeakers, respectively, positioned in a closed-field arrangement. A custom-designed probe equipped with a low-noise ER10B+ microphone (TDT, USA) facilitated precise stimulus presentation and recording. DPOAEs were measured with a fixed f2/f1 ratio of 1.2, where the intensity of f2 (L2) was set to 10 dB lower than f1 (L1). Frequencies ranged from 5.6 to 32 kHz in half-octave intervals. For each f2, L2 was varied from 70 to 10 dB SPL in 5 dB steps. The 2f1-f2 DPOAE amplitude and the surrounding noise floor were extracted for offline analysis."

In the caption to Fig 1i:

" i. DPOAE amplitudes were higher in Gsdmd-SC-cKO mice compared to Gsdmdflox/flox mice at 14 d-PNE (solid line), with no significant differences at Pre-noise exposure (dashed line)."

Again it seems you began your measurements of DPOAE at high levels and descended the SPL to low levels, which may have prejudiced the sensitivity of the cochlea. It may have been helpful to design a protocol that extracted the DPOAEs continuously during the measurement. Such a protocol would enable you to monitor the DPOAES directly and to store them and the raw data. This would have provided you with an immediate visual display of the sensitivity of the cochlea. Can you please explain why you chose your approach and not ones that enabled you to monitor DPOAE directly and to start from low SPLs to high, to preserve cochlear sensitivity.

Thanks for the comments and suggestions. Again, we start our DPOAE from high sound level to ensure that we have a good result, which can be jeopardized by wrong insertion of earpiece and other actions in setting up. Do DPOAE in upward sequencing would end up with failure thereafter and waste more time, which is limited by the short effective time of the anesthesia. The anesthesia we choice is the best in terms of safety, but the appropriate anesthetics can be maintained for less than one hour with the light anesthesia protocol. This approach is taken due to the concern of losing animals. Considering the measurements of both ABR and DPOAE, we must do everything quickly. In addition, the highest sound level for DPOAE is 70-80 dB SPL, noise exposure at level is generally safe, there is not data supporting a significant HL by noise exposure at such a low level, especially for such a short duration.

Regarding the continuous monitoring of DPOAE, we don't really understand the point of the reviewer in terms of "immediate visual display of the sensitivity of the cochlea".

We guess that the reviewer is concerned about the change of cochlear sensitivity due to the manipulation of the DPOAE test. The first reason for not taking the suggested approach is that our system does not allow us to do a continuous recording with an immediate visual display. DPOAE results are shown in frequency domain, while recording is the temporal waveform. Converting the signal in real time to frequency domain is very challenging. Therefore, we record the waveform in time domain for each frequency and level long enough for averaging and converting to frequency domain signal, which is saved. Moreover, continuous monitoring requires stimulation to be fixed in level and frequency, while a comprehensive evaluation on DPOAE requires it to be tested across a wide range of frequency and level. In conclusion, we set up our DPOAE protocol with careful planning and tried different ways. We are pleased to validate your proposed approach in follow-up studies. Should it demonstrate superior protection of auditory sensitivity, we will refine our current experimental protocol accordingly. We sincerely appreciate your rigorous perspective, which strengthens our methodology.

6. Also, please state the SPL at which you measured the DPOAEs. This does not appear to be given in the caption of Figure 1. Additionally, please define dBV and why DPOAE magnitude is not given as SPL.

We thank the reviewer for identifying these important issues. Regarding Fig. 1:

(1) The DPOAE data were acquired in down sequence starting at 80 dB SPL F1 stimulation

(2) The unit was incorrectly labeled as dBV due to a documentation error. Per TDT's DPOAE conversion protocol (<https://www.tdt.com/docs/dpoe-user-guide/converting-dbv-to-db-spl>), values should be presented in dB SPL.

We further acknowledge your valuable suggestion about DPOAE amplitude interpretation. Given the nonlinear response pattern (including plateau effects) of DPOAE responses across intensity levels, we agree that single-intensity amplitude measurements risk ambiguous interpretation. Accordingly, we have replaced the DPOAE amplitude-frequency curves (responses at L2 = 70 dB SPL) with frequency-specific DPOAE thresholds in the revised figures (**Fig. 1i, 5g, 5h**). This approach provides a more physiologically meaningful characterization of OHC function.

7. I strongly suggest that the presentation of DPOAE ISO response curves would have given a clearer understanding of the frequency dependence of DPOAE generation in the cochlea under different conditions. Amplitude can give a confusing impression because, especially at high levels, e.g F1 at 50 dB SPL, DPOAE increase with level is very nonlinear and can reach a plateau, depending also on stimulus frequency.

Thanks. We assume that the ISO response refers to the level-frequency curve to show that, at each frequency, what sound level can evoke a target DPOAE amplitude across frequencies. We think that revised threshold-frequency curve serves this purpose since the threshold is determined by a certain amount of response amplitude above the noise floor (**Fig. 1i, 5g, 5h**). We revised the method section by adding the definition of the DPOAE threshold in **Line 617-618** of the revised manuscript and copied here:

DPOAE threshold was defined as the interpolated value of f_2 intensity required to generate a 0 dB SPL DPOAE.

8. Figure 1k. There appears to be no OHC loss for frequencies below about the 3.5 mm location, from apex. According to a well-known mouse cochlear frequency map (Müller et al., Hearing Research 202 (2005) 63–73), this is close to the 34 kHz frequency place. Do these findings, in part, account for why there appears to be no complete recovery from NIHL?

To our knowledge, it is common to see discrepancy between functional deterioration and OHC loss in the study of NIHL. The threshold elevation is better matched by OHC loss in very high frequency region but in low frequency region, we often see threshold shift without any OHC loss. It is concluded that OHCs may have been functionally damaged but keep morphological integrity as shown by the most common methods for HC count.

(see Chen G D, Fechter L D. The relationship between noise-induced hearing loss and hair cell loss in rats. Hear Res, 2003, 177(1-2): 81-90 as an exemplary research).

Our result showed also the discrepancy at the low frequency region below 34 kHz, suggesting incomplete recovery of hearing sensitivity.

9. Figure 1m. Please state that this figure is based on sampling in the 32 kHz region where OHC loss is expected to be minimal, or indeed absent.

The sampling location for synapse investigation is stated in the figure legend of 1l as 32 kHz region. 1m is the statistical evaluation of sample in 1l. To make it clearer, we revised the figure legend of 1m as follow (**Line 154-156**):

Statistics of samples shown in 1 from 32 kHz region where OHC loss is minimal, or indeed absent, showing no statistical differences in synapse density (averaged from eight IHCs per cochlea) across groups.

10. A remarkable finding, not commented on in the paper, is that there is no hair cell loss at the location of the noise exposure or at a location a half octave above this where

noise is most effective at desensitizing the cochlea (Cody and Johnstone, 1981 J. Acoust. Sot. Am. 70, 707-711; Cody and Russell, 1985, Nature 315, 662-665). Instead, hair cell loss is confined to frequencies above about 34 kHz, a region where the cochlea is most sensitive noise insult. Thus presentation of an 8-16 kHz noise band causes no measurable change in sensitivity in the cochlea region apical to this, it causes loss of sensitivity to all frequency regions at and basal to the 8-16 kHz band, and causes 0% - 20% OHC loss at frequencies a half octave above the 8-16 kHz band and up to 100% OHC loss at frequencies above this (in the 40 kHz – 90 kHz region). Could it be that there are differences in the action of Gasdermin D on NIHL at the bases and hook region of the cochlea from its action in the more apical regions? Perhaps this point could be raised in the discussion.

The famous half-octave law in NIHL is not established on HC loss. The study by Cody and Johnstone (JASA 1981) was on TTS of single unit ANF. No data on OHC loss was seen in this paper. This is also true for the study by Cody and Russell 1985 (Cody A R, Russell I J. Outer hair cells in the mammalian cochlea and noise-induced hearing loss. Nature, 1985, 315: 662-665.) in which no HC count data was available. As we responded above, there is a big discrepancy between threshold shifts and OHC loss in the low frequency region after NIHL was established: often time, the threshold shift is NOT matched by the amount of OHC loss.

Using tone of middle frequency (8-16) is a common practice in the studies of NIHL in mice

(1. Furman A C, Kujawa S G, Liberman M C. Noise-induced cochlear neuropathy is selective for fibers with low spontaneous rates. *J Neurophysiol*, 2013, 110(3): 577-586. DOI: 10.1152/jn.00164.2013

2. Kujawa S G, Liberman M C. Adding insult to injury: cochlear nerve degeneration after "temporary" noise-induced hearing loss. *J Neurosci*, 2009, 29(45): 14077-14085. DOI: 29/45/14077 10.1523/JNEUROSCI.2845-09.2009

3. Maison S F, Liberman M C. Predicting vulnerability to acoustic injury with a noninvasive assay

of olivocochlear reflex strength. J Neurosci, 2000, 20(12): 4701-4707.)

Exposure at such frequency can cause NIHL spread to almost the whole cochlea, more towards the high-frequency end. With the intensity and duration used in the present experiment, OHC loss is expected to be limited to the very high-frequency region (particularly in the hook area), and not in the region half an octave above the frequency region of acoustic overstimulation.

The question “Could it be that there are differences in the action of Gasdermin D on NIHL at the bases and hook region of the cochlea from its action in the more apical regions?” is difficult to answer since we do not have evidence to support or reject the speculation. The important fact to our study is the protective effect against noise-induced OHC loss appears and can only be demonstrated in the region where OHC loss is seen. Recent literature demonstrates that noise exposure induces a gradient increase in NOX2-positive OHCs from apical to basal turns, with NOX2 deficiency partially attenuating cochlear damage. This suggests basal OHCs experience higher oxidative stress, potentially predisposing this region to initiate the damaging ROS-GSDMD positive feedback loop.

Qi, M., Gao, Z., Qiu, Y., Wang, R., Tian, K., Yue, B., Zhang, X., Zhang, P., Wu, Z., Zhu, Q., Liu, Z., Ma, Z., Zhou, X., Han, Y., Chen, J. et al. NOX2 Contributes to High-Frequency Outer Hair Cell Vulnerability in the Cochlea. Advanced Science (2025).

However, due to the current lack of a specific antibody reliably detecting the GSDMD-N fragment, it is currently unfeasible to definitively determine via immunolabeling whether noise exposure indeed triggers a higher gradient of GSDMD activation specifically within the basal turn. We have revised the Discussion about this limitation: **(Line 476-480, New data in Extended Data Fig. S11)**

However, spatial mapping of GSDMD activation remains technically constrained: Commercially available GSDMD-N antibodies showed non-specific binding in cochlear tissue and failed validation using Gsdmd-KO controls, precluding reliable immunolabeling of GSDMD-N fragments in HCs or SCs.

We thank the reviewers for their thorough and constructive feedback. Addressing these comments has significantly strengthened our study. In revising the manuscript, we have conducted additional experiments and comprehensively addressed all points through careful analysis of available data.

This manuscript by Xiao et al has elucidated mechanisms by which non-sensory cochlear supporting cells (SCs) interact with sensory hair cells (HCs) through Gasdermin D (GSDMD) to influence noise-induced loss of hearing and HCs. Using SC specific GSDMD knockout along with neurophysiology and molecular biology approaches, authors report that noise-induced oxidative stress in HC causes activation of GSDMD in SC through EGFR-p-ERK-Caspase-11/1 pathway in SC. Such GSDMD activation in SCs, in turn augments oxidative damage in HCs leading to their death and thus, forming a positive loop between SCs and HCs. Knocking out GSDMD from SC completely abolishes oxidative damage in HCs. Moreover, blocking GSDMD in SCs but not in HCs or overexpressing antioxidant enzyme GPX4 in HCs significantly reduces noise-induced hearing loss and preserves outer hair cells. Overall, this is a rigorously designed and well written manuscript highlighting another critical role for SCs in regulating hair cell degeneration in the context of noise-induced hearing loss. This study also reveals molecular mechanisms of coupling of oxidative stress with inflammation in auditory pathophysiology.

Following are some concerns that need author's attention.

1. The major weakness in the manuscript is the evidence for how activated GSDMD-N in SCs increases oxidative stress in HCs, leading to their degeneration. It will strengthen the manuscript if authors could show that GSDMD-N localizes to mitochondria in HCs and mediate mitochondrial damage and increase inflammatory cytokines and ROS production.

Thank you for raising this point. We fully acknowledge the lack of evidence how GSDMD-N in supporting cells (SCs) mediate oxidative stress in hair cells (HCs) as the key limitation of our current study and have addressed it in our Discussion. But our data points the role of SC-HC link. It is NOT likely that GSDMD-N in HC mitochondria would play a role in this link, since *Gsdmd*-HC-cKO mice demonstrate that GSDMD activation within HCs is NOT the primary driver of noise-induced HC degeneration, as its conditional knockout (cKO) in HCs provided no protective effect (Extended Data Fig. S3f-k). Therefore, the effect of GSDMD-N in SCs on increasing oxidative stress in HCs likely occurs independently of GSDMD activation within the HCs themselves. In contrast to its unclear role in auditory research, recent study showed that GSDMD-N can be released via extracellular vesicles, and the fragment has the potential to induce pyroptosis in recipient cells.

Wright, S.S., Kumari, P., Fraile-Ágreda, V., Wang, C., Shivcharan, S., Kappelhoff, S., Margheritis, E.G., Matz, A., Vasudevan, S.O., Rubio, I., Bauer, M., Zhou, B., Vanaja, S.K., Cosentino, K., Ruan, J. et al. Transplantation of gasdermin pores by extracellular vesicles propagates pyroptosis to bystander cells. Cell 188, 280-291.e17 (2025).

This suggests a plausible alternative mechanism whereby GSDMD-N could be transferred indirectly from SCs to HCs—potentially localizing to the HC membrane to exert its damaging effects. Unfortunately, technical limitations prevented direct visualization of this potential transfer. Commercially available antibodies for GSDMD-N immunolabeling, which we obtained, exhibited non-specific binding in cochlear tissues. Critically, both antibodies failed validation using cochlear tissue from global *Gsdmd*-KO mice, confirming their lack of specificity in this context (***New data in Extended Data Fig. S11***). Consequently, we were unable to employ antibody-based methods to detect GSDMD-N fragments or pores within HCs or SCs.

We have expanded the Discussion section to incorporate the potential role of intercellular transfer of GSDMD-N as a noteworthy pathway warranting further investigation (**Line 469-480**):

Recent study revealed that extracellular vesicles can transplant GSDMD pores onto the plasma membrane of bystander cells, inducing their death⁴³. In the cochlear context, exosomes carrying heat shock proteins have been shown to protect against aminoglycoside-induced vestibular HCs' death via paracrine signaling⁴⁴. These findings collectively suggest a plausible mechanism whereby GSDMD-N could be transferred indirectly—potentially via extracellular vesicles—from supporting cells to hair cells, subsequently exerting its damaging effect by integrating into the hair cell membrane and directly inducing pyroptosis. However, spatial mapping of GSDMD activation remains technically constrained. Our commercially available GSDMD-N antibodies showed non-specific binding in cochlear tissue and failed validation using Gsdmd-KO cochlear sample (Extended Data Fig. S11), precluding reliable immunolabeling of GSDMD-N fragments in HCs or SCs.

2. Authors should show the expression of GSDMD after noise exposure by immunohistochemistry to verify its expression or upregulation in HCs after injury.

Recently, we performed additional cochlear immunohistochemistry using a GSDMD antibody rigorously validated against cochlear tissue from *Gsdmd* KO mice

Our analysis revealed no significant change in GSDMD immunolabeling within the HC region following noise exposure. Critically, this finding aligns with our functional data from *Gsdmd*-HC-cKO mice, which exhibited no protection against noise-induced cochlear damage (Extended Data Fig. S3f-k).

Taken together, these results demonstrate two key points:

- (1) GSDMD expression in HCs is not detectably upregulated by noise trauma.
- (2) HCs are not the primary cell type mediating GSDMD-dependent NIHL.

3. It is interesting that GSDMD KO or GPX4 overexpression does not seem to affect the ABR thresholds and DPOAE levels at 1 day post noise exposure but rather at 14 days. How do authors interpret these data. This should be discussed in the manuscript. Also, it will be important to discuss why the protection in GSDMD KO is selective to outer hair cells since GSDMD seems to be expressed by inner and outer pillar cells besides Claudius cells.

We appreciate your insightful questions regarding the temporal dynamics and cell-type specificity of protection.

(1) Temporal dynamics (Protection at 14 days, not 1 day):

Unlike the hearing loss caused by ototoxic drugs, NIHL has a specific temporal development: a large threshold shift is seen (and labelled as temporal threshold shift, or TTS) immediately after the noise exposure is ceased and is followed by a recovery period up to 2 weeks. The threshold shift left 2 weeks after the noise exposure is usually called permanent threshold shift (PTS). It has been recognized that the recoverable part of the hearing loss is largely due to the damage to the stereocilia that controls the mechanical-electrical transduction of HCs. This part of the damage has been found to be largely self-repairable and would not be influenced by GSDMD. However, PTS is primarily associated with damage to outer hair cell (OHC) bodies due to the accumulation of reactive oxygen species (ROS), leading to progressive and irreversible injury. The time-course of the apparent threshold change after the noise exposure is determined by the combination of the two processes. Our explanation for the protection observed at later time points, but not at 1 d-PNE, is that *Gsdmd* KO affects only the damage process responsible for PTS, which involves injury to cell bodies. In the wild-type (WT) mice of our experiment, the threshold recovery after 1 day post noise is small and is likely due to the accumulation of ROS stress, which largely “cancel” the effect of TTS recovery rooted from the self-repair of the stereocilia. In the KO mice, the accumulation of ROS is interrupted and therefore, we see much more “recovery” in ABR threshold. We are not sure if those explanations should be integrated into the manuscript. Please advise.

(2) No protection for IHCs (vs. OHCs):

The damage to inner hair cells (IHCs) by the noise could only be demonstrated by the synaptic loss since there is no IHC loss here. The reason for not seeing the protective effect by GSDMD-KO on the synapse count is likely due to the lack of connection between the synaptic damage and ROS stress. It is recognized that the synaptic damage is resulted from the over release of glutamate, the excitatory neurotransmitter for signaling from IHC to spiral ganglion neurons (SGN), which

damage the post-synaptic terminal of the synapses. The mechanism for the loss of the presynaptic component (the ribbon) is unclear right now. The damage of this part is related to the Ca^{2+} overload but not linked to the ROS stress which is the main target of GSDMD-related mechanisms for OHC protection. The failure of *Gsdmd* KO in preventing noise-induced ribbon synapse loss suggested that GSDMD activation is not a key mechanism for IHC synaptic injury. Although GSDMD is highly expressed in several subtypes of SCs, we cannot currently determine if GSDMD activation mediating noise-induced cochlear damage originates from specific SC subtype (e.g., outer pillar cells). Validating this would require generating GSDMD conditional knockout models targeting specific SC subpopulations. Furthermore, it remains possible that only OHCs, not IHCs, can establish a damaging ROS-GSDMD positive feedback loop with supporting cells. Both mechanisms represent plausible explanations for the observed OHC-selective protection in *Gsdmd* KO mice and merit further investigation.

We have expanded the Discussion section in **Line 507-512**:

Although GSDMD is highly expressed in several subtypes of SC, we cannot currently determine if GSDMD activation mediating cochlear damage originates from specific SC subtype (e.g., outer pillar cells). Furthermore, it remains possible that only OHCs, not IHCs, can establish a damaging ROS-GSDMD positive feedback loop with SCs, which may lead to the failure of Gsdmd KO in preventing IHC's ribbon synapse from noise-induced damage.

4. GSDMD knockdown in SCs (Pillar and Claudius cell) via AAV should be verified with immunohistochemistry besides Western blot.

We have now performed the additional immunohistochemistry verification as requested. Using the same antibody concentration and experimental conditions applied in our Western blot analysis, cochlear sections from WT and *Gsdmd*-SC-cKO mice were immunolabeled with the knockout-validated GSDMD antibody. Consistent with our

previous data, immunohistochemistry analysis confirmed a marked reduction in GSDMD immunoreactivity specifically within pillar and Claudius cells of *Gsdmd*-SC-cKO mice compared to control littermates. This result was incorporated into **Extended Data Fig. S2h**.

5. It will strengthen the manuscript if authors overexpress GSDMD-N protein in SCs and HCs to see if it reverses the protective effect of absence of GSDMD against noise-induced hearing loss.

We strongly agree that demonstrating functional reversal through GSDMD-N overexpression would significantly strengthen our conclusions. Initial characterization using dual-reporter mice confirmed efficient Cre-mediated recombination in SCs, but not HCs, with our AAV-ie-SC-Cre vector in adult mice. However, the efficiency of SC-promoter in driving Cre-expression significantly decreased in mature animals (>4 weeks old), limiting its utility for adult gain-of-function studies.

To address this technical constraint, we adopted a recently published SC promoter (*Lfng*) that maintains robust activity in adulthood. We constructed AAV-ie-*Lfng*-EGFP to validate promoter specificity and efficiency in adult mice. Following confirmation, we generated AAV-ie-*Lfng*-*Gsdmd*-N-Flag and delivered it neonatally to both WT and *Gsdmd* KO mice.

Due to revision constraints, we simultaneously packaged AAV-ie-Lfng-EGFP (transfection efficiency tracer) and AAV-ie-Lfng-*Gsdmd*-N-Flag (Addgene #80951). Adult WT mice injected with AAV-ie-Lfng-EGFP showed low transfection efficiency in supporting cells (SCs), confirming technical challenges. Paradoxically, both WT and *Gsdmd* KO mice injected with AAV-ie-Lfng-*Gsdmd*-N-Flag developed significant hearing loss. Western blot analysis of *Gsdmd* KO cochleae revealed a 35 kDa GSDMD-N band and a prominent 50 kDa band (hypothesized to represent oligomers). In WT cochleae, only the 50 kDa band was detected without the 35 kDa species. Given GSDMD-N's pore-forming oligomerization drives pathology, the exacerbated hearing loss in WT versus *Gsdmd* KO mice overexpressing GSDMD-N suggests accelerated oligomer assembly in endogenous GSDMD-expressing cells.

These results establish that SC-targeted GSDMD-N expression is sufficient to drive

cochlear degeneration independent of noise exposure, further underscoring its pivotal role in SC-mediated otopathology. The effect of this overexpression on hearing makes it impossible to test its potential to reverse the protection of *Gsdmd* KO against NIHL.

6. Figure 1, panel c, please check the immunolabel for hair cells; is it Spectrin or Myo7a.

Thank you for highlighting this detail. In Figure 1c, the immunolabel shows Myo7a (a hair cell-specific marker), not α II-Spectrin. Figure 1b displays α II-Spectrin labeling for cytoskeletal integrity. To prevent ambiguity, we have explicitly clarified these distinct immunolabels in the revised figure legend (*Line 138*).

7. It is unclear if the western blots shown in the manuscript are performed on protein lysates prepared from whole cochlea or isolated organ of Corti. If it is the former then that confounds the data due to the presence of other cochlear compartments like spiral ligament and stria vascularis that also expresses the factors in the pathway which are studied in the manuscript.

We appreciate your valid concern regarding the potential confounding effect of using whole cochlear lysates for Western blotting, given the expression of studied factors in compartments like the spiral ligament and stria vascularis. Isolation of the organ of Corti alone for Western blot analysis remains technically challenging in mice due to the cochlea's minute size and complex architecture. Consequently, as is common practice in the field, we utilized protein lysates prepared from whole cochlear tissue. We fully acknowledge this limitation and its implications for signal specificity.

While ideally we would perform immunostaining for all key molecules analyzed by WB, this faces a major hurdle——Lack of validated cleavage-specific antibodies: Commercially available antibodies specific for the cleaved/active forms (e.g., GSDMD-N, cleaved Caspase-1, Caspase-11 p26, mature IL-1 β) either do not exist or have not been reliably validated for specificity in cochlear tissue, preventing their

confident use for immunostaining in our system.

To address this inherent limitation and probe cell-specific molecular events, we implemented a complementary strategy:

(1) Cell-type-specific genetic manipulation: We employed HC-specific (*Gsdmd*-HC-cKO) and SC-specific (*Gsdmd*-SC-cKO) knockout mice, and HC-targeted GPX4 overexpression. This approach allows us to attribute functional outcomes and molecular changes primarily to the manipulated cell type, significantly reducing interpretative ambiguity from other cochlear compartments.

(2) Spatial validation via immunostaining: Recognizing that immunostaining can provide definitive spatial resolution, we performed critical validations:

1) Immunostained for the oxidative stress marker 4-HNE in HCs within *Gsdmd* KO and *Gsdmd*-SC-cKO mice (to assess downstream effects in HCs after SC manipulation).

2) Immunostained for p-ERK in HCs within GPX4-overexpression (OE) mice (to link SC manipulation to signaling pathway activation in HCs).

These IHC results provide direct spatial evidence supporting the proposed SC-HC signaling axis.

Thank you for prompting this important clarification. We agree that the use of whole cochlear lysates is a limitation for mechanistic granularity. We explicitly discussed this technical constraint and its potential implications in the revised Discussion section (**Line 514-520**):

*While whole cochlear lysates were used for WB to elucidate mechanism due to technical constraints, we mitigated interpretative limitations through complementary approaches: (1) Cell-type-specific manipulations functionally attributed molecular changes to targeted cells, such as *Gsdmd*-cKO in SCs and GPX4 overexpression in HCs; (2) Spatial immunostaining validated key SC-HC intercellular signaling events, including 4-HNE accumulation in HCs of *Gsdmd*-SC-cKO mice, and p-ERK level in SCs following HC-targeted GPX4 overexpression.*

8. Figure 3, it is unclear if GSDMD global or SCs conditional knockout was used to examine oxidative stress in HCs after acoustic trauma.

Thank you for raising this important clarification. For the data presented in Fig. 3, we utilized *Gsdmd* KO mice. However, acknowledging your valid concern about cell-type specificity, we have now performed additional experiments using *Gsdmd*-SC-cKO mice. WB analysis demonstrates that conditional knockout of GSDMD in SCs significantly reduces cochlear 4-HNE levels versus WT controls after noise exposure. Critically, immunostaining confirms markedly reduced 4-HNE within the hair cell region of *Gsdmd*-SC-cKO mice at 8 h post noise exposure. These results validate that GSDMD activation in SCs drives oxidative stress in HCs post-noise exposure, reinforcing our model of SC-mediated pathogenesis. We incorporated this data in the revised manuscript (*New data in Fig. 4e-h*).

9. How do authors interpret the data on preservation of IHC synapses with GPX4 overexpression but not with GSDMD knockout.

We appreciate your insightful question regarding the differential effects of GPX4 overexpression versus *Gsdmd* knockout on IHC synapse preservation. Under our noise

exposure paradigm, the primary sites of damage are OHCs and cochlear synapses. While *Gsdmd* KO specifically attenuated OHC loss, it did not significantly reduce synaptic loss (Fig. 1m). This indicates that GSDMD activation is not a primary driver of noise-induced synaptic injury. While GSDMD ablation protects OHCs via reduced ROS amplification, it does not prevent noise-induced GPX4 downregulation (new data: unchanged GPX4 levels in *Gsdmd* KO cochlea at 2h PNE, *New data in Extended Data Fig. S7*). Thus, *Gsdmd* KO fails to rescue the synaptic vulnerability potentially linked to GPX4 downregulation in cochlea.

In addition, the AAV-Php.eB vector used for GPX4 overexpression, driven by ubiquitous CMV promoter, leads to transgene expression not only in hair cells but also in SGNs. Prior evidence indicates that glutamate excitotoxicity thereby leading to activation of calcium-permeable AMPA receptors on the postsynaptic membrane, is a dominant mechanism for synaptic loss after noise trauma. GPX4's effects extend beyond the ROS-GSDMD axis, likely conferring direct antioxidant protection to SGNs against glutamate excitotoxicity.

Conclusion: The synapse protection seen with GPX4 OE, but not GSDMD KO, arises because:

- (1) Synaptic injury is primarily excitotoxicity-driven, not GSDMD-dependent;
- (2) GPX4 overexpression directly targets SGNs, counteracting synaptic damage;

(3) GPX4 confers cytoprotection broader than GSDMD pathway modulation alone.

10. Authors have indicated the use of male and female mice across the experiments; however the none of data hasn't been separated based on sex to determine if there is any sexual dimorphisms with the claimed phenomenon.

We appreciate your suggestion regarding potential sexual dimorphism in our observed phenomena and agree that sex-based analysis strengthens the study's rigor. We have now performed sex-stratified analyses for all key auditory functional phenotypes (ABR/DPOAE thresholds) across major experimental groups (*Gsdmd* KO, *Gsdmd*-SC-cKO, GPX4-OE, and relevant controls). No statistically significant sex-dependent differences were detected in: (1) Baseline hearing thresholds, (2) Noise-induced threshold shifts, and (3) Protective effects of genetic manipulations.

These results indicate that the core phenomena described in our study are consistent across sexes under the experimental conditions tested. The complete sex-stratified datasets have been added to the Supplementary Information (*New data in Extended Data Fig. S10*). Thank you for prompting this important validation.

11. Please check the references, since some of references related to ERK inhibitors in noise-induced hearing loss have not been cited.

We thank you for identifying this important literature gap. We have carefully evaluated the suggested work and incorporated it into our revised manuscript. Recently, we observed that ASN007 (a selective potent and oral inhibitor for ERK phosphorylation) also blocked ROS-GSDMD signaling. These data demonstrated p-ERK's pathogenic role in driving GSDMD activation and oxidative stress. (*New data in Extended Data Fig. S9, Line 385-386*)

We thank the reviewers for their thorough and constructive feedback. Addressing these comments has significantly strengthened our study. In revising the manuscript, we have conducted additional experiments and comprehensively addressed all points through careful analysis of available data.

This study by Xiao et al. presents an intriguing set of data on the intercellular signaling pathway between cochlear hair cells (HCs) and supporting cells (SCs) in response to noise exposure. Traditionally, HC death and synapse loss due to noise exposure have been discussed without considering SC interactions. However, SCs have recently been recognized for their diverse immune functions, which vary depending on the cochlear environment. This study contributes to this evolving understanding by demonstrating aspects of SC immune function. The authors effectively utilized two interesting AAV vectors, each selectively targeting either HCs or SCs. However, additional experiments using these vectors could be considered, as outlined below. Furthermore, spatial information is crucial when discussing signaling pathways between neighboring cells. To enhance this aspect, immunostaining should be presented alongside Western blotting.

Below are some specific notes that may help improve the manuscript.

1. Page 7, the authors used a cochlear SC-specific promoter for AAV-ie to efficiently transfer Cre to SCs. However, they do not specify which SC-specific gene's promoter was chosen for this experiment, either here or in the Methods section. Providing this detail would improve clarity and reproducibility.

We thank you for raising this valid point regarding promoter specificity, which is essential for reproducibility. The supporting cell (SC)-specific promoter used in our AAV-ie constructs was provided by OBio Technology (China, Shanghai), with commercial approval by Professor Guisheng Zhong's laboratory (ShanghaiTech University). Due to confidentiality obligations and ongoing intellectual property

protection, the promoter information may be disclosed at further time. We regret any inconvenience this limitation may cause and appreciate your understanding of these contractual constraints. We have now explicitly acknowledged this source in the Methods section (***Line 556-558***).

2. Page 7, Western blotting is used to confirm the effectiveness of *Gsdmd*-SC-cKO in Figure S2. Since the *Gsdmd* expression in SCs of the WT cochlea is clearly shown by immunostaining in Figure 1, its decrease in SCs of the cKO cochlea could be exhibited also by immunostaining.

In revised manuscript, we performed additional immunohistochemical analysis comparing GSDMD expression in *Gsdmd*-SC-cKO mice versus WT controls (***New data in Extended Data Fig. S2h***). Consistent with our WB data (Extended Data Fig.S2g), immunostaining revealed a marked reduction or absence of GSDMD signal specifically within SCs—notably in pillar cells and Claudius cells—of *Gsdmd*-SC-cKO cochlea. These results provide spatial validation of efficient GSDMD ablation achieved via AAV-ie-SC-Cre-mediated knockout.

3. Page 8, there was no difference in ABR thresholds and OHC loss after noise exposure between *Gsdmd*-HC-cKO mice and *Gsdmd* flox mice. However, the authors suggested in Page 7 that GSDMD in non-SC cells may play a protective role in acoustic trauma,

rather than the harmful role in SCs. If so, Gsdmd-HC-cKO should show elevated ABR thresholds and severer OHC loss as compared with Gsdmd flox mice.

Our idea about the protective role of GSDMD in non-SCs is based on the difference between global KO and SC selective KO of *Gsdmd*, since the global KO exhibits less protection as compared with SC selective KO (Fig. 1k). We then postulate that GSDMD in non-SC cells may play a beneficial effect. The global KO eliminate this good part of GSDMD effect, so that less effect of protection is seen as compared with SC-selective KO. This demonstrates the complexity of GSDMD-involved cellular process. We do not see how the idea stated above would lead logic error suggested by the last sentence in the reviewer's comment above, since GSDMD KO in hair cells (HCs) is irrelevant with the issue.

4. Page 10, the elevation of GSDMD-N following noise exposure is demonstrated by Western blotting. However, other cochlear cell types besides SCs may also express GSDMD in response to noise exposure. To exclude this possibility, it would be important to confirm that GSDMD-N elevation occurs specifically in SCs using immunostaining.

We fully agree that cell-type-specific localization of GSDMD-N would strengthen our mechanistic claims. To address this, we rigorously validated commercially available GSDMD-N antibodies (CST #36425 and HUABIO #ER1901-37). However, both antibodies lacked specificity in cochlear tissue, as evidenced by detectable bands in *Gsdmd* KO mice samples, precluding their reliable use for immunostaining in our system.

We acknowledge that using whole cochlear lysates limits mechanistic resolution. We explicitly discussed this technical constraint and its implications in the revised Discussion, ensuring proper contextualization of results. We appreciate this valuable suggestion for improved scientific rigor. (*Line 476-480 and 514-520*)

However, spatial mapping of GSDMD activation remains technically constrained. Our commercially available GSDMD-N antibodies showed non-specific binding in cochlear tissue and failed validation using *Gsdmd*-KO cochlear sample (Extended Data Fig. S11), precluding reliable immunolabeling of GSDMD-N fragments in HCs or SCs.

While whole cochlear lysates were used for WB to elucidate mechanism due to technical constraints, we mitigated interpretative limitations through complementary approaches: (1) Cell-type-specific manipulations functionally attributed molecular changes to targeted cells, such as *Gsdmd*-cKO in SCs and *GPX4* overexpression in HCs; (2) Spatial immunostaining validated key SC-HC intercellular signaling events, including 4-HNE accumulation in HCs of *Gsdmd*-SC-cKO mice, and *p*-ERK level in SCs following HC-targeted *GPX4* overexpression.

5. Page 11, Caspase-1/11 double KO mice are used to conclude that Caspase-11 is the primary mediator of GSDMD cleavage during noise exposure. However, to rule out potential interactions between Caspase-1 and Caspase-11, conducting experiments with Caspase-11 single KO mice would be important.

We agree that delineating the individual contributions of caspase-1 versus caspase-11

is essential for mechanistic clarity. According to your suggestion, we have obtained *Casp11* KO mice (generously provided by Prof. Feng Shao, National Institute of Biological Sciences, Beijing) and subjected them to our standard noise exposure paradigm. Western blot analysis of cochlear lysates at 8 h post-noise exposure revealed a complete absence of GSDMD-N cleavage product in *Casp11* KO mice group, but robust GSDMD-N signal in WT controls. (**New data in Fig. 2d, g; Line184-187**). This demonstrates that caspase-11 is necessary and sufficient for noise-induced GSDMD activation in our model, effectively ruling out compensatory interactions with caspase-1. The phenotype mirrors that of *Casp1/11* DKO mice (*Fig. 2b, e*), confirming caspase-11's primary role.

6. Page 13-14, global *Gsdmd* KO mice are used to conclude that activated GSDMD enhances noise-induced oxidative stress. However, to fully support the authors' hypothesis throughout the manuscript, experiments using *Gsdmd*-SC-cKO mice instead of the global ones would be crucial for clarifying the intercellular signaling pathway between SCs and HCs in response to sound exposure. Also, 4HNE expression in HCs is exhibited by immunostaining, but showing the SC layer in addition to the HC layer in this immunostaining would be important.

We appreciate your suggestions regarding cell-specific validation and spatial analysis. Critical experiments using *Gsdmd*-SC-cKO mice demonstrate significantly attenuated 4-HNE levels in cochlear lysates via Western blotting, with immunostaining confirming reduced oxidative stress specifically in outer hair cells (OHCs). Spatial analysis revealed concurrent 4-HNE elevation in both HCs and SCs of noise-exposed *Gsdmd*

flox/flox controls, whereas *Gsdmd*-SC-cKO mice exhibited selective 4-HNE reduction in HCs despite persistent oxidative stress in SCs. This compartmentalized protection pattern confirms that SC-specific GSDMD ablation shields HCs from oxidative damage despite ongoing SC stress, providing direct evidence for GSDMD-mediated intercellular (non-cell-autonomous) oxidative stress propagation. These results fully align with global KO data, establishing SCs as the primary site of GSDMD-dependent pathology. We incorporated this new data in the revised manuscript (**Fig. 4e-h**).

7. Page 16, the authors discuss the significance of GPX4 in HCs based on Western

blotting of the whole cochlea. Immunostaining for GPX4 to demonstrate its specific expression in HCs would be of great interest also here.

Prior study demonstrate that HC-specific *Gpx4* knockout induces hearing impairment, whereas SC-specific ablation shows no auditory phenotype, indicating GPX4's functional indispensability in HCs. In direct response to your suggestion, our immunostaining reveals robust GPX4 expression in both HCs and SCs. Notably, noise exposure triggered concomitant GPX4 downregulation in both cell types at 2h post noise exposure (2 h-PNE).

8. Page 17, GPX4 overexpression is shown to reduce noise-induced GSDMD activation and oxidative stress. To confirm that GPX1 does not contribute to this effect, additional experiments with GPX1 overexpression should be included.

We thank the reviewer for this suggestion. To specifically address the potential role of GPX1, we have performed additional experiments: Western blot analysis of cochlear lysates from mice injected with AAV-CMV-*Gpx1* revealed that GPX1 overexpression failed to significantly reduce 4-HNE levels or suppress GSDMD cleavage following noise exposure (*New data in Extended Data Fig. S6; Line 283-285*). These results demonstrate that, unlike GPX4, GPX1 overexpression is insufficient to confer protection against noise-induced oxidative stress or GSDMD activation.

9. Page 18, "sarifice" is a typo in Figure 4.

We thank you for catching this error. The typo "sarifice" in Figure 4 has been corrected to "sacrifice" in the revised manuscript. We have diligently conducted full proofreading of the text and figures to ensure manuscript accuracy and prevent similar errors.

10. Page 20, the authors demonstrate the critical role of GPX4 in HCs in downregulating ROS, using GPX4 overexpression system in HCs. However, in order to exclude the possibility that GPX4 may also function in SCs, additional experiments with GPX4 overexpression in SCs would be important.

We appreciate the reviewer's suggestion to clarify the cell-type specificity of GPX4 function. To directly address whether GPX4 overexpression in SCs confers protection, we generated AAV.ie-SC-*Gpx4*. Our findings reveal that GPX4 overexpression specifically in SCs failed to significantly mitigate noise-induced hearing loss. Furthermore, Western blot analysis demonstrated no significant reduction in GSDMD-

N levels following noise exposure in AAV.ie-SC-*Gpx4* mice compared to non-AAV controls. This result, combined with previous literature demonstrating that GPX4 knockout in SCs does not affect hearing, whereas knockout in HCs leads to auditory dysfunction, collectively suggests that GPX4 plays a critical role in maintaining HC function under both physiological and noise-exposure pathological conditions. The specific functional contribution of GPX4 in SCs warrants further investigation. These new data have been added to the Supplementary Information (*New data in Extended Data Fig. S8*).

11. Page 27, the authors conclude that phosphorylation of ERK in SCs occurs prior to HC damage in response to noise exposure. They also cite a study (Ref. 33) showing that ERK activation in SCs leads to HC damage. However, ERK is also well known for its role in cell survival and proliferation, and some studies report that ERK and Akt

activation in SCs can promote HC protection. These findings highlight the multifunctional nature of ERK in cochlear SCs. It would be valuable for the authors to discuss this broader context.

We thank the reviewer for this thoughtful comment. We have revised the Discussion to acknowledge its dual roles and emphasize that the specific outcome likely depends on the nature and duration of the stimulus. (*Line 445-452*)

Previous study also showed that p-ERK level is up-regulated in cochlear SCs upon mechanical and noise damage^{34,35}. Our pharmacological inhibition of p-ERK activity blocked noise-induced GSDMD-dependent damage cascades, indicating p-ERK exacerbates cochlear injury. This contrasts with prior work showing Erk2-cKO in HCs increased susceptibility to NIHL, particularly enhancing IHC loss, which suggesting ERK2 mediates protective signaling in HCs³⁶. This functional dichotomy demonstrates that EGFR-ERK signaling exhibits cell type- and context-dependent heterogeneity.

We thank the reviewer for these insightful and constructive suggestions, which have significantly improved the quality of our manuscript. In response, we have provided detailed point-by-point explanations and have supplemented additional data wherever feasible within the revision timeframe.

Reviewer's Comments:

In the revised version of this manuscript, the authors added data using Caspase-11 single KO mice, *Gsdmd*-supporting cell (SC)-cKO mice, AAV.ie-CMV-Gpx1 mice, and AAV.ie-SC-Gpx4 mice. These datasets are impressive and provide valuable insights for readers into the complex intercellular signaling pathways between cochlear hair cells (HCs) and supporting cells. However, I would like to suggest additional revisions on some specific points. Each numbered comment corresponds to the numbering in the authors' response. At the end, I have included a crucial comment addressing an important enigma in the signaling pathway.

3. As the authors suggest, GSDMD in non-SC cells (including HCs) may play a beneficial role based on Fig. 1h, i, k. However, in Extended Data Fig. S3h, k, GSDMD-HC-cKO mice did not exhibit higher ABR thresholds or greater OHC loss than control mice. If GSDMD in HCs does not contribute to protection against acoustic trauma, in which cochlear cell types do the authors hypothesize that GSDMD exerts its protective effect? For example, could this involve other cell components within the organ of Corti or specific populations in the lateral wall? Clarifying this point would help readers better understand the proposed cellular targets and mechanisms.

We agree with the reviewer's perspective that elucidating the functional role of GSDMD in other cochlear cell types is of significant interest. Further investigation on this front will, however, rely on technical advances—particularly the generation of cell-type-specific *Gsdmd* conditional knockout models, such as in cells located in the lateral wall. To date, there have been relatively few reports on the protective roles of GSDMD. Seminal, though limited, evidence — such as that from studies on macrophages—suggests that GSDMD may mediate protective functions.

(1). Chi, Z., Chen, S., Yang, D., Cui, W., Lu, Y., Wang, Z., Li, M., Yu, W., Zhang, J., Jiang, Y., Sun, R., Yu, Q., Hu, T.,

Lu, X., Deng, Q. et al. Gasdermin D-mediated metabolic crosstalk promotes tissue repair. *Nature* (2024).

(2). Manickam, V., Gawande, D.Y., Stothert, A.R., Clayman, A.C., Batalina, L., Warchol, M.E., Ohlemiller, K.K. & Kaur, T. Macrophages Promote Repair of Inner Hair Cell Ribbon Synapses following Noise-Induced Cochlear Synaptopathy. *J Neurosci* 43, 2075-2089 (2023).

Previous literature has demonstrated that macrophages play a protective role in noise-induced cochlear injury, leading us to hypothesize that GSDMD within macrophages may contribute to protection during acoustic trauma. Additionally, given the functional heterogeneity among macrophage subtypes in cochlear pathology, it may be necessary to generate conditional knockouts of *Gsdmd* in distinct macrophage subpopulations. We aim to systematically address this in future studies and have incorporated this perspective in Discussion section (**Line 499-506**).

6. I would like to suggest that the authors include in the revised manuscript the images of the SC layer (prepared in their response) showing that Sox2-positive SCs and 4HNE-positive cells (HCs) do not overlap. Adding these data will make the point clearer to readers and strengthen the manuscript.

We appreciate the reviewer's suggestion and have now included this figure as supplementary material in the revised manuscript (**Extended Data Fig. S5**).

7. In Fig. 6c, d, the authors demonstrate that GPX4 is downregulated in HCs at 2 hours post noise exposure, but they do not provide evidence that GPX4 expression in SCs is unchanged. The cryosection images provided in their response are not explicit on this point. I therefore suggest that the authors include representative images of the SC layer in the manuscript, in order to clearly demonstrate HC specificity in the role of GPX4.

We understand that you are seeking further clarification regarding the cell-specific role of GPX4, particularly whether it functions uniquely in HCs rather than SCs. Previous studies have demonstrated that mice with *Gpx4* knockout specifically in HCs—but not in SCs—exhibit significant hearing loss, which can be rescued by antioxidant treatment.

Liu, Z., Zhang, H., Hong, G., Bi, X., Hu, J., Zhang, T., An, Y., Guo, N., Dong, F., Xiao, Y., Li, W., Zhao, X., Chu, B., Guo, S., Zhang, X. et al. Inhibition of Gpx4-mediated ferroptosis alleviates cisplatin-induced hearing loss in C57BL/6 mice. Mol Ther 32, 1387-1406 (2024).

This supports the view that GPX4 plays an essential and specific role in HCs under physiological conditions. In line with your previous suggestion, we performed and observed that overexpression of GPX4 specifically in HCs—but not in SCs—significantly attenuates GSDMD-mediated injury signaling. Moreover, in response to your comment here, we performed additional GPX4 staining in the SC layer. Although a slight decreasing trend in GPX4 expression was observed in SCs at 2 hours after noise exposure, the difference was not statistically significant.

Taken together with previous findings and our results, we propose that GPX4 primarily functions in HCs rather than SCs, both under normal conditions and in response to noise-induced injury. We truly appreciate your insightful comments, which have been invaluable in guiding these additional analyses.

11. In the revised manuscript, the authors acknowledge the multifunctional nature of ERK in SCs but there are more key studies demonstrating ERK's protective effect on HCs. For completeness and balance, I recommend citing representative work in this area, including the study from Kyoto University (Insulin-like growth factor 1 inhibits hair cell apoptosis and promotes the cell cycle of supporting cells by activating different downstream cascades after pharmacological hair cell injury in neonatal mice. Netrin 1 mediates protective effects exerted by insulin-like growth factor 1 on cochlear hair cells.), which specifically demonstrated ERK-mediated HC protection. Including these references will provide readers with a more accurate and nuanced understanding of ERK's dual roles in cochlear SCs.

We appreciate this suggestion and thank the reviewer for pointing out these relevant studies. We have now incorporated the recommended reference (*Line 451-452* in the revised manuscript), along with other key works highlighting the protective role of ERK in hair cells. This addition provides a more balanced and comprehensive perspective on the multifaceted functions of ERK signaling in the cochlea, thereby enhancing the objectivity and quality of our manuscript.

There are several types of SCs in the organ of Corti, including pillar cells, Deiter's cells, Hensen's cells, Claudius' cells, and inner phalangeal cells. Each SC type has distinct phenotypes and roles in supporting HCs. Overall, it would be helpful if the authors could clarify which specific SC types constitute or are most crucial in the intercellular signaling pathway described in this manuscript.

We fully agree that this is an important point worthy of further investigation. Addressing this question would require the development of promoters capable of distinguishing between subtypes of supporting cells, which in turn would enable subtype-specific conditional knockout experiments. As revised in the Discussion (*Line 508-511*), this aspect deserves more systematic elucidation in future studies. Progress in this area will also benefit from broader collaborative efforts. We would greatly value your continued suggestions as we advance this research.

We thank the reviewers for their insightful and constructive feedback, which has greatly improved the quality of our manuscript and inspired the design of our future studies. For ease of reference, all changes in our response to Reviewer are highlighted in yellow.

Reviewer's Comments:

Many thanks for very carefully and fully dealing with my questions about your paper. I have made some comments below on some of your responses that you may wish to consider, and most of them do not require any changes to the changes you have made to your paper in response to my original comments. Below I have reproduced by original comments and responses and have underlined those responses where I have made further comments. You may wish to consider my further questions and comments in sections 2 and 3 They are intended for clarity and understanding of your data. Other comments may influence future experimental design.

1. To assess the hearing of the mice, the authors used two non-invasive techniques. They measured the ABR and DPOAEs. ABR measurements provide an overall measure of the hearing sensitivity of the mice in their experiments. This measure does not, on its own, enable the experimenters to discover the precise origin of any alteration in the hearing of the mice from control.

We acknowledge your valid point regarding ABR's limitations in localizing the precise origin of hearing alterations. However, ABR remains the best option for non-invasive, frequency-selective assessment of hearing sensitivity, which is the major concern to the present study. It is well recognized that NIHL impacts mainly on the function of outer hair cells (OHCs) which are responsible for the hearing sensitivity. Therefore, the location of the hearing loss is already clear and is not the focus of the present study. Furthermore, ABR is the ONLY method that can be tested repeatedly (as needed in the present study) in small rodents.

This is patently not true. The standard, and much more direct, non-invasive procedure for testing OHC activity is the measurement of DPOAEs. This technique has been used in mammals including baby microchiroptera weighing less than 5gm to new borne human babies without disturbing their sleep. DPOAEs are measured routinely and regularly to test the hearing of mice at all stages of development.

We appreciate your comment and agree with your perspective. We acknowledge that our previous statement in the last revision—that “ABR is the ONLY method that can be tested repeatedly (as required in the present study) in small rodents”—was inaccurate and regret the oversight.

While ABR cannot differentiate sensory versus neural contributions, we mitigated this limitation through complementary approaches: (1) DPOAEs specifically evaluated OHC function. (2) Cytocochleograms quantified sensory cell loss. (3) Ribbon synaptic counts assessed integrity of IHC-to-SGNs transmission. This multimodal strategy ensures ABR threshold shifts primarily reflect OHC-mediated sensitivity loss - the core focus of our NIHL investigation.

2. DPOAE measurements reflect the performance of the nonlinear amplification of the outer

hair cells. As a first point, and to enable an assessment of the quality of the data, will the authors please state in the methods the intrinsic distortion levels of the measurement system and hence its threshold for measuring acoustic distortion.

Thanks for the suggestion. We revised the method section by adding the examination of the intrinsic distortion as shown below: (Line 613-618 in the revised manuscript)

The potential intrinsic distortion and the noise floor were examined by recording the sound via the coupling cavity with the stimulation used for DPOAE. The noise floor of the testing system was below 0 dB SPL and there was no significant intrinsic distortion when tested via the physical coupling cavity utilized for the DPOAE. DPOAE threshold was defined as the interpolated value of f2 intensity required to generate a 0 dB SPL DPOAE.

Many thanks for this, but a more intuitive and direct way of presenting acoustic distortion in the sound system is to say that it is so many dB below the level of the primary tones. My guess is that for your system, it is about 80 dB. This means, for example, that there could be 10 dB SPL of distortion in tones presented at 90 dB SPL. Can you please express distortion levels in this format.

Thanks a lot for this consideration. We think that we should call the relevant measurement as “noise floor” rather than distortion. We refined the statement in manuscript for “noise floor” as you suggested (**Line 615-619 in the revised manuscript**):

The potential intrinsic distortion and the noise floor were examined by recording the sound in the coupler to which the primary tones were presented at the maximal level (80 dB SPL) for DPOAE test. The noise floor across the whole frequency range and around the targeted frequency for DPOAE was more than 80 dB below the level of the primary tones.

3. Figures 1e, h. What is “dB” relative to? Is it a value common to all measurements such as SPL? Or is it relative to each measurement? Ideally it should be referred to a commonly accepted standard, like SPL. It is easier to understand. Could you please, therefore, clarify what is meant by “dB” and ideally, for increased clarity and understanding, convert to SPL.

Figure 1e and h concern level shift, which is relative to baseline values, they should not be relative to a standard like SPL but relative to each other, since a shift would be the same in dB no matter whether SPL or other metric is used.

What baseline value were you using? Was it the threshold stimulus tone SPL at each particular frequency? For example, did you have to use tones close to 100 dB SPL at 32 kHz to elicit a signal from Gsdmd KO at 1 day and ~ 70 dB SPL at day 14? the absolute SPL you use is important because it shows if the stimuli fall within the linear dynamic range of your sound system.

The baseline values refer to the ABR thresholds measured prior to noise exposure. All ABR thresholds are expressed in dB SPL, where SPL is referenced to 20 μ Pa for the sound measurement in air. Threshold shifts represent the difference in dB between pre- and post-exposure thresholds. We think the reason is that the threshold uses 20 μ Pa as references,

while the threshold shifts use the baseline as the reference. If the baseline ABR threshold at a frequency is 30 dB SPL and increases to 90 dB SPL after noise exposure, the threshold shift is 60 dB, which is the difference between the two measurements.

An analogy may help illustrate our perspective: consider the height difference between two people on separate floors of a building. Whether one is on the 10th and the other on the 11th floor, or one on the 30th and the other on the 31st floor, their difference in height remains one floor. This difference is independent of their absolute height above the ground—which, in this context, may be compared to the 0 dB SPL reference.

We hope this clarification aligns with the reviewer's concern. Besides, we also acknowledge that both "dB" and "dB SPL" have been used in the previous literature to report threshold shifts, which may reflect differing conventions rather than a strictly resolved nomenclature issue.

4. In your methods you state:

"Briefly, acoustic stimuli were presented via custom-designed plastic tubing connected to an MF1 Multi-Field Magnetic Speaker (TDT, USA) positioned in the ear canal. Puretone stimuli of 10- ms (0.5- ms rise/fall) were presented at 21.1/s across intended frequencies, starting from 90 dB SPL and decreasing in steps of 5 dB. Evoked responses were amplified 10,000× (PA4 bio-amplifier, TDT) and averaged 500 times with 0.3 to 3 kHz band-pass 565 filter." Thus, at the very beginning of your measurements you presented 500 intense tone bursts. The higher frequency regions of the mouse cochlea (above about 30 kHz) are very noise sensitive, and your regime could prejudice the sensitivity of the cochlea. In the methods, or in the results section, please explain why you chose to deliver the tones descending from a high to low levels instead of ascending from low to high levels, or perhaps with a degree of randomization in the presentation of both frequencies and sound levels, which would tend to preserve hearing sensitivity.

Thanks for your comments. The reason to start testing sequence from high sound level (90 dB SPL) is to make sure that the mouse has a good ABR response. This approach reduces uncertainty due to any wrong setting for the test. The impact of high-level stimulation on hearing sensitivity is unlikely based on several facts. First, we do not need to repeat 500 times in each trial at high sound level(s), as less as 100-200 repeats is usually good enough to obtain clear responses. Secondly, we do not really do 5-dB steps at high sound level but drop by 10 or even 20 dB. 5-dB steps are used when the test approaches the threshold. In this way, the exact noise exposure is trivial. Thirdly, the tone burst exposure at 90 dB SPL with the rate used in our testing does not significantly change the ABR threshold even up to thousands of repeats. Ideally, randomization in level should minimize the potential impact of high-level tone exposure on thresholds. But our equipment does not allow such a randomization. However, we are confident that the method we use is safe. In addition, we stuck the method in the same way across all the tests. Therefore, the results should reasonably reflect the real change in hearing sensitivity. Finally, the down-sequencing in ABR test is generally acceptable. To address whether our descending-level protocol (90 → lower dB SPL) affects cochlear sensitivity, we performed direct comparisons in a cohort of mice (n = 6) using:

- (1) Low-to-high intensity ascending protocol
- (2) Our high-to-low intensity descending protocol

on the same ears, with all frequencies tested at 500 repetitions per stimulus level.

We found no significant differences in ABR thresholds at any frequency (8, 16, 32kHz) and in ABR wave-I amplitude at different sound level (60, 70, 80 and 90 dB SPL).

According to the data you presented, this does not appear to be the case. This finding is to be expected, according to recent papers e.g <https://doi.org/10.1523/ENEURO.0250-18.2018>. In the quoted paper, the noise level was 100 dB SPL, which is 10 dB SPL higher than the loudest test tones used in your experiments. All four ABR peaks were reduced in size by the noise exposure, but peaks 2 to 4 were sensitized. Peak 1 arises directly from the response of the auditory nerve. Peaks 2 to 4 arise from excitatory, inhibitory, interaction between neurons in the brainstem. There is evidence of this in your figure, especially in response to 16 kHz and 32 kHz tones. You say in the paper: "Thresholds were defined as the lowest stimulus level that produced two or more discernible ABR waveforms (waves 1 to 5)." Measurements should have been based on the responses of the first peak of the ABR, if the measurements are meant to most closely reflect the responses of the OHCs. Later peaks reflect brainstem responses that can be modified by neuronal interaction. These comments pertain to your choice of measurement, which you found convenient, but DPOAE measurement, which only involve carefully placing a speculum in the auditory meatus of a lightly anaesthetised mouse, provides a more direct measure of OHC responses.

Thank you for raising this insightful point. We appreciate your detailed explanation and the relevant reference, which have drawn our attention to important methodological details we had previously overlooked.

These data confirmed our stimulus paradigm does not compromise ABR testing sensitivity. While our descending-level protocol does not affect standard ABR metrics (thresholds/amplitude), we acknowledge that subtler impacts on hearing sensitivity—potentially detectable through compound action potential or other electrophysiological measures. This methodological consideration provides valuable guidance for our future experimental design where ultra-high-frequency sensitivity preservation is paramount.

This is not really the case. Compound AP, or single fiber, measurements are invasive and indirectly signal the responses of the afferent fibres. CM measurements can be confusing and provide misleading information, especially in noise studies. See: J Neurophysiol 127: 1574–1585, 2022. doi:10.1152/jn.00402.2021.

We sincerely appreciate you sharing your deep understanding and perspective in the field of auditory electrophysiology. Thank you for providing the relevant reference as well.

Here, we revised one sentence in the method section to address the sequencing: (Line 601-603)

At each frequency, the test started from 90 dB SPL to ensure a clear response; and stepped down in 10 dB steps at high levels and then 5 dB when approaching the threshold.

I would keep this revised sentence. My comments are meant for consideration in designing further experiments and improvements in future experimental techniques.

We fully recognize that standardized auditory electrophysiological testing is essential for obtaining high-quality experimental data. We greatly appreciate your valuable input and suggestions, which will undoubtedly help guide the design of our future experiments and the refinement of our technical approaches.

5. Figure 1i. In the methods you state:

“In DPOAEs testing, the primary tones f_1 and f_2 were delivered through two MF1 loudspeakers, respectively, positioned in a closed-field arrangement. A custom designed probe equipped with a low-noise ER10B+ microphone (TDT, USA) facilitated precise stimulus presentation and recording. DPOAEs were measured with a fixed f_2/f_1 ratio of 1.2, where the intensity of f_2 (L_2) was set to 10 dB lower than f_1 (L_1). Frequencies ranged from 5.6 to 32 kHz in half-octave intervals. For each f_2 , L_2 was varied from 70 to 10 dB SPL in 5 dB steps. The $2f_1-f_2$ DPOAE amplitude and the surrounding noise floor were extracted for offline analysis.” In the caption to Fig1i: “i. DPOAE amplitudes were higher in Gsdmd-SC-cKO mice compared to Gsdmdflox/flox mice at 14 d-PNE (solid line), with no significant differences at Prenoise exposure (dashed line).” Again it seems you began your measurements of DPOAE at high levels and descended the SPL to low levels, which may have prejudiced the sensitivity of the cochlea. It may have been helpful to design a protocol that extracted the DPOAEs continuously during the measurement. Such a protocol would enable you to monitor the DPOAEs directly and to store them and the raw data. This would have provided you with an immediate visual display of the sensitivity of the cochlea. Can you please explain why you chose your approach and not ones that enabled you to monitor DPOAE directly and to start from low SPLs to high, to preserve cochlear sensitivity.

Thanks for the comments and suggestions. Again, we start our DPOAE from high sound level to ensure that we have a good result, which can be jeopardized by wrong insertion of earpiece and other actions in setting up. Do DPOAE in upward sequencing would end up with failure thereafter and waste more time, which is limited by the short effective time of the anesthesia. The anesthesia we choice is the best in terms of safety, but the appropriate anesthetics can be maintained for less than one hour with the light anesthesia protocol. This approach is taken due to the concern of losing animals.

Considering the measurements of both ABR and DPOAE, we must do everything quickly. In addition, the highest sound level for DPOAE is 70-80 dB SPL, noise exposure at level is generally safe, there is not data supporting a significant HL by noise exposure at such a low level, especially for such a short duration. Regarding the continuous monitoring of DPOAE, we don't really understand the point of the reviewer in terms of "immediate visual display of the sensitivity of the cochlea". We guess that the reviewer is concerned about the change of cochlear sensitivity due to the manipulation of the DPOAE test. The first reason for not taking the suggested approach is that our system does not allow us to do a continuous recording with an immediate visual display.

Our lab also present DPOAE data in the frequency domain, but we also have severe time restraints on our data collection and collect only 10s, rather than 100s of samples at each frequency. We thus get data that can be displayed very quickly. This technique might be further improved by a quasi-randomised technique where sampling could be stopped at particular frequencies and levels where the sampling has met a pre-determined signal to noise criteria. We have yet to implement such a technique.

A more intelligent strategy indeed appears highly beneficial for improving the quality of DPOAE data collection and accelerating experimental procedures, while also helping to preserve sensitivity of the cochlea.

DPOAE results are shown in frequency domain, while recording is the temporal waveform. Converting the signal in real time to frequency domain is very challenging. Therefore, we record the waveform in time domain for each frequency and level long enough for averaging and converting to frequency domain signal, which is saved. Moreover, continuous monitoring requires stimulation to be fixed in level and frequency, while a comprehensive evaluation on DPOAE requires it to be tested across a wide range of frequency and level. In conclusion, we set up our DPOAE protocol with careful planning and tried different ways. We are pleased to validate your proposed approach in follow-up studies. Should it demonstrate superior protection of auditory sensitivity, we will refine our current experimental protocol accordingly. We sincerely appreciate your rigorous perspective, which strengthens our methodology.

I have no comments on your responses from and including 6 onwards

Thank you for your insightful points.

Reviewer #1

We thank the reviewers for their thorough and constructive feedback. Addressing these comments has significantly strengthened our study. In revising the manuscript, we have conducted additional experiments and comprehensively addressed all points through careful analysis of available data.

The paper: ROS-GSDMD feedback loop between hair cells and supporting cells governs noise-induced hearing loss reports some potentially important findings.

Namely, they reveal a paradigm-shifting mechanism where supporting cells (SCs) orchestrate NIHL through Gasdermin D (GSDMD) with a non-canonical pathway.

Conditional knockout of GSDMD in SCs, but not in HCs, significantly reduces NIHL.

My main expertise is in cochlear physiology, and I have focussed largely on the physiological measurements reported in the paper. The physiological findings are important because they underpin the major findings reported in the paper.

1. To assess the hearing of the mice, the authors used two non-invasive techniques. They measured the ABR and DPOAEs. ABR measurements provide an overall measure of the hearing sensitivity of the mice in their experiments. This measure does not, on its own, enable the experimenters to discover the precise origin of any alteration in the hearing of the mice from control.

We acknowledge your valid point regarding ABR's limitations in localizing the precise origin of hearing alterations. However, ABR remains the best option for non-invasive, frequency-selective assessment of hearing sensitivity, which is the major concern to the present study. It is well recognized that NIHL impacts mainly on the function of outer hair cells (OHCs) which are responsible for the hearing sensitivity. Therefore, the location of the hearing loss is already clear and is not the focus of the present study.

Furthermore, ABR is the ONLY method that can be tested repeatedly (as needed in the present study) in small rodents.

While ABR cannot differentiate sensory versus neural contributions, we mitigated this limitation through complementary approaches: (1) DPOAEs specifically evaluated OHC function. (2) Cytocochleograms quantified sensory cell loss. (3) Ribbon synaptic counts assessed integrity of IHC-to-SGNs transmission. This multimodal strategy ensures ABR threshold shifts primarily reflect OHC-mediated sensitivity loss - the core focus of our NIHL investigation.

2. DPOAE measurements reflect the performance of the nonlinear amplification of the outer hair cells. As a first point, and to enable an assessment of the quality of the data, will the authors please state in the methods the intrinsic distortion levels of the measurement system and hence its threshold for measuring acoustic distortion.

Thanks for the suggestion. We revised the method section by adding the examination of the intrinsic distortion as shown below: (**Line 613-618** in the revised manuscript)

Vj g"r qvgnl'kpm kpuke" f kwqt vkqp" cpf "vj g"pqkug"l'iqqt" y gt g"gzco kpgf "d{ "tgeqtf kpi "vj g" uqwpf "xlc"vj g"eqwr r'kpi "ecxk\ "y kj "vj g"l'wko wcvkqp" wugf "lqt"FRQCGO"vj g"pqkug"l'iqqt" qh' vj g"vgu'kpi "l'wgo "y cu'dgrqy "2'f D'URN" cpf "vj gt g"y cu'pq'uki p'k'ecpv'kpm kpuke" f kwqt vkqp" y.j gp"vgwgf "xlc"vj g"rj { ukecrl'eqwr r'kpi "ecxk\ "w'k'k\ gf "lqt"vj g"FRQCGO"FRQCG"vj t guj qrf" y cu'f g'kpgf "cu"vj g"kvgt r q'cvgf "xcw'g"q'hl'h\ "k'p'v'g'p'uk\ "t'gs w'k'gf "vq"i gp'gt cv'g" c "2'f D'URN" FRQCGO

3. Figures 1e, h. What is dB relative to? Is it a value common to all measurements such as SPL? Or is it relative to each measurement? Ideally it should be referred to a commonly accepted standard, like SPL. It is easier to understand. Could you please, therefor, clarify what is meant by dB and ideally, for increased clarity and understanding, convert to SPL.

Figure 1e and h concern level shift, which is relative to baseline values, they should not be relative to a standard like SPL but relative to each other, since a shift would be the same in dB no matter whether SPL or other metric is used. Please note that dB SPL is used in d and g where the absolute thresholds are addressed using SPL. In such cases, a standard value (20 uPa in air) is used as reference.

4. In your methods you state:

Briefly, acoustic stimuli were presented via custom-designed plastic tubing connected

to an MF1 Multi-Field Magnetic Speaker (TDT, USA) positioned in the ear canal. Pure-tone stimuli of 10- ms (0.5- ms rise/fall) were presented at 21.1/s across intended frequencies, starting from 90 dB SPL and decreasing in steps of 5 dB. Evoked responses were amplified 10,000 (PA4 bio-amplifier, TDT) and averaged 500 times with 0.3 to 3 kHz band-pass 565 filter.

Thus, at the very beginning of your measurements you presented 500 intense tone bursts. The higher frequency regions of the mouse cochlea (above about 30 kHz) are very noise sensitive, and your regime could prejudice the sensitivity of the cochlea. In the methods, or in the results section, please explain why you chose to deliver the tones descending from a high to low levels instead of ascending from low to high levels, or perhaps with a degree of randomization in the presentation of both frequencies and sound levels, which would tend to preserve hearing sensitivity.

Thanks for your comments. The reason to start testing sequence from high sound level (90 dB SPL) is to make sure that the mouse has a good ABR response. This approach reduces uncertainty due to any wrong setting for the test. The impact of high-level stimulation on hearing sensitivity is unlikely based on several facts. First, we do not need to repeat 500 times in each trial at high sound level(s), as less as 100-200 repeats is usually good enough to obtain clear responses. Secondly, we do not really do 5-dB steps at high sound level but drop by 10 or even 20 dB. 5-dB steps are used when the test approaches the threshold. In this way, the exact noise exposure is trivial. Thirdly, the tone burst exposure at 90 dB SPL with the rate used in our testing does not significantly change the ABR threshold even up to thousands of repeats. Ideally, randomization in level should minimize the potential impact of high-level tone exposure on thresholds. But our equipment does not allow such a randomization. However, we are confident that the method we use is safe. In addition, we stuck the method in the same way across all the tests. Therefore, the results should reasonably reflect the real change in hearing sensitivity. Finally, the down-sequencing in ABR test is generally acceptable.

To address whether our descending-level protocol (90 lower dB SPL) affects cochlear sensitivity, we performed direct comparisons in a cohort of mice (n = 6) using:

- (1) Low-to-high intensity ascending protocol
- (2) Our high-to-low intensity descending protocol

on the same ears, with all frequencies tested at 500 repetitions per stimulus level.

We found no significant differences in ABR thresholds at any frequency (8, 16, 32kHz) and in ABR wave-I amplitude at different sound level (60, 70, 80 and 90 dB SPL).” These data confirmed our stimulus paradigm does not compromise ABR testing sensitivity. While our descending-level protocol does not affect standard ABR metrics (thresholds/amplitude), we acknowledge that subtler impacts on hearing sensitivity potentially detectable through compound action potential or other electrophysiological measures. This methodological consideration provides valuable guidance for our future experimental design where ultra-high-frequency sensitivity preservation is paramount.

Here, we revised one sentence in the method section to address the sequencing: (**Line 601-603**)

Cv'gcej "lts wgep. "vj g"vgiw'wctvgf "lto "; 2"fD"URN"vq"gpwtg"c"erect "tgur qpug="cpf" wgrrgf "f qy p"kp"32"fD"wgru"cv"j k j "ngxgu"cpf "vj gp"7"fD"y j gp"cr rtqcej kpi "vj g" vj tguj qrf 0

5. Figure 1i. In the methods you state:

In DPOAEs testing, the primary tones f1 and f2 were delivered through two MF1 loudspeakers, respectively, positioned in a closed-field arrangement. A custom-designed probe equipped with a low-noise ER10B+ microphone (TDT, USA) facilitated precise stimulus presentation and recording. DPOAEs were measured with a fixed f2/f1 ratio of 1.2, where the intensity of f2 (L2) was set to 10 dB lower than f1 (L1). Frequencies ranged from 5.6 to 32 kHz in half-octave intervals. For each f2, L2 was varied from 70 to 10 dB SPL in 5 dB steps. The 2f1-f2 DPOAE amplitude and the surrounding noise floor were extracted for offline analysis.

In the caption to Fig 1i:

i. DPOAE amplitudes were higher in Gsdmd-SC-cKO mice compared to Gsdmdflox/flox mice at 14 d-PNE (solid line), with no significant differences at Pre-noise exposure (dashed line).

Again it seems you began your measurements of DPOAE at high levels and descended the SPL to low levels, which may have prejudiced the sensitivity of the cochlea. It may have been helpful to design a protocol that extracted the DPOAEs continuously during the measurement. Such a protocol would enable you to monitor the DPOAES directly and to store them and the raw data. This would have provided you with an immediate visual display of the sensitivity of the cochlea. Can you please explain why you chose your approach and not ones that enabled you to monitor DPOAE directly and to start from low SPLs to high, to preserve cochlear sensitivity.

Thanks for the comments and suggestions. Again, we start our DPOAE from high sound level to ensure that we have a good result, which can be jeopardized by wrong insertion of earpiece and other actions in setting up. Do DPOAE in upward sequencing would end up with failure thereafter and waste more time, which is limited by the short effective time of the anesthesia. The anesthesia we choice is the best in terms of safety, but the appropriate anesthetics can be maintained for less than one hour with the light anesthesia protocol. This approach is taken due to the concern of losing animals. Considering the measurements of both ABR and DPOAE, we must do everything quickly. In addition, the highest sound level for DPOAE is 70-80 dB SPL, noise exposure at level is generally safe, there is not data supporting a significant HL by noise exposure at such a low level, especially for such a short duration.

Regarding the continuous monitoring of DPOAE, we don t really understand the point of the reviewer in terms of immediate visual display of the sensitivity of the cochlea .

We guess that the reviewer is concerned about the change of cochlear sensitivity due to the manipulation of the DPOAE test. The first reason for not taking the suggested approach is that our system does not allow us to do a continuous recording with an immediate visual display. DPOAE results are shown in frequency domain, while recording is the temporal waveform. Converting the signal in real time to frequency domain is very challenging. Therefore, we record the waveform in time domain for each frequency and level long enough for averaging and converting to frequency domain signal, which is saved. Moreover, continuous monitoring requires stimulation to be fixed in level and frequency, while a comprehensive evaluation on DPOAE requires it to be tested across a wide range of frequency and level. In conclusion, we set up our DPOAE protocol with careful planning and tried different ways. We are pleased to validate your proposed approach in follow-up studies. Should it demonstrate superior protection of auditory sensitivity, we will refine our current experimental protocol accordingly. We sincerely appreciate your rigorous perspective, which strengthens our methodology.

6. Also, please state the SPL at which you measured the DPOAEs. This does not appear to be given in the caption of Figure 1. Additionally, please define dBV and why DPOAE magnitude is not given as SPL.

We thank the reviewer for identifying these important issues. Regarding Fig. 1:

1 The DPOAE data were acquired in down sequence starting at 80 dB SPL F1 stimulation

2 The unit was incorrectly labeled as dBV due to a documentation error. Per TDT's DPOAE conversion protocol (<https://www.tdt.com/docs/dpoe-user-guide/converting-dbv-to-db-spl>), values should be presented in dB SPL.

We further acknowledge your valuable suggestion about DPOAE amplitude interpretation. Given the nonlinear response pattern (including plateau effects) of DPOAE responses across intensity levels, we agree that single-intensity amplitude measurements risk ambiguous interpretation. Accordingly, we have replaced the DPOAE amplitude-frequency curves (responses at L2 = 70 dB SPL) with frequency-specific DPOAE thresholds in the revised figures (**Fig. 1i, 5g, 5h**). This approach provides a more physiologically meaningful characterization of OHC function.

7. I strongly suggest that the presentation of DPOAE ISO response curves would have given a clearer understanding of the frequency dependence of DPOAE generation in the cochlea under different conditions. Amplitude can give a confusing impression because, especially at high levels, e.g F1 at 50 dB SPL, DPOAE increase with level is very nonlinear and can reach a plateau, depending also on stimulus frequency.

Thanks. We assume that the ISO response refers to the level-frequency curve to show that, at each frequency, what sound level can evoke a target DPOAE amplitude across frequencies. We think that revised threshold-frequency curve serves this purpose since the threshold is determined by a certain amount of response amplitude above the noise floor (**Fig. 1i, 5g, 5h**). We revised the method section by adding the definition of the DPOAE threshold in **Line 617-618** of the revised manuscript and copied here:

FRQCG"vj t g u j q r f "y c u 'f g h k p g f "c u 'v j g "k p v g t r q r c v g f "x c n w g" q l 'h "k p v g p u k y { "t g s w k t g f "v q" i g p g t c v g 'c '2 f D'URNFRQCG0

8. Figure 1k. There appears to be no OHC loss for frequencies below about the 3.5 mm location, from apex. According to a well-known mouse cochlear frequency map (Miller et al., Hearing Research 202 (2005) 63-73), this is close to the 34 kHz frequency place. Do these findings, in part, account for why there appears to be no complete recovery from NIHL?

To our knowledge, it is common to see discrepancy between functional deterioration and OHC loss in the study of NIHL. The threshold elevation is better matched by OHC loss in very high frequency region but in low frequency region, we often see threshold shift without any OHC loss. It is concluded that OHCs may have been functionally damaged but keep morphological integrity as shown by the most common methods for HC count.

(see Chen G D, Fechter L D. The relationship between noise-induced hearing loss and hair cell loss in rats. Hear Res, 2003, 177(1-2): 81-90 as an exemplary research).

Our result showed also the discrepancy at the low frequency region below 34 kHz, suggesting incomplete recovery of hearing sensitivity.

9. Figure 1m. Please state that this figure is based on sampling in the 32 kHz region where OHC loss is expected to be minimal, or indeed absent.

The sampling location for synapse investigation is stated in the figure legend of 1l as 32 kHz region. 1m is the statistical evaluation of sample in 1l. To make it clearer, we revised the figure legend of 1m as follow (**Line 154-156**):

Uc vkakcu'ql'uco r rgu'uj qy p"kp"l"lt qo "54"nJ | "t gi kqp"y j gt g"QJ E"rqui"ku"o kpkø cn"qt "
*kpf ggf "cdugpv."uj qy kpi "pq"uc vkakcu'f kktgpegu"kp"u{pcrug"fg puka} "*c xgt ci gf "lt qo "*
gki j v'KJ Eu'r gt 'eqej rgc +c etquu'i tqwr uO'

10. A remarkable finding, not commented on in the paper, is that there is no hair cell loss at the location of the noise exposure or at a location a half octave above this where

noise is most effective at desensitizing the cochlea (Cody and Johnstone, 1981 J. Acoust. Sot. Am. 70, 707-711; Cody and Russell, 1985, Nature 315, 662-665). Instead, hair cell loss is confined to frequencies above about 34 kHz, a region where the cochlea is most sensitive noise insult. Thus presentation of an 8-16 kHz noise band causes no measurable change in sensitivity in the cochlea region apical to this, it causes loss of sensitivity to all frequency regions at and basal to the 8-16 kHz band, and causes 0% - 20% OHC loss at frequencies a half octave above the 8-16 kHz band and up to 100% OHC loss at frequencies above this (in the 40 kHz - 90 kHz region). Could it be that there are differences in the action of Gasdermin D on NIHL at the bases and hook region of the cochlea from its action in the more apical regions? Perhaps this point could be raised in the discussion.

The famous half-octave law in NIHL is not established on HC loss. The study by Cody and Johnstone (JASA 1981) was on TTS of single unit ANF. No data on OHC loss was seen in this paper. This is also true for the study by Cody and Russel 1985 (Cody A R, Russell I J. Outer hair cells in the mammalian cochlea and noise-induced hearing loss. Nature, 1985, 315: 662-665.) in which no HC count data was available. As we responded above, there is a big discrepancy between threshold shifts and OHC loss in the low frequency region after NIHL was established: often time, the threshold shift is NOT matched by the amount of OHC loss.

Using tone of middle frequency (8-16) is a common practice in the studies of NIHL in mice

*30 Hwt o cp "C" E. "Mwcy c" UI . "Nkdgt o cp" O "EOP qkug/kpf wegf "eqej rgt "pgwt qrcyj { "ku'ugrgevkg" lqt " hldgtu" y kj " rjy " urqpxpgqwu" tcvgu0' " L" P gwtqrj {ukqn" 4235." 332*5+<' 799/7: 80' F QK" 3208374 Ilp0238604235

40 Mwcy c "U" I . "Nkdgt o cp" O "E0' Cff kpi "kpuwn" \vq" kplwt { <'eqej rgt "pgt.xg" f gi gpgt cvkqp" chgt " \$go rqt ct { \$" p qkug/kpf wegf " j gct kpi " rqu0' " L" P gwt quek " 422; ." 4; *67+<' 36299/362: 70' F QK" 4; k67B6299"3208745 ILPGWTQUEK#: 67/2; 022;

50 Ockuqp "UH" Nkdgt o cp" O "E0Rt gf kevpi 'xwpgt cdkrk { 'vq' c eqwnte' kplwt { 'y kj 'c' p qkpxcukxg' cuuc { "

qllqrlxqeqej rgct 'tghgz'wtgpi vj 0''LP gwtquek '4222.'42*34+<6923/69290+

Exposure at such frequency can cause NIHL spread to almost the whole cochlea, more towards the high-frequency end. With the intensity and duration used in the present experiment, OHC loss is expected to be limited to the very high-frequency region (particularly in the hook area), and not in the region half an octave above the frequency region of acoustic overstimulation.

The question Could it be that there are differences in the action of Gasdermin D on NIHL at the bases and hook region of the cochlea from its action in the more apical regions? is difficult to answer since we do not have evidence to support or reject the speculation. The important fact to our study is the protective effect against noise-induced OHC loss appears and can only be demonstrated in the region where OHC loss is seen. Recent literature demonstrates that noise exposure induces a gradient increase in NOX2-positive OHCs from apical to basal turns, with NOX2 deficiency partially attenuating cochlear damage. This suggests basal OHCs experience higher oxidative stress, potentially predisposing this region to initiate the damaging ROS-GSDMD positive feedback loop.

Sk'OOI cq.\ 0'S kw.[O'Y cpi .T0'Vkp.'M0'[wg.'D0'\j cpi .Z0'\j cpi .R0'Y w.\ 0'\j w.'S 0'Nkw.\ 0' Oc.\ 0'\j qw.'Z0'J cp.[O'Ej gp.'LO'gv'crl0PQZ4'Eqvtdkwgu'\q'J ki j -Frequency Outer Hair Cell Vulnerability in the Cochlea. Advanced Science (2025).

However, due to the current lack of a specific antibody reliably detecting the GSDMD-N fragment, it is currently unfeasible to definitively determine via immunolabeling whether noise exposure indeed triggers a higher gradient of GSDMD activation specifically within the basal turn. We have revised the Discussion about this limitation: **(Line 476-480, New data in Extended Data Fig. S11)**

However, spatial mapping of GSDMD activation remains technically constrained: Commercially available GSDMD-N antibodies showed non-specific binding in cochlear tissue and failed validation using Gsdmd-KO controls, precluding reliable immunolabeling of GSDMD-N fragments in HCs or SCs.

We thank the reviewers for their insightful and constructive feedback, which has greatly improved the quality of our manuscript and inspired the design of our future studies. For ease of reference, all changes in our response to Reviewer#1 are highlighted in yellow.

Reviewer #1 (Remarks to the Author):

Many thanks for very carefully and fully dealing with my questions about your paper. I have made some comments below on some of your responses that you may wish to consider, and most of them do not require any changes to the changes you have made to your paper in response to my original comments. Below I have reproduced by original comments and responses and have underlined those responses where I have made further comments. You may wish to consider my further questions and comments in sections 2 and 3 They are intended for clarity and understanding of your data. Other comments may influence future experimental design.

1. To assess the hearing of the mice, the authors used two non-invasive techniques. They measured the ABR and DPOAEs. ABR measurements provide an overall measure of the hearing sensitivity of the mice in their experiments. This measure does not, on its own, enable the experimenters to discover the precise origin of any alteration in the hearing of the mice from control.

We acknowledge your valid point regarding ABR's limitations in localizing the precise origin of hearing alterations. However, ABR remains the best option for non-invasive, frequency-selective assessment of hearing sensitivity, which is the major concern to the present study. It is well recognized that NIHL impacts mainly on the function of outer hair cells (OHCs) which are responsible for the hearing sensitivity. Therefore, the location of the hearing loss is already clear and is not the focus of the present study. Furthermore, ABR is the ONLY method that can be tested repeatedly (as needed in the present study) in small rodents.

This is patently not true. The standard, and much more direct, non-invasive procedure for testing OHC activity is the measurement of DPOAEs. This technique has been used in mammals including baby microchiroptera weighing less than 5gm to new borne human

babies without disturbing their sleep. DPOAEs are measured routinely and regularly to test the hearing of mice at all stages of development.

We appreciate your comment and agree with your perspective. We acknowledge that our previous statement in the last revision was inaccurate and regret the oversight.

While ABR cannot differentiate sensory versus neural contributions, we mitigated this limitation through complementary approaches: (1) DPOAEs specifically evaluated OHC function. (2) Cytocochleograms quantified sensory cell loss. (3) Ribbon synaptic counts assessed integrity of IHC-to-SGNs transmission. This multimodal strategy ensures ABR threshold shifts primarily reflect OHC-mediated sensitivity loss - the core focus of our NIHL investigation.

2. DPOAE measurements reflect the performance of the nonlinear amplification of the outer hair cells. As a first point, and to enable an assessment of the quality of the data, will the authors please state in the methods the intrinsic distortion levels of the measurement system and hence its threshold for measuring acoustic distortion.

Thanks for the suggestion. We revised the method section by adding the examination of the intrinsic distortion as shown below: (Line 613-618 in the revised manuscript)

The potential intrinsic distortion and the noise floor were examined by recording the sound via the coupling cavity with the stimulation used for DPOAE. The noise floor of the testing system was below 0 dB SPL and there was no significant intrinsic distortion when tested via the physical coupling cavity utilized for the DPOAE. DPOAE threshold was defined as the interpolated value of f2 intensity required to generate a 0 dB SPL DPOAE.

Many thanks for this, but a more intuitive and direct way of presenting acoustic distortion in the sound system is to say that it is so many dB below the level of the primary tones. My guess is that for your system, it is about 80 dB. This means, for example, that there could be 10 dB SPL of distortion in tones presented at 90 dB SPL. Can you please express distortion levels in this format.

Thanks a lot for this consideration. We think that we should call the relevant measurement as We refined the statement in manuscript for noise floor as you suggested (**Line 615-619 in the revised manuscript**) :

The potential intrinsic distortion and the noise floor were examined by recording the sound in the coupler to which the primary tones were presented at the maximal level (80 dB SPL) for DPOAE test. The noise floor across the whole frequency range and around the targeted frequency for DPOAE was more than 80 dB below the level of the primary tones.

such as SPL? Or is it relative to each measurement? Ideally it should be referred to a commonly

accepted standard, like SPL. It is easier to understand. Could you please, therefor, clarify what understanding, convert to SPL.

Figure 1e and h concern level shift, which is relative to baseline values, they should not be relative to a standard like SPL but relative to each other, since a shift would be the same in dB no matter whether SPL or other metric is used.

What baseline value were you using? Was it the threshold stimulus tone SPL at each particular frequency? For example, did you have to use tones close to 100 dB SPL at 32 kHz to elicit a signal from Gsdmd KO at 1 day and ~ 70 dB SPL at day 14? the absolute SPL you use is important because it shows if the stimuli fall within the linear dynamic range of your sound system.

The baseline values refer to the ABR thresholds measured prior to noise exposure. All ABR for the sound measurement in air. Threshold shifts represent the difference in dB between pre- and post-exposure thresholds and should not be quantified as dB SPL. This is due to the fact that the threshold uses 20 Pa as references, while the threshold shifts use the baseline as the reference. If the baseline ABR threshold at a frequency is 30 dB SPL and increases to 90 dB SPL after noise exposure, the threshold shift is 60 dB, which is the difference between the two measurements. When talking about the shift, the reference for 0 dB (in SPL system) does not matter. This is similar to the height difference between two people who stand one floor apart. In one scenario, they are in 10th and 11th floor separately, the difference is one floor. In the other scenario, they are in 30th and 31st floor. Their difference is still one floor apart. Therefore, the difference is NOT related to how high they are from the ground (which could be considered as 0 dB SPL).

However, we report threshold shifts, which may reflect differing conventions rather than a strictly resolved nomenclature issue.

4. In your methods you state:

-designed plastic tubing connected to an MF1 Multi-Field Magnetic Speaker (TDT, USA) positioned in the ear canal. Puretone stimuli of 10- ms (0.5- ms rise/fall) were presented at 21.1/s across intended frequencies, starting from 90 dB SPL and decreasing in steps of 5 dB. Evoked responses bio-amplifier, TDT) and averaged 500 times with 0.3 to 3 kHz band- Thus, at the very beginning of your measurements you presented 500 intense tone bursts. The higher frequency regions of the mouse cochlea (above about 30 kHz) are very noise sensitive, and your regime could prejudice the sensitivity of the cochlea. In the methods, or in the results section, please explain why you chose to deliver the tones descending from a high to low levels instead of ascending from low to high levels, or perhaps with a degree of randomization in the presentation of both frequencies and sound levels, which would tend to preserve hearing sensitivity.

Thanks for your comments. The reason to start testing sequence from high sound level (90 dB SPL) is to make sure that the mouse has a good ABR response. This approach reduces

uncertainty due to any wrong setting for the test. The impact of high-level stimulation on hearing sensitivity is unlikely based on several facts. First, we do not need to repeat 500 times in each trial at high sound level(s), as less as 100-200 repeats is usually good enough to obtain clear responses. Secondly, we do not really do 5-dB steps at high sound level but drop by 10 or even 20 dB. 5-dB steps are used when the test approaches the threshold. In this way, the exact noise exposure is trivial. Thirdly, the tone burst exposure at 90 dB SPL with the rate used in our testing does not significantly change the ABR threshold even up to thousands of repeats. Ideally, randomization in level should minimize the potential impact of high-level tone exposure on thresholds. But our equipment does not allow such a randomization. However, we are confident that the method we use is safe. In addition, we stuck the method in the same way across all the tests. Therefore, the results should reasonably reflect the real change in hearing sensitivity. Finally, the down-sequencing in ABR test is generally acceptable. To address whether our descending-

cochlear sensitivity, we performed direct comparisons in a cohort of mice (n = 6) using:

- (1) Low-to-high intensity ascending protocol
- (2) Our high-to-low intensity descending protocol

on the same ears, with all frequencies tested at 500 repetitions per stimulus level.

We found no significant differences in ABR thresholds at any frequency (8, 16, 32kHz) and in ABR wave-I amplitude at different sound level (60, 70, 80 and 90 dB SPL).

According to the data you presented, this does not appear to be the case. This finding is to be expected, according to recent papers e.g <https://doi.org/10.1523/ENEURO.0250-18.2018>. In the quoted paper, the noise level was 100 dB SPL, which is 10 dB SPL higher than the loudest test tones used in your experiments. All four ABR peaks were reduced in size by the noise exposure, but peaks 2 to 4 were sensitized. Peak 1 arises directly from the response of the auditory nerve. Peaks 2 to 4 arise from excitatory, inhibitory, interaction between neurons in the brainstem. There is evidence of this in your figure, especially in response to 16 kHz and 32 kHz tones. You say in the paper: "Thresholds were defined as the lowest stimulus level that produced two or more discernible ABR waveforms (waves 1 to 5)." Measurements should have been based on the responses of the first peak of the ABR, if the measurements are meant to most closely reflect the responses of the OHCs. Later peaks reflect brainstem responses that can be modified by neuronal interaction. These comments pertain to your choice of measurement, which you found convenient, but DPOAE measurement, which only involve carefully placing a speculum in the auditory meatus of a lightly anaesthetised mouse, provides a more direct measure of OHC responses.

Thank you for raising this insightful point. We appreciate your detailed explanation and the relevant reference, which have drawn our attention to important methodological details we had previously overlooked.

These data confirmed our stimulus paradigm does not compromise ABR testing sensitivity. While our descending-level protocol does not affect standard ABR metrics (thresholds/amplitude), we acknowledge that subtler impacts on hearing sensitivity potentially detectable through compound action potential or other electrophysiological

measures. This methodological consideration provides valuable guidance for our future experimental design where ultra-high-frequency sensitivity preservation is paramount.

This is not really the case. Compound AP, or single fiber, measurements are invasive and indirectly signal the responses of the afferent fibres. CM measurements can be confusing and provide misleading information, especially in noise studies. See: J Neurophysiol 127: 1574-1585, 2022. doi:10.1152/jn.00402.2021.

We sincerely appreciate you sharing your deep understanding and perspective in the field of auditory electrophysiology. Thank you for providing the relevant reference as well.

Here, we revised one sentence in the method section to address the sequencing: (Line 601-603)

At each frequency, the test started from 90 dB SPL to ensure a clear response; and stepped down in 10 dB steps at high levels and then 5 dB when approaching the threshold.

I would keep this revised sentence. My comments are meant for consideration in designing further experiments and improvements in future experimental techniques.

We fully recognize that standardized auditory electrophysiological testing is essential for obtaining high-quality experimental data. We greatly appreciate your valuable input and suggestions, which will undoubtedly help guide the design of our future experiments and the refinement of our technical approaches.

5. Figure 1i. In the methods you state:

loudspeakers, respectively, positioned in a closed-field arrangement. A custom designed probe equipped with a low-noise ER10B+ microphone (TDT, USA) facilitated precise stimulus presentation and recording. DPOAEs were measured with a fixed f_2/f_1 ratio of 1.2, where the intensity of f_2 (L2) was set to 10 dB lower than f_1 (L1). Frequencies ranged from 5.6 to 32 kHz in half-octave intervals. For each f_2 , L2 was varied from 70 to 10 dB SPL in 5 dB steps. The $2f_1-f_2$ DPOAE amplitude and the

In the caption to Fig1i: -SC-cKO mice compared to Gsdmdflox/flox mice at 14 d-PNE (solid line), with no significant differences at Prenoise

Again it seems you began your measurements of DPOAE at high levels and descended the SPL to low levels, which may have prejudiced the sensitivity of the cochlea. It may have been helpful to design a protocol that extracted the DPOAEs continuously during the measurement. Such a protocol would enable you to monitor the DPOAEs directly and to store them and the raw data. This would have provided you with an immediate visual display of the sensitivity of the cochlea. Can you please explain why you chose your approach and not ones that enabled you to monitor DPOAE directly and to start from low SPLs to high, to preserve cochlear sensitivity.

Thanks for the comments and suggestions. Again, we start our DPOAE from high sound level

to ensure that we have a good result, which can be jeopardized by wrong insertion of earpiece and other actions in setting up. Do DPOAE in upward sequencing would end up with failure thereafter and waste more time, which is limited by the short effective time of the anesthesia. The anesthesia we choice is the best in terms of safety, but the appropriate anesthetics can be maintained for less than one hour with the light anesthesia protocol. This approach is taken due to the concern of losing animals.

Considering the measurements of both ABR and DPOAE, we must do everything quickly. In addition, the highest sound level for DPOAE is 70-80 dB SPL, noise exposure at level is generally safe, there is not data supporting a significant HL by noise exposure at such a low level, especially for such a short duration. Regarding the continuous monitoring of DPOAE,

We guess that the reviewer is concerned about the change of cochlear sensitivity due to the manipulation of the DPOAE test. The first reason for not taking the suggested approach is that our system does not allow us to do a continuous recording with an immediate visual display.

Our lab also present DPOAE data in the frequency domain, but we also have severe time restraints on our data collection and collect only 10s, rather than 100s of samples at each frequency. We thus get data that can be displayed very quickly. This technique might be further improved by a quasi-randomised technique where sampling could be stopped at particular frequencies and levels where the sampling has met a pre-determined signal to noise criteria. We have yet to implement such a technique.

A more intelligent strategy indeed appears highly beneficial for improving the quality of DPOAE data collection and accelerating experimental procedures, while also helping to preserve sensitivity of the cochlea.

DPOAE results are shown in frequency domain, while recording is the temporal waveform. Converting the signal in real time to frequency domain is very challenging. Therefore, we record the waveform in time domain for each frequency and level long enough for averaging and converting to frequency domain signal, which is saved. Moreover, continuous monitoring requires stimulation to be fixed in level and frequency, while a comprehensive evaluation on DPOAE requires it to be tested across a wide range of frequency and level. In conclusion, we set up our DPOAE protocol with careful planning and tried different ways. We are pleased to validate your proposed approach in follow-up studies. Should it demonstrate superior protection of auditory sensitivity, we will refine our current experimental protocol accordingly. We sincerely appreciate your rigorous perspective, which strengthens our methodology.

I have no comments on your responses from and including 6 onwards

Thank you for your insightful points.

Reviewer #1 (Remarks to the Author)

My comments are focused on the physiological methods employed in the investigation of

ROS-GSDMD feedback loop between hair cells 1 and supporting cells governs noise-induced hearing loss . I am satisfied with the authors' responses to my questions and comments.

Reviewer #2

We thank the reviewers for their thorough and constructive feedback. Addressing these comments has significantly strengthened our study. In revising the manuscript, we have conducted additional experiments and comprehensively addressed all points through careful analysis of available data.

This manuscript by Xiao et al has elucidated mechanisms by which non-sensory cochlear supporting cells (SCs) interact with sensory hair cells (HCs) through Gasdermin D (GSDMD) to influence noise-induced loss of hearing and HCs. Using SC specific GSDMD knockout along with neurophysiology and molecular biology approaches, authors report that noise-induced oxidative stress in HC causes activation of GSDMD in SC through EGFR-p-ERK-Caspase-11/1 pathway in SC. Such GSDMD activation in SCs, in turn augments oxidative damage in HCs leading to their death and thus, forming a positive loop between SCs and HCs. Knocking out GSDMD from SC completely abolishes oxidative damage in HCs. Moreover, blocking GSDMD in SCs but not in HCs or overexpressing antioxidant enzyme GPX4 in HCs significantly reduces noise-induced hearing loss and preserves outer hair cells. Overall, this is a rigorously designed and well written manuscript highlighting another critical role for SCs in regulating hair cell degeneration in the context of noise-induced hearing loss. This study also reveals molecular mechanisms of coupling of oxidative stress with inflammation in auditory pathophysiology.

Following are some concerns that need author's attention.

1. The major weakness in the manuscript is the evidence for how activated GSDMD-N in SCs increases oxidative stress in HCs, leading to their degeneration. It will strengthen the manuscript if authors could show that GSDMD-N localizes to mitochondria in HCs and mediate mitochondrial damage and increase inflammatory cytokines and ROS production.

Thank you for raising this point. We fully acknowledge the lack of evidence how GSDMD-N in supporting cells (SCs) mediate oxidative stress in hair cells (HCs) as the key limitation of our current study and have addressed it in our Discussion. But our data points the role of SC-HC link. It is NOT likely that GSDMD-N in HC mitochondria would play a role in this link, since *Gsdmd*-HC-cKO mice demonstrate that GSDMD activation within HCs is NOT the primary driver of noise-induced HC degeneration, as its conditional knockout (cKO) in HCs provided no protective effect (Extended Data Fig. S3f-k). Therefore, the effect of GSDMD-N in SCs on increasing oxidative stress in HCs likely occurs independently of GSDMD activation within the HCs themselves. In contrast to its unclear role in auditory research, recent study showed that GSDMD-N can be released via extracellular vesicles, and the fragment has the potential to induce pyroptosis in recipient cells.

Wright, S.S., Kumari, P., Fraile-Ágreda, V., Wang, C., Shivcharan, S., Kappelhoff, S., Margheritis, E.G., Matz, A., Vasudevan, S.O., Rubio, I., Bauer, M., Zhou, B., Vanaja, S.K., Cosentino, K., Ruan, J. et al. Transplantation of gasdermin pores by extracellular vesicles propagates pyroptosis to bystander cells. Cell 188, 280-291.e17 (2025).

This suggests a plausible alternative mechanism whereby GSDMD-N could be transferred indirectly from SCs to HCs—potentially localizing to the HC membrane to exert its damaging effects. Unfortunately, technical limitations prevented direct visualization of this potential transfer. Commercially available antibodies for GSDMD-N immunolabeling, which we obtained, exhibited non-specific binding in cochlear tissues. Critically, both antibodies failed validation using cochlear tissue from global *Gsdmd*-KO mice, confirming their lack of specificity in this context (***New data in Extended Data Fig. S11***). Consequently, we were unable to employ antibody-based methods to detect GSDMD-N fragments or pores within HCs or SCs.

We have expanded the Discussion section to incorporate the potential role of intercellular transfer of GSDMD-N as a noteworthy pathway warranting further investigation (**Line 469-480**):

Recent study revealed that extracellular vesicles can transplant GSDMD pores onto the plasma membrane of bystander cells, inducing their death⁴³. In the cochlear context, exosomes carrying heat shock proteins have been shown to protect against aminoglycoside-induced vestibular HCs' death via paracrine signaling⁴⁴. These findings collectively suggest a plausible mechanism whereby GSDMD-N could be transferred indirectly—potentially via extracellular vesicles—from supporting cells to hair cells, subsequently exerting its damaging effect by integrating into the hair cell membrane and directly inducing pyroptosis. However, spatial mapping of GSDMD activation remains technically constrained. Our commercially available GSDMD-N antibodies showed non-specific binding in cochlear tissue and failed validation using Gsdmd-KO cochlear sample (Extended Data Fig. S11), precluding reliable immunolabeling of GSDMD-N fragments in HCs or SCs.

2. Authors should show the expression of GSDMD after noise exposure by immunohistochemistry to verify its expression or upregulation in HCs after injury.

Recently, we performed additional cochlear immunohistochemistry using a GSDMD antibody rigorously validated against cochlear tissue from *Gsdmd* KO mice

Our analysis revealed no significant change in GSDMD immunolabeling within the HC region following noise exposure. Critically, this finding aligns with our functional data from *Gsdmd*-HC-cKO mice, which exhibited no protection against noise-induced cochlear damage (Extended Data Fig. S3f-k).

Taken together, these results demonstrate two key points:

- (1) GSDMD expression in HCs is not detectably upregulated by noise trauma.
- (2) HCs are not the primary cell type mediating GSDMD-dependent NIHL.

3. It is interesting that GSDMD KO or GPX4 overexpression does not seem to affect the ABR thresholds and DPOAE levels at 1 day post noise exposure but rather at 14 days. How do authors interpret these data. This should be discussed in the manuscript. Also, it will be important to discuss why the protection in GSDMD KO is selective to outer hair cells since GSDMD seems to be expressed by inner and outer pillar cells besides Claudius cells.

We appreciate your insightful questions regarding the temporal dynamics and cell-type specificity of protection.

(1) Temporal dynamics (Protection at 14 days, not 1 day):

Unlike the hearing loss caused by ototoxic drugs, NIHL has a specific temporal development: a large threshold shift is seen (and labelled as temporal threshold shift, or TTS) immediately after the noise exposure is ceased and is followed by a recovery period up to 2 weeks. The threshold shift left 2 weeks after the noise exposure is usually called permanent threshold shift (PTS). It has been recognized that the recoverable part of the hearing loss is largely due to the damage to the stereocilia that controls the mechanical-electrical transduction of HCs. This part of the damage has been found to be largely self-repairable and would not be influenced by GSDMD. However, PTS is primarily associated with damage to outer hair cell (OHC) bodies due to the accumulation of reactive oxygen species (ROS), leading to progressive and irreversible injury. The time-course of the apparent threshold change after the noise exposure is determined by the combination of the two processes. Our explanation for the protection observed at later time points, but not at 1 d-PNE, is that *Gsdmd* KO affects only the damage process responsible for PTS, which involves injury to cell bodies. In the wild-type (WT) mice of our experiment, the threshold recovery after 1 day post noise is small and is likely due to the accumulation of ROS stress, which largely “cancel” the effect of TTS recovery rooted from the self-repair of the stereocilia. In the KO mice, the accumulation of ROS is interrupted and therefore, we see much more “recovery” in ABR threshold. We are not sure if those explanations should be integrated into the manuscript. Please advise.

(2) No protection for IHCs (vs. OHCs):

The damage to inner hair cells (IHCs) by the noise could only be demonstrated by the synaptic loss since there is no IHC loss here. The reason for not seeing the protective effect by GSDMD-KO on the synapse count is likely due to the lack of connection between the synaptic damage and ROS stress. It is recognized that the synaptic damage is resulted from the over release of glutamate, the excitatory neurotransmitter for signaling from IHC to spiral ganglion neurons (SGN), which

damage the post-synaptic terminal of the synapses. The mechanism for the loss of the presynaptic component (the ribbon) is unclear right now. The damage of this part is related to the Ca^{2+} overload but not linked to the ROS stress which is the main target of GSDMD-related mechanisms for OHC protection. The failure of *Gsdmd* KO in preventing noise-induced ribbon synapse loss suggested that GSDMD activation is not a key mechanism for IHC synaptic injury. Although GSDMD is highly expressed in several subtypes of SCs, we cannot currently determine if GSDMD activation mediating noise-induced cochlear damage originates from specific SC subtype (e.g., outer pillar cells). Validating this would require generating GSDMD conditional knockout models targeting specific SC subpopulations. Furthermore, it remains possible that only OHCs, not IHCs, can establish a damaging ROS-GSDMD positive feedback loop with supporting cells. Both mechanisms represent plausible explanations for the observed OHC-selective protection in *Gsdmd* KO mice and merit further investigation.

We have expanded the Discussion section in **Line 507-512**:

Although GSDMD is highly expressed in several subtypes of SC, we cannot currently determine if GSDMD activation mediating cochlear damage originates from specific SC subtype (e.g., outer pillar cells). Furthermore, it remains possible that only OHCs, not IHCs, can establish a damaging ROS-GSDMD positive feedback loop with SCs, which may lead to the failure of Gsdmd KO in preventing IHC's ribbon synapse from noise-induced damage.

4. GSDMD knockdown in SCs (Pillar and Claudius cell) via AAV should be verified with immunohistochemistry besides Western blot.

We have now performed the additional immunohistochemistry verification as requested. Using the same antibody concentration and experimental conditions applied in our Western blot analysis, cochlear sections from WT and *Gsdmd*-SC-cKO mice were immunolabeled with the knockout-validated GSDMD antibody. Consistent with our

previous data, immunohistochemistry analysis confirmed a marked reduction in GSDMD immunoreactivity specifically within pillar and Claudius cells of *Gsdmd*-SC-cKO mice compared to control littermates. This result was incorporated into **Extended Data Fig. S2h**.

5. It will strengthen the manuscript if authors overexpress GSDMD-N protein in SCs and HCs to see if it reverses the protective effect of absence of GSDMD against noise-induced hearing loss.

We strongly agree that demonstrating functional reversal through GSDMD-N overexpression would significantly strengthen our conclusions. Initial characterization using dual-reporter mice confirmed efficient Cre-mediated recombination in SCs, but not HCs, with our AAV-ie-SC-Cre vector in adult mice. However, the efficiency of SC-promoter in driving Cre-expression significantly decreased in mature animals (>4 weeks old), limiting its utility for adult gain-of-function studies.

To address this technical constraint, we adopted a recently published SC promoter (*Lfng*) that maintains robust activity in adulthood. We constructed AAV-ie-*Lfng*-EGFP to validate promoter specificity and efficiency in adult mice. Following confirmation, we generated AAV-ie-*Lfng*-*Gsdmd*-N-Flag and delivered it neonatally to both WT and *Gsdmd* KO mice.

Due to revision constraints, we simultaneously packaged AAV-ie-Lfng-EGFP (transfection efficiency tracer) and AAV-ie-Lfng-*Gsdmd*-N-Flag (Addgene #80951). Adult WT mice injected with AAV-ie-Lfng-EGFP showed low transfection efficiency in supporting cells (SCs), confirming technical challenges. Paradoxically, both WT and *Gsdmd* KO mice injected with AAV-ie-Lfng-*Gsdmd*-N-Flag developed significant hearing loss. Western blot analysis of *Gsdmd* KO cochleae revealed a 35 kDa GSDMD-N band and a prominent 50 kDa band (hypothesized to represent oligomers). In WT cochleae, only the 50 kDa band was detected without the 35 kDa species. Given GSDMD-N's pore-forming oligomerization drives pathology, the exacerbated hearing loss in WT versus *Gsdmd* KO mice overexpressing GSDMD-N suggests accelerated oligomer assembly in endogenous GSDMD-expressing cells.

These results establish that SC-targeted GSDMD-N expression is sufficient to drive

cochlear degeneration independent of noise exposure, further underscoring its pivotal role in SC-mediated otopathology. The effect of this overexpression on hearing makes it impossible to test its potential to reverse the protection of *Gsdmd* KO against NIHL.

6. Figure 1, panel c, please check the immunolabel for hair cells; is it Spectrin or Myo7a.

Thank you for highlighting this detail. In Figure 1c, the immunolabel shows Myo7a (a hair cell-specific marker), not α II-Spectrin. Figure 1b displays α II-Spectrin labeling for cytoskeletal integrity. To prevent ambiguity, we have explicitly clarified these distinct immunolabels in the revised figure legend (*Line 138*).

7. It is unclear if the western blots shown in the manuscript are performed on protein lysates prepared from whole cochlea or isolated organ of Corti. If it is the former then that confounds the data due to the presence of other cochlear compartments like spiral ligament and stria vascularis that also expresses the factors in the pathway which are studied in the manuscript.

We appreciate your valid concern regarding the potential confounding effect of using whole cochlear lysates for Western blotting, given the expression of studied factors in compartments like the spiral ligament and stria vascularis. Isolation of the organ of Corti alone for Western blot analysis remains technically challenging in mice due to the cochlea's minute size and complex architecture. Consequently, as is common practice in the field, we utilized protein lysates prepared from whole cochlear tissue. We fully acknowledge this limitation and its implications for signal specificity.

While ideally we would perform immunostaining for all key molecules analyzed by WB, this faces a major hurdle——Lack of validated cleavage-specific antibodies: Commercially available antibodies specific for the cleaved/active forms (e.g., GSDMD-N, cleaved Caspase-1, Caspase-11 p26, mature IL-1 β) either do not exist or have not been reliably validated for specificity in cochlear tissue, preventing their

confident use for immunostaining in our system.

To address this inherent limitation and probe cell-specific molecular events, we implemented a complementary strategy:

(1) Cell-type-specific genetic manipulation: We employed HC-specific (*Gsdmd*-HC-cKO) and SC-specific (*Gsdmd*-SC-cKO) knockout mice, and HC-targeted GPX4 overexpression. This approach allows us to attribute functional outcomes and molecular changes primarily to the manipulated cell type, significantly reducing interpretative ambiguity from other cochlear compartments.

(2) Spatial validation via immunostaining: Recognizing that immunostaining can provide definitive spatial resolution, we performed critical validations:

1) Immunostained for the oxidative stress marker 4-HNE in HCs within *Gsdmd* KO and *Gsdmd*-SC-cKO mice (to assess downstream effects in HCs after SC manipulation).

2) Immunostained for p-ERK in HCs within GPX4-overexpression (OE) mice (to link SC manipulation to signaling pathway activation in HCs).

These IHC results provide direct spatial evidence supporting the proposed SC-HC signaling axis.

Thank you for prompting this important clarification. We agree that the use of whole cochlear lysates is a limitation for mechanistic granularity. We explicitly discussed this technical constraint and its potential implications in the revised Discussion section (**Line 514-520**):

*While whole cochlear lysates were used for WB to elucidate mechanism due to technical constraints, we mitigated interpretative limitations through complementary approaches: (1) Cell-type-specific manipulations functionally attributed molecular changes to targeted cells, such as *Gsdmd*-cKO in SCs and GPX4 overexpression in HCs; (2) Spatial immunostaining validated key SC-HC intercellular signaling events, including 4-HNE accumulation in HCs of *Gsdmd*-SC-cKO mice, and p-ERK level in SCs following HC-targeted GPX4 overexpression.*

8. Figure 3, it is unclear if GSDMD global or SCs conditional knockout was used to examine oxidative stress in HCs after acoustic trauma.

Thank you for raising this important clarification. For the data presented in Fig. 3, we utilized *Gsdmd* KO mice. However, acknowledging your valid concern about cell-type specificity, we have now performed additional experiments using *Gsdmd*-SC-cKO mice. WB analysis demonstrates that conditional knockout of GSDMD in SCs significantly reduces cochlear 4-HNE levels versus WT controls after noise exposure. Critically, immunostaining confirms markedly reduced 4-HNE within the hair cell region of *Gsdmd*-SC-cKO mice at 8 h post noise exposure. These results validate that GSDMD activation in SCs drives oxidative stress in HCs post-noise exposure, reinforcing our model of SC-mediated pathogenesis. We incorporated this data in the revised manuscript (*New data in Fig. 4e-h*).

9. How do authors interpret the data on preservation of IHC synapses with GPX4 overexpression but not with GSDMD knockout.

We appreciate your insightful question regarding the differential effects of GPX4 overexpression versus *Gsdmd* knockout on IHC synapse preservation. Under our noise

exposure paradigm, the primary sites of damage are OHCs and cochlear synapses. While *Gsdmd* KO specifically attenuated OHC loss, it did not significantly reduce synaptic loss (Fig. 1m). This indicates that GSDMD activation is not a primary driver of noise-induced synaptic injury. While GSDMD ablation protects OHCs via reduced ROS amplification, it does not prevent noise-induced GPX4 downregulation (new data: unchanged GPX4 levels in *Gsdmd* KO cochlea at 2h PNE, *New data in Extended Data Fig. S7*). Thus, *Gsdmd* KO fails to rescue the synaptic vulnerability potentially linked to GPX4 downregulation in cochlea.

In addition, the AAV-Php.eB vector used for GPX4 overexpression, driven by ubiquitous CMV promoter, leads to transgene expression not only in hair cells but also in SGNs. Prior evidence indicates that glutamate excitotoxicity thereby leading to activation of calcium-permeable AMPA receptors on the postsynaptic membrane, is a dominant mechanism for synaptic loss after noise trauma. GPX4's effects extend beyond the ROS-GSDMD axis, likely conferring direct antioxidant protection to SGNs against glutamate excitotoxicity.

Conclusion: The synapse protection seen with GPX4 OE, but not GSDMD KO, arises because:

- (1) Synaptic injury is primarily excitotoxicity-driven, not GSDMD-dependent;
- (2) GPX4 overexpression directly targets SGNs, counteracting synaptic damage;

(3) GPX4 confers cytoprotection broader than GSDMD pathway modulation alone.

10. Authors have indicated the use of male and female mice across the experiments; however the none of data hasn't been separated based on sex to determine if there is any sexual dimorphisms with the claimed phenomenon.

We appreciate your suggestion regarding potential sexual dimorphism in our observed phenomena and agree that sex-based analysis strengthens the study's rigor. We have now performed sex-stratified analyses for all key auditory functional phenotypes (ABR/DPOAE thresholds) across major experimental groups (*Gsdmd* KO, *Gsdmd*-SC-cKO, GPX4-OE, and relevant controls). No statistically significant sex-dependent differences were detected in: (1) Baseline hearing thresholds, (2) Noise-induced threshold shifts, and (3) Protective effects of genetic manipulations.

These results indicate that the core phenomena described in our study are consistent across sexes under the experimental conditions tested. The complete sex-stratified datasets have been added to the Supplementary Information (*New data in Extended Data Fig. S10*). Thank you for prompting this important validation.

11. Please check the references, since some of references related to ERK inhibitors in noise-induced hearing loss have not been cited.

We thank you for identifying this important literature gap. We have carefully evaluated the suggested work and incorporated it into our revised manuscript. Recently, we observed that ASN007 (a selective potent and oral inhibitor for ERK phosphorylation) also blocked ROS-GSDMD signaling. These data demonstrated p-ERK's pathogenic role in driving GSDMD activation and oxidative stress. (*New data in Extended Data Fig. S9, Line 385-386*)

Reviewer #2 (Remarks to the Author):

The authors have addressed the comments appropriately in the revised manuscript.

Reviewer #3

We thank the reviewers for their thorough and constructive feedback. Addressing these comments has significantly strengthened our study. In revising the manuscript, we have conducted additional experiments and comprehensively addressed all points through careful analysis of available data.

This study by Xiao et al. presents an intriguing set of data on the intercellular signaling pathway between cochlear hair cells (HCs) and supporting cells (SCs) in response to noise exposure. Traditionally, HC death and synapse loss due to noise exposure have been discussed without considering SC interactions. However, SCs have recently been recognized for their diverse immune functions, which vary depending on the cochlear environment. This study contributes to this evolving understanding by demonstrating aspects of SC immune function. The authors effectively utilized two interesting AAV vectors, each selectively targeting either HCs or SCs. However, additional experiments using these vectors could be considered, as outlined below. Furthermore, spatial information is crucial when discussing signaling pathways between neighboring cells. To enhance this aspect, immunostaining should be presented alongside Western blotting.

Below are some specific notes that may help improve the manuscript.

1. Page 7, the authors used a cochlear SC-specific promoter for AAV-ie to efficiently transfer Cre to SCs. However, they do not specify which SC-specific gene's promoter was chosen for this experiment, either here or in the Methods section. Providing this detail would improve clarity and reproducibility.

We thank you for raising this valid point regarding promoter specificity, which is essential for reproducibility. The supporting cell (SC)-specific promoter used in our AAV-ie constructs was provided by OBio Technology (China, Shanghai), with commercial approval by Professor Guisheng Zhong's laboratory (ShanghaiTech

University). Due to confidentiality obligations and ongoing intellectual property protection, the promoter information may be disclosed at further time. We regret any inconvenience this limitation may cause and appreciate your understanding of these contractual constraints. We have now explicitly acknowledged this source in the Methods section (***Line 556-558***).

2. Page 7, Western blotting is used to confirm the effectiveness of *Gsdmd*-SC-cKO in Figure S2. Since the *Gsdmd* expression in SCs of the WT cochlea is clearly shown by immunostaining in Figure 1, its decrease in SCs of the cKO cochlea could be exhibited also by immunostaining.

In revised manuscript, we performed additional immunohistochemical analysis comparing GSDMD expression in *Gsdmd*-SC-cKO mice versus WT controls (***New data in Extended Data Fig. S2h***). Consistent with our WB data (Extended Data Fig.S2g), immunostaining revealed a marked reduction or absence of GSDMD signal specifically within SCs notably in pillar cells and Claudius cells of *Gsdmd*-SC-cKO cochlea. These results provide spatial validation of efficient GSDMD ablation achieved via AAV-ie-SC-Cre-mediated knockout.

3. Page 8, there was no difference in ABR thresholds and OHC loss after noise exposure between *Gsdmd*-HC-cKO mice and *Gsdmd* flox mice. However, the authors suggested

in Page 7 that GSDMD in non-SC cells may play a protective role in acoustic trauma, rather than the harmful role in SCs. If so, Gsdmd-HC-cKO should show elevated ABR thresholds and severer OHC loss as compared with Gsdmd flox mice.

Our idea about the protective role of GSDMD in non-SCs is based on the difference between global KO and SC selective KO of *Gsdmd*, since the global KO exhibits less protection as compared with SC selective KO (Fig. 1k). We then postulate that GSDMD in non-SC cells may play a beneficial effect. The global KO eliminate this good part of GSDMD effect, so that less effect of protection is seen as compared with SC-selective KO. This demonstrates the complexity of GSDMD-involved cellular process. We do not see how the idea stated above would lead logic error suggested by the last sentence in the reviewer s comment above, since GSDMD KO in hair cells (HCs) is irrelevant with the issue.

4. Page 10, the elevation of GSDMD-N following noise exposure is demonstrated by Western blotting. However, other cochlear cell types besides SCs may also express GSDMD in response to noise exposure. To exclude this possibility, it would be important to confirm that GSDMD-N elevation occurs specifically in SCs using immunostaining.

We fully agree that cell-type-specific localization of GSDMD-N would strengthen our mechanistic claims. To address this, we rigorously validated commercially available GSDMD-N antibodies (CST #36425 and HUABIO #ER1901-37). However, both antibodies lacked specificity in cochlear tissue, as evidenced by detectable bands in *Gsdmd* KO mice samples, precluding their reliable use for immunostaining in our system.

We acknowledge that using whole cochlear lysates limits mechanistic resolution. We explicitly discussed this technical constraint and its implications in the revised Discussion, ensuring proper contextualization of results. We appreciate this valuable

suggestion for improved scientific rigor. (*Line 476-480 and 514-520*)

However, spatial mapping of GSDMD activation remains technically constrained. Our commercially available GSDMD-N antibodies showed non-specific binding in cochlear tissue and failed validation using Gsdmd-KO cochlear sample (Extended Data Fig. S11), precluding reliable immunolabeling of GSDMD-N fragments in HCs or SCs.

While whole cochlear lysates were used for WB to elucidate mechanism due to technical constraints, we mitigated interpretative limitations through complementary approaches: (1) Cell-type-specific manipulations functionally attributed molecular changes to targeted cells, such as Gsdmd-cKO in SCs and GPX4 overexpression in HCs; (2) Spatial immunostaining validated key SC-HC intercellular signaling events, including 4-HNE accumulation in HCs of Gsdmd-SC-cKO mice, and p-ERK level in SCs following HC-targeted GPX4 overexpression.

5. Page 11, Caspase-1/11 double KO mice are used to conclude that Caspase-11 is the primary mediator of GSDMD cleavage during noise exposure. However, to rule out potential interactions between Caspase-1 and Caspase-11, conducting experiments with Caspase-11 single KO mice would be important.

We agree that delineating the individual contributions of caspase-1 versus caspase-11 is essential for mechanistic clarity. According to your suggestion, we have obtained *Casp11* KO mice (generously provided by Prof. Feng Shao, National Institute of Biological Sciences, Beijing) and subjected them to our standard noise exposure paradigm. Western blot analysis of cochlear lysates at 8 h post-noise exposure revealed a complete absence of GSDMD-N cleavage product in *Casp11* KO mice group, but robust GSDMD-N signal in WT controls. (**New data in Fig. 2d, g; Line184-187**). This demonstrates that caspase-11 is necessary and sufficient for noise-induced GSDMD activation in our model, effectively ruling out compensatory interactions with caspase-1. The phenotype mirrors that of *Casp1/11* DKO mice (*Fig. 2b, e*), confirming caspase-11's primary role.

6. Page 13-14, global *Gsdmd* KO mice are used to conclude that activated GSDMD enhances noise-induced oxidative stress. However, to fully support the authors hypothesis throughout the manuscript, experiments using *Gsdmd*-SC-cKO mice instead of the global ones would be crucial for clarifying the intercellular signaling pathway between SCs and HCs in response to sound exposure. Also, 4HNE expression in HCs is exhibited by immunostaining, but showing the SC layer in addition to the HC layer in this immunostaining would be important.

We appreciate your suggestions regarding cell-specific validation and spatial analysis. Critical experiments using *Gsdmd*-SC-cKO mice demonstrate significantly attenuated 4-HNE levels in cochlear lysates via Western blotting, with immunostaining confirming reduced oxidative stress specifically in outer hair cells (OHCs). Spatial analysis

revealed concurrent 4-HNE elevation in both HCs and SCs of noise-exposed *Gsdmd*^{flox/flox} controls, whereas *Gsdmd*-SC-cKO mice exhibited selective 4-HNE reduction in HCs despite persistent oxidative stress in SCs. This compartmentalized protection pattern confirms that SC-specific GSDMD ablation shields HCs from oxidative damage despite ongoing SC stress, providing direct evidence for GSDMD-mediated intercellular (non-cell-autonomous) oxidative stress propagation. These results fully align with global KO data, establishing SCs as the primary site of GSDMD-dependent pathology. We incorporated this new data in the revised manuscript (**Fig. 4e-h**).

7. Page 16, the authors discuss the significance of GPX4 in HCs based on Western blotting of the whole cochlea. Immunostaining for GPX4 to demonstrate its specific expression in HCs would be of great interest also here.

Prior study demonstrate that HC-specific *Gpx4* knockout induces hearing impairment, whereas SC-specific ablation shows no auditory phenotype, indicating GPX4's functional indispensability in HCs. In direct response to your suggestion, our immunostaining reveals robust GPX4 expression in both HCs and SCs. Notably, noise exposure triggered concomitant GPX4 downregulation in both cell types at 2h post noise exposure (2 h-PNE).

8. Page 17, GPX4 overexpression is shown to reduce noise-induced GSDMD activation and oxidative stress. To confirm that GPX1 does not contribute to this effect, additional experiments with GPX1 overexpression should be included.

We thank the reviewer for this suggestion. To specifically address the potential role of GPX1, we have performed additional experiments: Western blot analysis of cochlear lysates from mice injected with AAV-CMV-*Gpx1* revealed that GPX1 overexpression failed to significantly reduce 4-HNE levels or suppress GSDMD cleavage following noise exposure (*New data in Extended Data Fig. S6; Line 283-285*). These results demonstrate that, unlike GPX4, GPX1 overexpression is insufficient to confer protection against noise-induced oxidative stress or GSDMD activation.

9. Page 18, sarifice is a typo in Figure 4.

We thank you for catching this error. The typo "sarifice" in Figure 4 has been corrected to "sacrifice" in the revised manuscript. We have diligently conducted full proofreading of the text and figures to ensure manuscript accuracy and prevent similar errors.

10. Page 20, the authors demonstrate the critical role of GPX4 in HCs in downregulating ROS, using GPX4 overexpression system in HCs. However, in order to exclude the possibility that GPX4 may also function in SCs, additional experiments with GPX4 overexpression in SCs would be important.

We appreciate the reviewer's suggestion to clarify the cell-type specificity of GPX4 function. To directly address whether GPX4 overexpression in SCs confers protection, we generated AAV.ie-SC-*Gpx4*. Our findings reveal that GPX4 overexpression specifically in SCs failed to significantly mitigate noise-induced hearing loss.

Furthermore, Western blot analysis demonstrated no significant reduction in GSDMD-N levels following noise exposure in AAV.ie-SC-*Gpx4* mice compared to non-AAV controls. This result, combined with previous literature demonstrating that GPX4 knockout in SCs does not affect hearing, whereas knockout in HCs leads to auditory dysfunction, collectively suggests that GPX4 plays a critical role in maintaining HC function under both physiological and noise-exposure pathological conditions. The specific functional contribution of GPX4 in SCs warrants further investigation. These new data have been added to the Supplementary Information (*New data in Extended Data Fig. S8*).

11. Page 27, the authors conclude that phosphorylation of ERK in SCs occurs prior to HC damage in response to noise exposure. They also cite a study (Ref. 33) showing that ERK activation in SCs leads to HC damage. However, ERK is also well known for

its role in cell survival and proliferation, and some studies report that ERK and Akt activation in SCs can promote HC protection. These findings highlight the multifunctional nature of ERK in cochlear SCs. It would be valuable for the authors to discuss this broader context.

We thank the reviewer for this thoughtful comment. We have revised the Discussion to acknowledge its dual roles and emphasize that the specific outcome likely depends on the nature and duration of the stimulus. (*Line 445-452*)

Previous study also showed that p-ERK level is up-regulated in cochlear SCs upon mechanical and noise damage^{34,35}. Our pharmacological inhibition of p-ERK activity blocked noise-induced GSDMD-dependent damage cascades, indicating p-ERK exacerbates cochlear injury. This contrasts with prior work showing Erk2-cKO in HCs increased susceptibility to NIHL, particularly enhancing IHC loss, which suggesting ERK2 mediates protective signaling in HCs³⁶. This functional dichotomy demonstrates that EGFR-ERK signaling exhibits cell type- and context-dependent heterogeneity.

We thank the reviewer #3 for these insightful and constructive suggestions, which have significantly improved the quality of our manuscript. In response, we have provided detailed point-by-point explanations and have supplemented additional data wherever feasible within the revision timeframe.

Reviewer #3 (Remarks to the Author):

In the revised version of this manuscript, the authors added data using Caspase-11 single KO mice, Gsdmd-supporting cell (SC)-cKO mice, AAV.ie-CMV-Gpx1 mice, and AAV.ie-SC-Gpx4 mice. These datasets are impressive and provide valuable insights for readers into the complex intercellular signaling pathways between cochlear hair cells (HCs) and supporting cells. However, I would like to suggest additional revisions on some specific points. Each numbered comment corresponds to the numbering in the authors' response. At the end, I have included a crucial comment addressing an important enigma in the signaling pathway.

3. As the authors suggest, GSDMD in non-SC cells (including HCs) may play a beneficial role

based on Fig. 1h, i, k. However, in Extended Data Fig. S3h, k, GSDMD-HC-cKO mice did not exhibit higher ABR thresholds or greater OHC loss than control mice. If GSDMD in HCs does not contribute to protection against acoustic trauma, in which cochlear cell types do the authors hypothesize that GSDMD exerts its protective effect? For example, could this involve other cell components within the organ of Corti or specific populations in the lateral wall? Clarifying this point would help readers better understand the proposed cellular targets and mechanisms.

We agree with the reviewer's perspective that elucidating the functional role of GSDMD in other cochlear cell types is of significant interest. Further investigation on this front will, however, rely on technical advances—particularly the generation of cell-type-specific *Gsdmd* conditional knockout models, such as in cells located in the lateral wall. To date, there have been relatively few reports on the protective roles of GSDMD. Seminal, though limited, evidence — such as that from studies on macrophages—suggests that GSDMD may mediate protective functions.

(1). Chi, Z., Chen, S., Yang, D., Cui, W., Lu, Y., Wang, Z., Li, M., Yu, W., Zhang, J., Jiang, Y., Sun, R., Yu, Q., Hu, T., Lu, X., Deng, Q. et al. Gasdermin D-mediated metabolic crosstalk promotes tissue repair. *Nature* (2024).

(2). Manickam, V., Gawande, D.Y., Stothert, A.R., Clayman, A.C., Bataikina, L., Warchol, M.E., Ohlemiller, K.K. & Kaur, T. Macrophages Promote Repair of Inner Hair Cell Ribbon Synapses following Noise-Induced Cochlear Synaptopathy. *J Neurosci* 43, 2075-2089 (2023).

Previous literature has demonstrated that macrophages play a protective role in noise-induced cochlear injury, leading us to hypothesize that GSDMD within macrophages may contribute to protection during acoustic trauma. Additionally, given the functional heterogeneity among macrophage subtypes in cochlear pathology, it may be necessary to generate conditional knockouts of *Gsdmd* in distinct macrophage subpopulations. We aim to systematically address this in future studies and have incorporated this perspective in Discussion section (**Line 499-506**).

6. I would like to suggest that the authors include in the revised manuscript the images of the SC layer (prepared in their response) showing that Sox2-positive SCs and 4HNE-positive cells (HCs) do not overlap. Adding these data will make the point clearer to readers and strengthen the manuscript.

We appreciate the reviewer's suggestion and have now included this figure as supplementary material in the revised manuscript (*Extended Data Fig. S5*).

7. In Fig. 6c, d, the authors demonstrate that GPX4 is downregulated in HCs at 2 hours post noise exposure, but they do not provide evidence that GPX4 expression in SCs is unchanged. The cryosection images provided in their response are not explicit on this point. I therefore suggest that the authors include representative images of the SC layer in the manuscript, in order to clearly demonstrate HC specificity in the role of GPX4.

We understand that you are seeking further clarification regarding the cell-specific role of GPX4, particularly whether it functions uniquely in HCs rather than SCs. Previous studies have demonstrated that mice with *Gpx4* knockout specifically in HCs—but not in SCs—exhibit significant hearing loss, which can be rescued by antioxidant treatment.

Liu, Z., Zhang, H., Hong, G., Bi, X., Hu, J., Zhang, T., An, Y., Guo, N., Dong, F., Xiao, Y., Li, W., Zhao, X., Chu, B., Guo, S., Zhang, X. et al. Inhibition of *Gpx4*-mediated ferroptosis alleviates cisplatin-induced hearing loss in C57BL/6 mice. *Mol Ther* 32, 1387-1406 (2024).

This supports the view that GPX4 plays an essential and specific role in HCs under physiological conditions. In line with your previous suggestion, we performed and observed that overexpression of GPX4 specifically in HCs—but not in SCs—significantly attenuates GSDMD-mediated injury

signaling. Moreover, in response to your comment here, we performed additional GPX4 staining in the SC layer. Although a slight decreasing trend in GPX4 expression was observed in SCs at 2 hours after noise exposure, the difference was not statistically significant.

Taken together with previous findings and our results, we propose that GPX4 primarily functions in HCs rather than SCs, both under normal conditions and in response to noise-induced injury. We truly appreciate your insightful comments, which have been invaluable in guiding these additional analyses.

11. In the revised manuscript, the authors acknowledge the multifunctional nature of ERK in SCs but there are more key studies demonstrating ERK's protective effect on HCs. For completeness and balance, I recommend citing representative work in this area, including the study from Kyoto University (Insulin-like growth factor 1 inhibits hair cell apoptosis and promotes the cell cycle of supporting cells by activating different downstream cascades after pharmacological hair cell injury in neonatal mice. Netrin 1 mediates protective effects exerted by insulin-like growth factor 1 on cochlear hair cells.), which specifically demonstrated ERK-mediated HC protection. Including these references will provide readers with a more accurate and nuanced understanding of ERK's dual roles in cochlear SCs.

We appreciate this suggestion and thank the reviewer for pointing out these relevant studies. We have now incorporated the recommended reference (*Line 451-452* in the revised manuscript), along with other key works highlighting the protective role of ERK in hair cells. This addition provides a more balanced and comprehensive perspective on the multifaceted functions of ERK signaling in the cochlea, thereby enhancing the objectivity and quality of our manuscript.

There are several types of SCs in the organ of Corti, including pillar cells, Deiter's cells, Hensen's cells, Claudius' cells, and inner phalangeal cells. Each SC type has distinct phenotypes and roles in supporting HCs. Overall, it would be helpful if the authors could clarify which specific SC types constitute or are most crucial in the intercellular signaling pathway described in this manuscript.

We fully agree that this is an important point worthy of further investigation. Addressing this question would require the development of promoters capable of distinguishing between subtypes of supporting cells, which in turn would enable subtype-specific conditional knockout experiments. As revised in the Discussion (*Line 508-511*), this aspect deserves more systematic elucidation in future studies. Progress in this area will also benefit from broader collaborative efforts. We would greatly value your continued suggestions as we advance this research.

Reviewer #3 (Remarks to the Author)

In the revised version of this manuscript, the authors added some discussion in response to my earlier comments. While no new experiments were performed, they outlined future directions. I have one additional suggestion:

7. I would like to suggest that the authors include in the revised manuscript the images and fluorescence analysis of GPX4 expression showing that there was no significant difference in its expression in supporting cells between pre-NE and 2 h-PNE. Adding these data will help readers better understand the intercellular signaling pathway between hair cells and supporting cells.

We appreciate the reviewer's suggestion. We have included the GPX4 expression images as Supplementary Fig. 7 in the revised manuscript.

Many thanks for very carefully and fully dealing with my questions about your paper. I have made some comments below on some of your responses that you may wish to consider, and most of them do not require any changes to the changes you have made to your paper in response to my original comments. Below I have reproduced by original comments and responses and have underlined those responses where I have made further comments. You may wish to consider my further questions and comments in sections 2 and 3 They are intended for clarity and understanding of your data. Other comments may influence future experimental design.

1. To assess the hearing of the mice, the authors used two non-invasive techniques.

They measured the ABR and DPOAEs. ABR measurements provide an overall measure

of the hearing sensitivity of the mice in their experiments. This measure does not, on its own, enable the experimenters to discover the precise origin of any alteration in the hearing of the mice from control.

We acknowledge your valid point regarding ABR's limitations in localizing the precise origin of hearing alterations. However, ABR remains the best option for non-invasive, frequency-selective assessment of hearing sensitivity, which is the major concern to the present study. It is well recognized that NIHL impacts mainly on the function of outer hair cells (OHCs) which are responsible for the hearing sensitivity. Therefore, the location of the hearing loss is already clear and is not the focus of the present study.

Furthermore, ABR is the ONLY method that can be tested repeatedly (as needed in the present study) in small rodents.

This is patently not true. The standard, and much more direct, non-invasive procedure for testing OHC activity is the measurement of DPOAEs. This technique has been used in mammals including baby microchiroptera weighing less than 5gm to new borne human babies without disturbing their sleep... DPOAEs are measured routinely and regularly to test the hearing of mice at all stages of development.

While ABR cannot differentiate sensory versus neural contributions, we mitigated this limitation through complementary approaches: (1) DPOAEs specifically evaluated OHC function. (2) Cytocochleograms quantified sensory cell loss. (3) Ribbon synaptic counts assessed integrity of IHC-to-SGNs transmission. This multimodal strategy

ensures ABR threshold shifts primarily reflect OHC-mediated sensitivity loss - the core focus of our NIHL investigation.

2. DPOAE measurements reflect the performance of the nonlinear amplification of the outer hair cells. As a first point, and to enable an assessment of the quality of the data, will the authors please state in the methods the intrinsic distortion levels of the measurement system and hence its threshold for measuring acoustic distortion.

Thanks for the suggestion. We revised the method section by adding the examination of the intrinsic distortion as shown below: (**Line 613-618** in the revised manuscript)

The potential intrinsic distortion and the noise floor were examined by recording the sound via the coupling cavity with the stimulation used for DPOAE. The noise floor of the testing system was below 0 dB SPL and there was no significant intrinsic distortion when tested via the physical coupling cavity utilized for the DPOAE. DPOAE threshold was defined as the interpolated value of f2 intensity required to generate a 0 dB SPL DPOAE.

Many thanks for this ,but a more intuitive and direct way of presenting acoustic distortion in the sound system is to say that it is so many dB below the level of the primary tones. My guess is that for your system, it is about 80 dB. This means, for example, that there could be 10 dB SPL of distortion in tones presented at 90 dB SPL. Can you please express distortion levels in this format.

3. Figures 1e, h. What is “dB” relative to? Is it a value common to all measurements such as SPL? Or is it relative to each measurement? Ideally it should be referred to a commonly accepted standard, like SPL. It is easier to understand. Could you please, therefor, clarify what is meant by “ dB ” and ideally, for increased clarity and understanding, convert to SPL.

Figure 1e and h concern level shift, which is relative to baseline values, they should not be relative to a standard like SPL but relative to each other, since a shift would be the same in dB no matter whether SPL or other metric is used.

What baseline value were you using? Was it the threshold stimulus tone SPL at each particular frequency? For example, did you have to use tones close to 100 dB SPL at 32 kHz to elicit a signal from Gsdmd KO at 1 day and ~ 70 dB SPL at day 14? the absolute SPL you use is important because it shows if the stimuli fall within the linear dynamic range of your sound system.

Please note that dB SPL is

used in d and g where the absolute thresholds are addressed using SPL. In such cases, a standard value (20 uPa in air) is used as reference.

4. In your methods you state:

“Briefly, acoustic stimuli were presented via custom-designed plastic tubing connected to an MF1 Multi-Field Magnetic Speaker (TDT, USA) positioned in the ear canal.

Puretone

stimuli of 10- ms (0.5- ms rise/fall) were presented at 21.1/s across intended frequencies, starting from 90 dB SPL and decreasing in steps of 5 dB. Evoked responses

were amplified 10,000× (PA4 bio-amplifier, TDT) and averaged 500 times with 0.3 to 3 kHz band-pass 565 filter.”

Thus, at the very beginning of your measurements you presented 500 intense tone bursts.

The higher frequency regions of the mouse cochlea (above about 30 kHz) are very noise

sensitive, and your regime could prejudice the sensitivity of the cochlea. In the methods, or in the results section, please explain why you chose to deliver the tones descending from a high to low levels instead of ascending from low to high levels, or perhaps with a degree of randomization in the presentation of both frequencies and sound levels, which would tend to preserve hearing sensitivity.

Thanks for your comments. The reason to start testing sequence from high sound level (90 dB SPL) is to make sure that the mouse has a good ABR response. This approach reduces uncertainty due to any wrong setting for the test. The impact of high-level stimulation on hearing sensitivity is unlikely based on several facts. First, we do not

need to repeat 500 times in each trial at high sound level(s), as less as 100-200 repeats is usually good enough to obtain clear responses. Secondly, we do not really do 5-dB steps at high sound level but drop by 10 or even 20 dB. 5-dB steps are used when the test approaches the threshold. In this way, the exact noise exposure is trivial. Thirdly, the tone burst exposure at 90 dB SPL with the rate used in our testing does not significantly change the ABR threshold even up to thousands of repeats. Ideally, randomization in level should minimize the potential impact of high-level tone exposure on thresholds. But our equipment does not allow such a randomization. However, we are confident that the method we use is safe. In addition, we stuck the method in the same way across all the tests. Therefore, the results should reasonably reflect the real change in hearing sensitivity. Finally, the down-sequencing in ABR test is generally acceptable.

To address whether our descending-level protocol (90 → lower dB SPL) affects cochlear sensitivity, we performed direct comparisons in a cohort of mice (n = 6) using:

- (1) Low-to-high intensity ascending protocol
- (2) Our high-to-low intensity descending protocol

on the same ears, with all frequencies tested at 500 repetitions per stimulus level.

We found no significant differences in ABR thresholds at any frequency (8, 16, 32kHz) and in ABR wave-I amplitude at different sound level (60, 70, 80 and 90 dB SPL).

According to the data you presented, this does not appear to be the case. This finding is to be expected, according to recent papers e.g <https://doi.org/10.1523/ENEURO.0250-18.2018>. In the quoted paper, the noise level was 100 dB SPL, which is 10 dB SPL higher than the loudest test tones used in your experiments. All four ABR peaks were reduced in size by the noise exposure, but peaks 2 to 4 were sensitized. Peak 1 arises directly from the response of the auditory nerve. Peaks 2 to 4 arise from excitatory, inhibitory, interaction between neurons in the brainstem. There is evidence of this in your figure, especially in response to 16 kHz and 32 kHz tones. You say in the paper: "Thresholds were defined as the lowest stimulus level that produced two or more discernible ABR waveforms (waves 1 to 5)." Measurements should have been based on the responses of the first peak of the ABR, if the measurements are meant to most closely reflect the responses of the OHCs. Later peaks reflect brainstem responses that can be modified by neuronal interaction. These comments pertain to your choice of measurement, which you found convenient, but DPOAE measurement, which only

involve carefully placing a speculum in the auditory meatus of a lightly anaesthetised mouse, provides a more direct measure of OHC responses.

These data confirmed our stimulus paradigm does not compromise ABR testing sensitivity. While our descending-level protocol does not affect standard ABR metrics (thresholds/amplitude), we acknowledge that subtler impacts on hearing sensitivity—potentially detectable through compound action potential or other electrophysiological measures. This methodological consideration provides valuable guidance for our future experimental design where ultra-high-frequency sensitivity preservation is paramount.

This is not really the case. Compound AP, or single fiber, measurements are invasive and indirectly signal the responses of the afferent fibres. CM measurements can be confusing and provide misleading information, especially in noise studies. See: J Neurophysiol 127: 1574–1585, 2022. doi:10.1152/jn.00402.2021.

Here, we revised one sentence in the method section to address the sequencing: (**Line 601-603**)

At each frequency, the test started from 90 dB SPL to ensure a clear response; and stepped down in 10 dB steps at high levels and then 5 dB when approaching the threshold.

I would keep this. My comments are meant for consideration in designing further experiments and improvements in future experimental techniques.

5. Figure 1i. In the methods you state:

“In DPOAEs testing, the primary tones f_1 and f_2 were delivered through two MF1 loudspeakers, respectively, positioned in a closed-field arrangement. A custom designed probe equipped with a low-noise ER10B+ microphone (TDT, USA) facilitated precise stimulus presentation and recording. DPOAEs were measured with a fixed f_2/f_1 ratio of 1.2, where the intensity of f_2 (L2) was set to 10 dB lower than f_1 (L1). Frequencies ranged from 5.6 to 32 kHz in half-octave intervals. For each f_2 , L2 was varied from 70 to 10 dB SPL in 5 dB steps. The $2f_1-f_2$ DPOAE amplitude and the surrounding noise floor were extracted for offline analysis.”

In the caption to Fig1i:

“ i. DPOAE amplitudes were higher in Gsdmd-SC-cKO mice compared to Gsdmdflox/flox mice at 14 d-PNE (solid line), with no significant differences at Prenoise exposure (dashed line).”

Again it seems you began your measurements of DPOAE at high levels and descended the SPL to low levels, which may have prejudiced the sensitivity of the cochlea. It may have been helpful to design a protocol that extracted the DPOAEs continuously during the measurement. Such a protocol would enable you to monitor the DPOAES directly and to store them and the raw data. This would have provided you with an immediate visual display of the sensitivity of the cochlea. Can you please explain why you chose your approach and not ones that enabled you to monitor DPOAE directly and to start from low SPLs to high, to preserve cochlear sensitivity.

Thanks for the comments and suggestions. Again, we start our DPOAE from high sound level to ensure that we have a good result, which can be jeopardized by wrong insertion of earpiece and other actions in setting up. Do DPOAE in upward sequencing would end up with failure thereafter and waste more time, which is limited by the short effective time of the anesthesia. The anesthesia we choice is the best in terms of safety, but the appropriate anesthetics can be maintained for less than one hour with the light anesthesia protocol. This approach is taken due to the concern of losing animals. Considering the measurements of both ABR and DPOAE, we must do everything quickly. In addition, the highest sound level for DPOAE is 70-80 dB SPL, noise exposure at level is generally safe, there is not data supporting a significant HL by noise exposure at such a low level, especially for such a short duration.

Regarding the continuous monitoring of DPOAE, we don't really understand the point of the reviewer in terms of "immediate visual display of the sensitivity of the cochlea". We guess that the reviewer is concerned about the change of cochlear sensitivity due to the manipulation of the DPOAE test. The first reason for not taking the suggested approach is that our system does not allow us to do a continuous recording with an

immediate visual display.

Our lab also present DPOAE data in the frequency domain, but we also have severe time restraints on our data collection and collect only 10s, rather than 100s of samples at each frequency. We thus get data that can be displayed very quickly. This technique might be further improved by a quasi-randomised technique where sampling could be stopped at particular frequencies and levels where the sampling has met a pre-determined signal to noise criteria. We have yet to implement such a technique.

DPOAE results are shown in frequency domain, while recording is the temporal waveform. Converting the signal in real time to frequency domain is very challenging. Therefore, we record the waveform in time domain for each frequency and level long enough for averaging and converting to frequency domain signal, which is saved. Moreover, continuous monitoring requires stimulation to be fixed in level and frequency, while a comprehensive evaluation on DPOAE requires it to be tested across a wide range of frequency and level. In conclusion, we set up our DPOAE protocol with careful planning and tried different ways. We are pleased to validate your proposed approach in follow-up studies. Should it demonstrate superior protection of auditory sensitivity, we will refine our current experimental protocol accordingly. We sincerely appreciate your rigorous perspective, which strengthens our methodology.

6. Also, please state the SPL at which you measured the DPOAEs. This does not appear to be given in the caption of Figure 1. Additionally, please define dBV and why DPOAE magnitude is not given as SPL.

We thank the reviewer for identifying these important issues. Regarding Fig. 1:

(1) The DPOAE data were acquired in down sequency starting at 80 dB SPL F1 stimulation

(2) The unit was incorrectly labeled as dBV due to a documentation error. Per TDT's DPOAE conversion protocol ([https://www.tdt.com/docs/dpoae-user-guide/convertingdbv-](https://www.tdt.com/docs/dpoae-user-guide/convertingdbv-to-db-spl)

to-db-spl), values should be presented in dB SPL.

We further acknowledge your valuable suggestion about DPOAE amplitude interpretation. Given the nonlinear response pattern (including plateau effects) of DPOAE responses across intensity levels, we agree that single-intensity amplitude measurements risk ambiguous interpretation. Accordingly, we have replaced the DPOAE amplitude-frequency curves (responses at L2 = 70 dB SPL) with frequencyspecific

DPOAE thresholds in the revised figures (**Fig. 1i, 5g, 5h**). This approach provides a more physiologically meaningful characterization of OHC function.

7. I strongly suggest that the presentation of DPOAE ISO response curves would have given a clearer understanding of the frequency dependence of DPOAE generation in the cochlea under different conditions. Amplitude can give a confusing impression because, especially at high levels, e.g F1 at 50 dB SPL, DPOAE increase with level is very nonlinear and can reach a plateau, depending also on stimulus frequency.

Thanks. We assume that the ISO response refers to the level-frequency curve to show that, at each frequency, what sound level can evoke a target DPOAE amplitude across frequencies. We think that revised threshold-frequency curve serves this purpose since the threshold is determined by a certain amount of response amplitude above the noise floor (**Fig. 1i, 5g, 5h**). We revised the method section by adding the definition of the DPOAE threshold in **Line 617-618** of the revised manuscript and copied here:

DPOAE threshold was defined as the interpolated value of f2 intensity required to generate a 0 dB SPL DPOAE.

8. Figure 1k. There appears to be no OHC loss for frequencies below about the 3.5 mm location, from apex. According to a well-known mouse cochlear frequency map (Müller et al., Hearing Research 202 (2005) 63–73), this is close to the 34 kHz frequency place. Do these findings, in part, account for why there appears to be no complete recovery from NIHL?

To our knowledge, it is common to see discrepancy between functional deterioration and OHC loss in the study of NIHL. The threshold elevation is better matched by OHC loss in very high frequency region but in low frequency region, we often see threshold

shift without any OHC loss. It is concluded that OHCs may have been functionally damaged but keep morphological integrity as shown by the most common methods for HC count.

(see Chen G D, Fechter L D. The relationship between noise-induced hearing loss and hair cell loss

in rats. *Hear Res*, 2003, 177(1-2): 81-90 as an exemplary research).

Our result showed also the discrepancy at the low frequency region below 34 kHz, suggesting incomplete recovery of hearing sensitivity.

9. Figure 1m. Please state that this figure is based on sampling in the 32 kHz region where OHC loss is expected to be minimal, or indeed absent.

The sampling location for synapse investigation is stated in the figure legend of 1l as 32 kHz region. 1m is the statistical evaluation of sample in 1l. To make it clearer, we revised the figure legend of 1m as follow (**Line 154-156**):

Statistics of samples shown in 1 from 32 kHz region where OHC loss is minimal, or indeed absent, showing no statistical differences in synapse density (averaged from eight IHCs per cochlea) across groups.

10. A remarkable finding, not commented on in the paper, is that there is no hair cell loss at the location of the noise exposure or at a location a half octave above this where noise is most effective at desensitizing the cochlea (Cody and Johnstone, 1981 *J. Acoust.*

Sot. Am. 70, 707-711; Cody and Russell, 1985, *Nature* 315, 662-665). Instead, hair cell loss is confined to frequencies above about 34 kHz, a region where the cochlea is most sensitive noise insult. Thus presentation of an 8-16 kHz noise band causes no measurable change in sensitivity in the cochlea region apical to this, it causes loss of sensitivity to all frequency regions at and basal to the 8-16 kHz band, and causes 0% - 20% OHC loss at frequencies a half octave above the 8-16 kHz band and up to 100% OHC loss at frequencies above this (in the 40 kHz – 90 kHz region). Could it be that there are differences in the action of Gasdermin D on NIHL at the bases and hook region

of the cochlea from its action in the more apical regions? Perhaps this point could be raised in the discussion.

The famous half-octave law in NIHL is not established on HC loss. The study by Cody and Johnstone (JASA 1981) was on TTS of single unit ANF. No data on OHC loss was seen in this paper. This is also true for the study by Cody and Russell 1985 (Cody A R, Russell I J. Outer hair cells in the mammalian cochlea and noise-induced hearing loss. *Nature*, 1985, 315: 662-665.) in which no HC count data was available. As we responded above, there is a big discrepancy between threshold shifts and OHC loss in the low frequency region after NIHL was established: often time, the threshold shift is NOT matched by the amount of OHC loss.

Using tone of middle frequency (8-16) is a common practice in the studies of NIHL in mice

(1. Furman A C, Kujawa S G, Liberman M C. Noise-induced cochlear neuropathy is selective for

fibers with low spontaneous rates. J Neurophysiol, 2013, 110(3): 577-586. DOI: 10.1152/jn.00164.2013

2. Kujawa S G, Liberman M C. Adding insult to injury: cochlear nerve degeneration after "temporary" noise-induced hearing loss. J Neurosci, 2009, 29(45): 14077-14085. DOI: 29/45/14077 10.1523/JNEUROSCI.2845-09.2009

3. Maison S F, Liberman M C. Predicting vulnerability to acoustic injury with a noninvasive assay

of olivocochlear reflex strength. J Neurosci, 2000, 20(12): 4701-4707.)

Exposure at such frequency can cause NIHL spread to almost the whole cochlea, more towards the high-frequency end. With the intensity and duration used in the present experiment, OHC loss is expected to be limited to the very high-frequency region (particularly in the hook area), and not in the region half an octave above the frequency region of acoustic overstimulation.

The question "Could it be that there are differences in the action of Gasdermin D on NIHL at the bases and hook region of the cochlea from its action in the more apical

regions?" is difficult to answer since we do not have evidence to support or reject the speculation. The important fact to our study is the protective effect against noise-induced OHC loss appears and can only be demonstrated in the region where OHC loss is seen. Recent literature demonstrates that noise exposure induces a gradient increase in NOX2-positive OHCs from apical to basal turns, with NOX2 deficiency partially attenuating cochlear damage. This suggests basal OHCs experience higher oxidative stress, potentially predisposing this region to initiate the damaging ROS-GSDMD positive feedback loop.

Qi, M., Gao, Z., Qiu, Y., Wang, R., Tian, K., Yue, B., Zhang, X., Zhang, P., Wu, Z., Zhu, Q., Liu, Z.,

Ma, Z., Zhou, X., Han, Y., Chen, J. et al. NOX2 Contributes to High-Frequency Outer Hair Cell

Vulnerability in the Cochlea. Advanced Science (2025).

However, due to the current lack of a specific antibody reliably detecting the GSDMDN fragment, it is currently unfeasible to definitively determine via immunolabeling whether noise exposure indeed triggers a higher gradient of GSDMD activation specifically within the basal turn. We have revised the Discussion about this limitation:

(Line 476-480, New data in Extended Data Fig. S11)

However, spatial mapping of GSDMD activation remains technically constrained: Commercially available GSDMD-N antibodies showed non-specific binding in cochlear tissue and failed validation using Gsdmd-KO controls, precluding reliable immunolabeling of GSDMD-N fragments in HCs or SCs.